# Epicardium-derived cells organize through tight junctions to replenish cardiac muscle in salamanders

Elif Eroglu [1 ✉], Christopher Y. T. Yen [1], Yat-Long Tsoi [1], Nevin Witman [1,2], Ahmed Elewa [1,4], Alberto Joven Araus [1], Heng Wang [1,5], Tamara Szattler [1], Chimezie H. Umeano [1,6], Jesper Sohlmér [1], Alexander Goedel [1,7], András Simon [1 ✉] and Kenneth R. Chien [1,3 ✉]

**The contribution of the epicardium, the outermost layer of the heart, to cardiac regeneration has remained controversial due to a lack of suitable analytical tools. By combining genetic marker-independent lineage-tracing strategies with transcriptional profiling and loss-of-function methods, we report here that the epicardium of the highly regenerative salamander species *Pleurodeles waltl* has an intrinsic capacity to differentiate into cardiomyocytes. Following cryoinjury, CLDN6+ epicardium-derived cells appear at the lesion site, organize into honeycomb-like structures connected via focal tight junctions and undergo transcriptional reprogramming that results in concomitant differentiation into de novo cardiomyocytes. Ablation of CLDN6+ differentiation intermediates as well as disruption of their tight junctions impairs cardiac regeneration. Salamanders constitute the evolutionarily closest species to mammals with an extensive ability to regenerate heart muscle and our results highlight the epicardium and tight junctions as key targets in efforts to promote cardiac regeneration.**

Regeneration of cardiac muscle in adult salamanders and teleost fish has been attributed to dedifferentiation and proliferation of pre-existing cardiomyocytes[1–9]. Cardiomyocytes can re-enter the cell cycle in the adult mammalian heart but this does not lead to functional regeneration after injury due to the low frequency of cardiomyocytes that complete the cell cycle and proliferate[10–12]. Hence, most efforts aim to identify paracrine cues that can stimulate cardiomyocyte proliferation[13–15]. This is however challenging given that cardiomyocytes are multinucleated and polyploid in adult mammals, in contrast to naturally regenerative non-mammalian vertebrates[13,16].

An alternative approach involves resident progenitor cells that become activated following injury and differentiate into cardiomyocytes. Epicardium, the epithelial layer enclosing the heart, is a prime candidate for such an approach as it is a source of multipotent progenitors during development[17,18]. Studies examining the contribution of epicardium to the regeneration of the zebrafish heart have highlighted its role as a source of paracrine signalling and extracellular-matrix molecules as well as a modulator of inflammation[19]. Genetic lineage tracing and transplantation studies in zebrafish did not show epicardial cell differentiation into cardiomyocytes[7,20–22]. The innate epicardial fate in mammals following injury has remained unresolved due to the paucity of tools for cell type-specific tracing, as current genetic markers also label epicardial derivatives and/or non-epicardial cell types[23–26]. Nevertheless, it has been reported that following stimulation with factors such as thymosin-ß4 and VEGF, epicardial cells differentiate into cardiomyocytes at a low frequency, indicating the regenerative potential of the epicardium. However, infrequent conversion rates and diminished lineage plasticity after injury necessitate further studies to identify ways to enhance the regenerative response of the epicardium[27–29].

Earlier studies of salamander heart regeneration established the ability of salamander cardiomyocytes to proliferate[3,30–32]. However, the role of the epicardium has not been investigated further than its upregulation of regeneration-specific matrix proteins and genetic markers of the salamander epicardium have not been identified[33]. Here we established a genetic marker-independent lineage tracing strategy in the salamander *Pleurodeles waltl* and show low-level conversion of epicardial cells into myocytes during homeostasis, a process that is greatly expanded in response to cryoinjury. Using single-cell RNA sequencing (scRNA-seq), we identify the tight junction protein CLDN6 as a specific marker of the homeostatic epicardium. Following cryoinjury, epicardium-derived cells (EPDCs) migrate to the injury site, form honeycomb-like structures decorated by CLDN6+ focal tight junctions and differentiate into cardiomyocytes, which engraft into the myocardium. Transcriptional profiling and trajectory analyses reveal the expression of the key cardiac transcription factors *Gata4*, *Gata6*, *Foxc1* and *Foxc2* during this cell-fate transition. Finally, we show that both ablation of CLDN6+ differentiation intermediates as well as disruption of tight junctions impairs cardiac regeneration in salamanders.

## Results

**Epicardium gives rise to cardiomyocytes under homeostasis.** To study the potency of the post-metamorphic salamander epicardium, we developed a lineage tracing strategy that selectively

[1]Department of Cell and Molecular Biology, Karolinska Institutet, Stockholm, Sweden. [2]Department of Clinical Neuroscience, Karolinska Institutet, Stockholm, Sweden. [3]Department of Medicine, Karolinska Institutet, Stockholm, Sweden. [4]Present address: Department of Genetics, Microbiology and Statistics, Faculty of Biology, University of Barcelona, Barcelona, Spain. [5]Present address: College of Animal Sciences and Technology, Huazhong Agricultural University, Wuhan, China. [6]Present address: Department of Molecular Medicine and Gene Therapy, Lunds Universitet, Lund, Sweden. [7]Present address: Klinik und Poliklinik für Innere Medizin I, Klinikum Rechts der Isar, Technical University of Munich, Munich, Germany. ✉e-mail: elif.eroglu@ki.se; andras.simon@ki.se; kenneth.chien@ki.se

labels the epicardium in an unbiased genetic marker-independent manner. As the epicardium forms a barrier between the underlying myocardium and the surrounding pericardial fluid (Fig. 1a,b)[34], we microinjected a cell-permeant Cre recombinase (TAT-Cre) into the pericardial cavity of tgTol2(*CAG:loxP-Cherry-loxP-H2B::YFP*)[Simon] (hereafter, *Cherry-loxP-H2B::YFP*) reporter animals (Fig. 1c,d)[35,36]. Thirty hours post microinjection (h.p.i.), nuclear Cre expression was confined to the epicardium and no labelled cells were found in the underlying myocardium (Extended Data Fig. 1a,b). Accordingly, we detected the emergence of yellow fluorescent protein (YFP) signal in the epicardial cells (Extended Data Fig. 1a). At 40 h.p.i., TAT-Cre-induced recombination yielded strong YFP expression in the outermost pan-cytokeratin (pan-CK)-expressing epicardial layer, detected with a polyclonal antibody recognizing a broad spectrum of keratins (Extended Data Fig. 1a,b). A labelling efficiency of 38% was observed and spontaneous recombination did not occur in the vehicle-injected animals (Extended Data Fig. 2a–c). Reflecting previous reports of TAT fusion proteins binding to the extracellular matrix surrounding muscle tissue, we observed some extracellular matrix-associated Cre fluorescence in the myocardium (Extended Data Figs. 1a and 2d)[37]. However, this did not transduce cardiomyocytes, as nuclear Cre signal was absent in the myocardium (Extended Data Fig. 1a,b).

From 40 h.p.i. we found rare YFP+ cells embedded within the myosin heavy chain (MHC)-expressing myocardium, which increased in number by 96 h.p.i. (Fig. 1e,f,i and Extended Data Fig. 2e). To assess the identity of these EPDCs, we performed lineage tracing in salamanders carrying the conditional reporter tgTol2(*CAG:loxP-GFP-loxP-Cherry*)[Simon] (hereafter *GFP-loxP-Cherry*), in which CHERRY expression on recombination is cytoplasmic and facilitates the assessment of cellular morphology (Fig. 1g). At 96 h.p.i., CHERRY+ cells coexpressing α-actinin lacked myofibrillar structures and had the appearance of immature cardiomyocytes (Extended Data Fig. 2f). In contrast, we observed CHERRY+ cells coexpressing MHC and α-actinin at 11 days post injection (d.p.i.; Fig. 1h and Extended Data Fig. 2g) that by virtue of their size, morphology and myofibrillar structure resembled mature cardiomyocytes. We found an increase in the number of labelled epicardium-derived cardiomyocytes over the course of 11 d (Fig. 1i), suggesting that ongoing low-level conversion of epicardial cells to cardiomyocytes contributes to cardiac homeostasis in salamanders.

**Injury induces epicardial cell-to-cardiomyocyte conversion.** To investigate whether epicardium contributes to cardiac regeneration in salamanders, we established a cryoinjury model. Using a liquid nitrogen-cooled probe, we injured the ventricular apex and analysed the extent of regeneration at 7, 14, 28, 64 and 210 days post cryoinjury (d.p.ci.; Fig. 2a,b and Extended Data Fig. 2h). The procedure consistently damaged approximately 25% of the ventricle, as measured at 7 d.p.ci. (Fig. 2b). Cryoinjured ventricles showed loss of myocardium as well as the deposition of collagen and fibrin (Fig. 2a). The lesion was reduced to approximately 12% and the fibrin clot started to resorb by 14 d.p.ci. (Fig. 2a,b). At 28 d.p.ci., the scar was reduced and composed of more prominent collagen networks (Fig. 2a,b). At 64 d.p.ci., the small remaining lesion—detectable by remnants of collagen—was interspersed with myocardial cells, indicating the replacement of the injury site by regenerating myocardium (Fig. 2a,b). There were no signs of injury and tissue organization was restored 210 d.p.ci. (Fig. 2a,b). We also monitored the regeneration process using echocardiography to measure the injury size for each animal and observed a gradual recovery, reflecting the histological analyses (Extended Data Fig. 2i). These results show that the salamander heart can regenerate damaged muscle tissue in response to cryoinjury.

To evaluate the cellular contribution of epicardium to the regenerating myocardium, we performed lineage tracing as described

earlier, followed by cryoinjuries at 40 h.p.i. (Fig. 2c). Absence of nuclear Cre signal in the myocardium following injury was confirmed (Extended Data Fig. 3a–c). Cryoinjury decreased the number of labelled epicardial cells, resulting in approximately 29% of the remaining epicardium being labelled, as assessed at 48 h post cryoinjury, a time point before the initiation of epicardial proliferation (Extended Data Fig. 3d–g). At 21 d.p.ci., we found CHERRY+ cells coexpressing MHC (Fig. 2d) and α-actinin (Fig. 2e) within the Tenascin-C+ apical region of the myocardium (Fig. 2f). We found an increase in the number of labelled cardiomyocytes of approximately 14-fold compared with the corresponding regions in the sham-operated hearts (Fig. 2g and Extended Data Fig. 3h), indicating a substantial expansion of epicardial cell conversion into cardiomyocytes after injury. Importantly, epicardium-derived cardiomyocytes analysed at 60 d.p.ci. showed elongated morphology and sarcomere formation, indicating long-term engraftment (Fig. 2h,i). Together, these results show that epicardium-derived cardiomyocytes not only contribute to the myocardium during homeostasis but also regenerate the myocardium after injury. We also observed occasional CHERRY+vimentin+ mesenchymal cells and CHERRY+α-smooth muscle actin+ smooth muscle cells/myofibroblasts, but not endothelial cells, indicating that epicardial cells may also give rise to non-myocyte lineages (Extended Data Fig. 3i–l).

To address whether epicardium-derived cardiomyocytes expand—that is, one epicardial cell gives rise to several cardiomyocytes—we utilized tgTol2(*CAG:Nucbow*)[Simon] reporter animals[38]. Here recombination results in random combinations of different fluorescent proteins to facilitate clonal analysis (Extended Data Fig. 4a). At 21 d.p.ci., approximately 54% of same-colour clones were comprised of two or more clonally related cells, indicating cell division (Extended Data Fig. 4b–f). Furthermore, we found that 22% of epicardium-derived cardiomyocytes were PCNA+ at 21 d.p.ci., with occasional expression of phospho-histone H3 (Extended Data Fig. 4g–j), which suggested that clonal expansion could result from the proliferation of epicardium-derived cardiomyocytes.

**Tight junction genes specifically mark epicardium.** To follow transcriptional changes accompanying regeneration and discover specific markers of the salamander epicardium, we performed scRNA-seq on 2,386 live cells collected from sham-operated and regenerating hearts at 7, 14 and 28 d.p.ci. (Fig. 3a, Extended Data Fig. 5a–d and Supplementary Table 1). Seventeen distinct cell clusters were identified through unbiased clustering and marker-gene expression (Fig. 3b and Supplementary Table 1). As expected, immune cells represented the majority of cells recovered at 7 and 14 d.p.ci., whereas clusters identified to be endothelial or endocardial cells (Clusters 0 and 5) and myocyte-like cells (Cluster 6) were less abundant at these time points (Fig. 3b and Extended Data Fig. 5e). A small cluster (Cluster 16) was annotated as transitioning cells based on their transient appearance at 7 and 14 d.p.ci. as well as expression of the epithelial-to-mesenchymal transition (EMT) markers (Fig. 3b,d, Extended Data Fig. 5e,f and Supplementary Table 1).

Three clusters (Clusters 9, 11 and 12) expressed the embryonic epicardial genes *Wt1*, *Tcf21* and *Tbx15* (*Tbx18* homologue in *P. waltl*)[39–41] (Fig. 3c). Cluster 9 cells expressed *Dkk2* (Fig. 3c,d), a Wnt-pathway inhibitor previously indicated in the specification of pro-epicardial cells, and *Hoxa5*, a marker of the axolotl epicardium (Fig. 3d)[42,43]. These cells also expressed regulators of angiogenesis such as the endothelial orphan G protein-coupled receptor *Adgrl4*, the transcription factor *Sox18* and the endothelial Rho guanine exchange factor *Fgd5* (Fig. 3d)[44–47], suggesting that they might represent subepicardial endothelial cells. Providing support for this, Gene Ontology characteristics related to angiogenesis and vasculature development were enriched in Cluster 9 (Extended Data Fig. 6a).

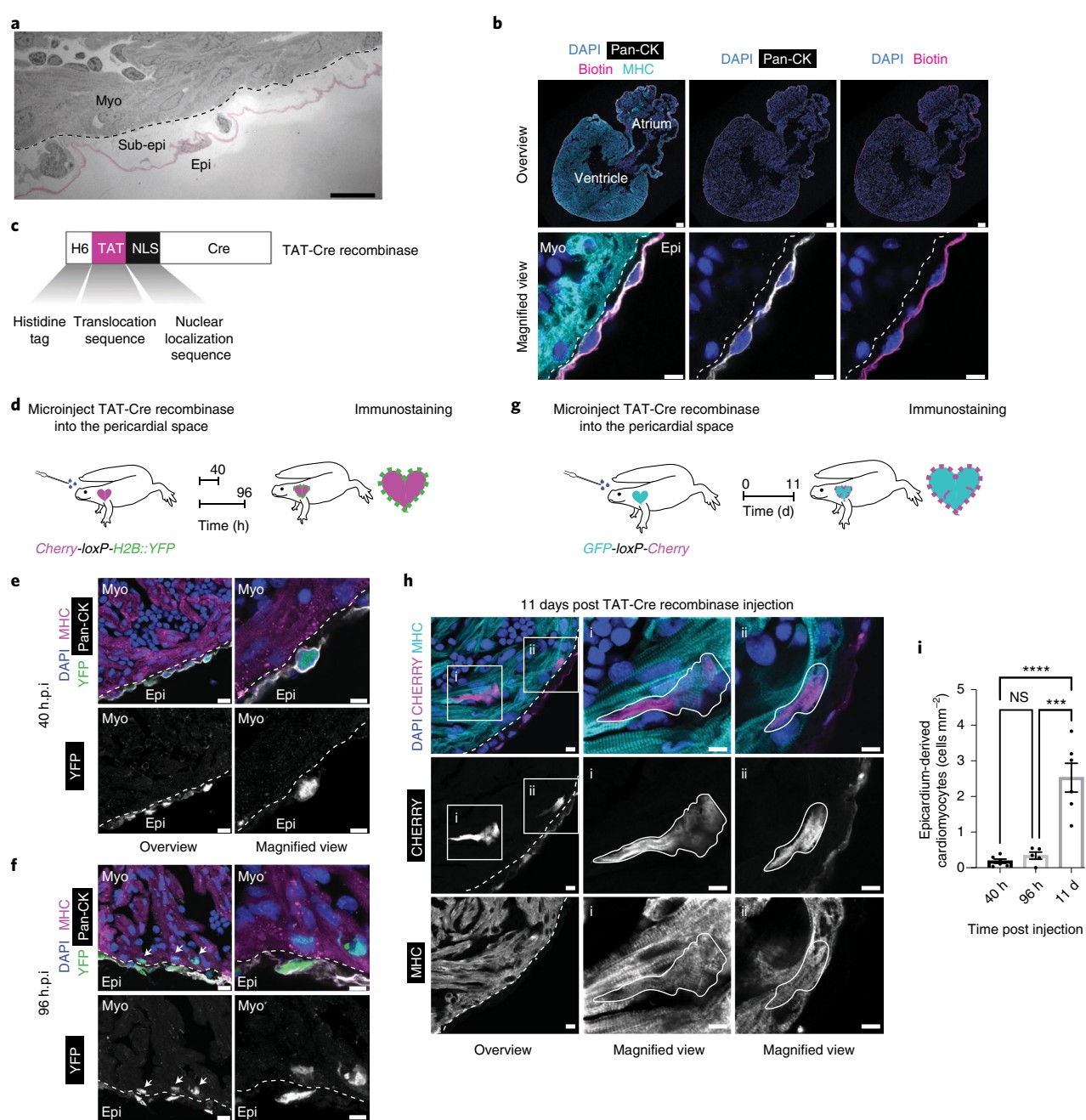

**Fig. 1 | Salamander epicardium gives rise to cardiomyocytes under homeostasis. a**, TEM image showing the organization of epicardium, subepicardium and myocardium. The epicardial layer is pseudocoloured. The dashed line marks the myocardium–subepicardium border. Scale bar, 25 μm. Data represent three animals. **b**, Salamander epicardium forms a barrier, as shown in a biotin permeability assay. Representative immunostainings for 4,6-diamidino-2-phenylindole (DAPI), MHC (cardiomyocyte marker), pan-CK (epicardial cell marker) and biotin. Biotin staining is confined to the epicardium. Data represent four animals. Scale bars, 200 μm (top) and 10 μm (bottom). **c**, Cell-permeable TAT-Cre recombinase fusion protein. **d**, Experimental design for permanent labelling of epicardial cells in *Cherry-loxP-H2B::YFP* reporter salamanders. **e**, TAT-Cre recombinase microinjection into the pericardial cavity of *Cherry-loxP-H2B::YFP* allows selective labelling of epicardial cells. Representative immunostainings for pan-CK, MHC and YFP at 40 h.p.i. Labelled cells occupy the epicardial layer. Data represent six animals. **f**, Epicardial-to-myocardial cell conversion during homeostasis. Representative immunostainings for pan-CK, MHC and YFP on TAT-Cre recombinase-injected *Cherry-loxP-H2B::YFP* ventricle sections at 96 h.p.i. Labelled cells are found in the epicardial and myocardial layer (arrows). Data represent five animals. **g**, Experimental design for permanent epicardial-cell labelling in *GFP-loxP-Cherry* animals. **h**, EPDCs in the myocardium co-stained for cytoplasmic CHERRY and MHC indicating epicardial cell-to-cardiomyocyte conversion. Representative immunostainings for DAPI, CHERRY and MHC at 11 d.p.i. Data represent six animals. CHERRY+MHC+ cells are outlined. Magnified views of the numbered regions in the overview images (left) are shown (middle and right). **e,f,h**, Scale bars, 20 μm (overview), 10 μm (magnified view). The myocardium–epicardium border is marked by dashed lines. **i**, The number of labelled cardiomyocytes in the ventricular area were quantified at 40 h.p.i., 96 h.p.i. and 11 d.p.i. The number of epicardium-derived cardiomyocytes increased with time. Data are the mean ± s.e.m. of *n* = 6 (40 h.p.i. and 11 d.p.i.) and 5 (96 h.p.i.) animals. One-way analysis of variance (ANOVA) with Tukey's multiple comparisons test; NS, not significant (*P* = 0.9099); ***P* = 0.0001 and *****P* < 0.0001. Epi, epicardium; sub-epi, subepicardium; and myo, myocardium.

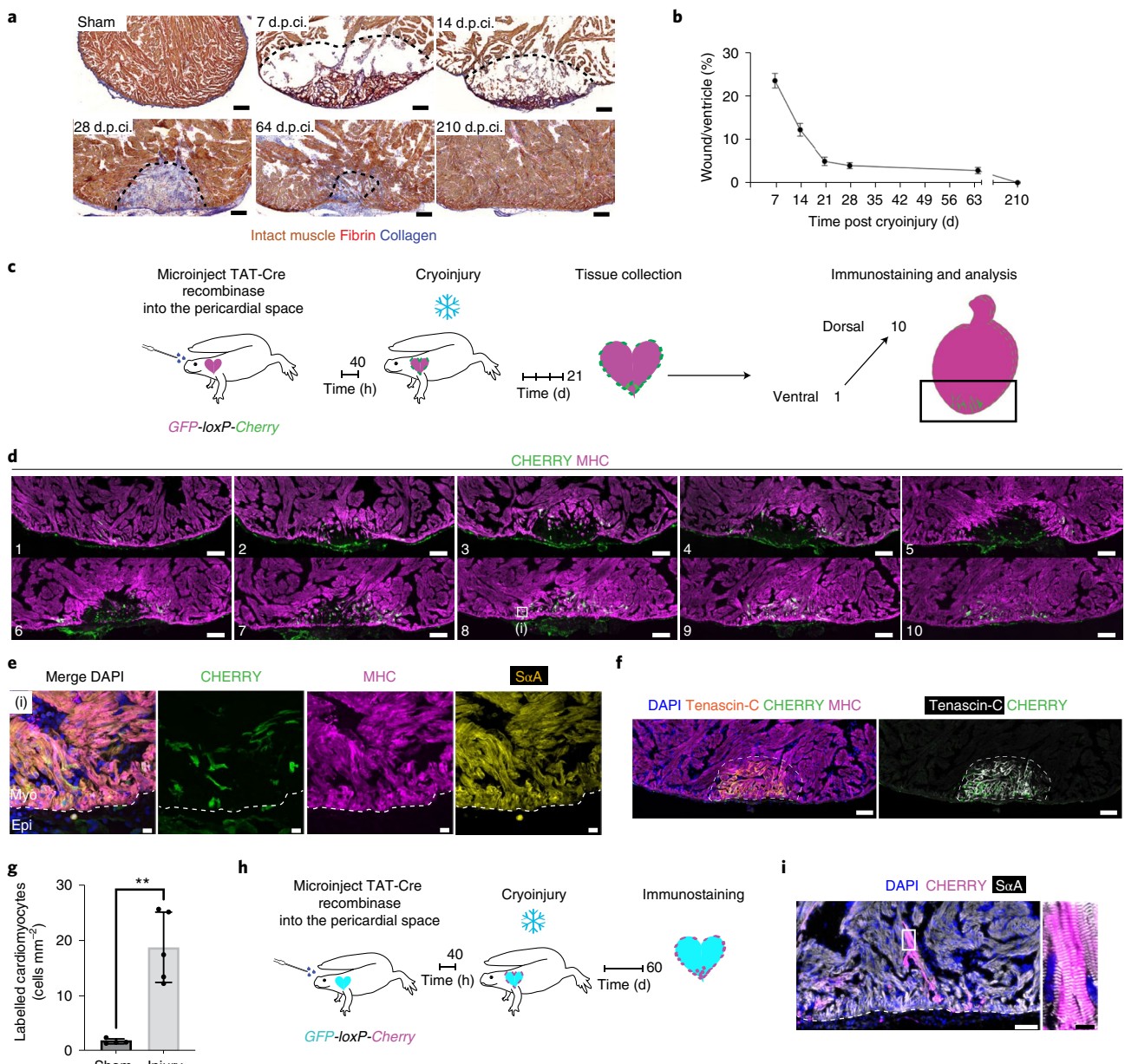

**Fig. 2 | Epicardial cell-to-myocyte conversion increases in response to cryoinjury. a**, Time course of regeneration following cryoinjury. Acid fuchsin orange G (AFOG) staining of the ventricular apex post cryoinjury. **b**, The size of the ventricular wound decreased with time. Data are the mean ± s.e.m. of n = 6 (7 d.p.ci.), 4 (14 and 210 d.p.ci.), 3 (21 and 64 d.p.ci.) and 5 (28 d.p.ci.) animals. **c**, Schematic of the permanent epicardial cell labelling combined with cryoinjury experiment. **d**, Epicardium-derived cardiomyocytes occupy the regenerating area. Representative images of CHERRY and MHC immunostaining at 21 d.p.ci. Ten of 258 representative frontal plane sections collected from an animal are shown, representing five animals. **a,d**, Scale bars, 200 μm. **e**, Magnified view of the box in frontal plane no. 8 in **d**. Immunostaining for CHERRY, MHC and sarcomeric α-actinin (SαA). The dashed line shows the myocardium (myo)–epicardium (epi) border. Scale bars, 10 μm. **f**, Tenascin-C marks the regenerating area. Representative images of DAPI, Tenascin-C, CHERRY and MHC immunostaining at 21 d.p.ci. on the section preceding frontal plane no. 9 in **d**. Note that the injured muscle at this section depth shows cardiac muscle regeneration, assessed by MHC staining, still displaying Tenascin-C positivity. Scale bars, 200 μm. **g**, Injury induces expansion of the epicardium-derived cardiomyocyte pool. The CHERRY⁺MHC⁺ cells were enumerated in sham-operated (n = 4 animals) and cryoinjured hearts (n = 5 animals) at 21 d.p.ci. Unpaired Student's t-test with Welch's two-tailed correction; \*\*P = 0.0033. Data are the mean ± s.d. **h**, Schematic of the experiment to assess epicardium-derived cardiomyocyte engraftment. **i**, Epicardium-derived cardiomyocytes show long-term engraftment to the regenerated myocardium. High-resolution immunofluorescence images demonstrate sarcomeric organization in epicardium-derived cardiomyocytes, with transverse orientation of α-actinin at z-lines overlapping with CHERRY signal. Overview (left) and magnified view of the region in the white box (right). Data represent eight animals. Scale bars, 100 μm (left) and 10 μm (right).

Cluster 11 cells expressed the mesothelial marker *Lrrn4* (ref. [48]); the epithelial markers *Cdh1*, *Epcam* and *Alcam*[49–51] and the cell-adhesion molecule *Flrt2*, a known epicardial marker[52] (Fig. 3c–e and Extended Data Fig. 6b,c), indicating this as the main epicardial sheet enveloping the heart. In addition, cells in this cluster expressed the *Claudin* (*Cldn*) gene family members *Cldn6*, *Cldn7* and *Cldn15*, important components of tight junction formation[53] (Fig. 3c–e). Accordingly, Gene Ontology term

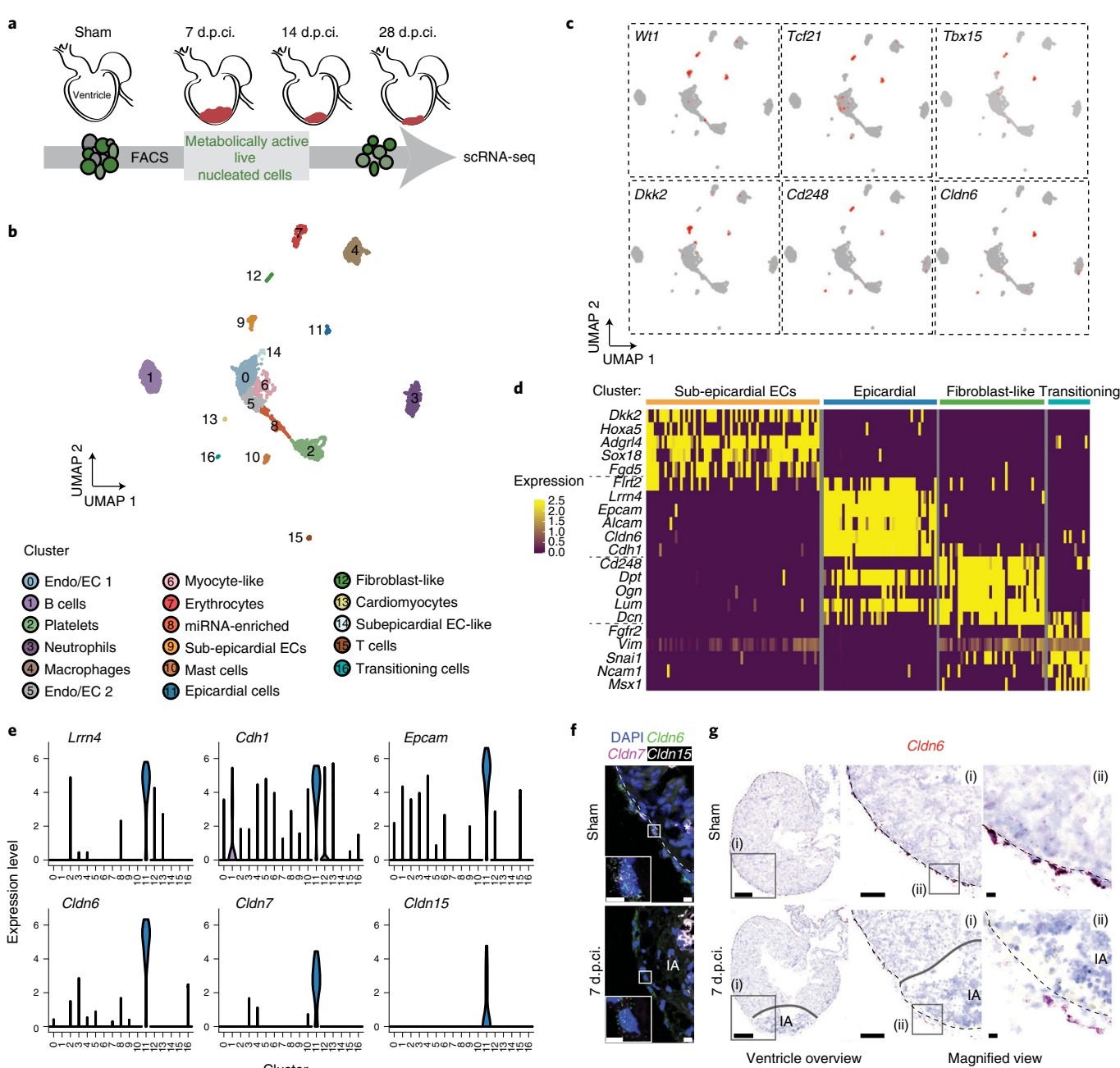

**Fig. 3 | *Cldn* family genes identified via scRNA-seq as specific markers of the salamander epicardium. a**, Schematic overview of the scRNA-seq experiments performed on metabolically active, live and nucleated cells of sham-operated and regenerating ventricles. **b**, Visualization of the scRNA-seq data from 2,386 individual cells using a uniform manifold approximation and projection (UMAP) plot. Endo/EC, endocardial/endothelial cell; and subepicardial ECs, subepicardial endothelial cells. **c**, UMAP plots showing the expression of the embryonic epicardial marker genes *Wt1*, *Tcf21* and *Tbx15* (top) as well as the marker genes identified in this study—that is, *Dkk2*, *Cd248* and *Cldn6* (bottom). **d**, Heatmap showing the expression levels of specific marker genes for the epicardium and EPDC cluster candidates and transitioning cells. See Extended Data Fig. 6b for the gene expression levels in all cell clusters. **e**, Distribution of expression for a subset of differentially expressed genes in cluster 11. **f**, *Cldn6*, *Cldn7* and *Cldn15* are coexpressed in epicardial cells. Representative images of fluorescent in situ hybridization of heart sections of sham-operated (top) and cryoinjured (bottom) animals at 7 d.p.ci. Note that the RNAscope fluorescent assay on fixed frozen salamander tissue causes background staining in blood cells (asterisks) that is easily distinguishable from real signal due to its non-dotty appearance. Insets: magnified views of the regions in the white boxes. Scale bars, 10 μm. **g**, Expression of *Cldn6* messenger RNA marks the epicardium. Representative ventricle overview (left) and magnified views (middle and right) of sham-operated (top) and 7-d.p.ci. (right) heart sections. The injury area (IA) is outlined and the boxes mark the areas selected for the magnified images. Scale bars, 500 μm (left), 200 μm (middle) and 10 μm (right). **f,g**, Dashed lines indicate the epicardium–myocardium border.

and reactome over-representation analyses highlighted genes associated with cell-junction organization and tight junction interactions as over-represented among the genes with the highest expression in this cluster (Extended Data Fig. 6a).

Cluster 12 cells expressed the stromal marker *Cd248* and showed enrichment for the expression of extracellular-matrix genes (*Dpt*, *Dcn*, *Ogn* and *Lum*; Fig. 3c,d and Extended Data Fig. 6b), suggesting that these cells are more mesenchymal in nature and

represent fibroblast-like cells. Gene Ontology term analysis showed enrichment of genes related to extracellular-matrix assembly and connective-tissue development (Extended Data Fig. 6a).

In situ hybridizations subsequently showed that *Dkk2*+ cells were localized to the subepicardial region (Extended Data Fig. 6d). *Cd248*+ cells were found in the subepicardium and myocardial interstitial space (Extended Data Fig. 6e). *Cldn6*, *Cldn7* and *Cldn15* expression overlapped and marked the outermost layer of the heart specifically (Fig. 3f,g). *Cldn6* was chosen as an epicardial cell marker for subsequent studies as it showed the highest level of expression among the tight junction genes (Fig. 3e). CHERRY+ epicardial cells marked after Cre-induced recombination in *GFP-loxP-Cherry* were *Cldn6*+ (Extended Data Fig. 6f), confirming that Cluster 11 cells represent the epicardium proper. Despite increased numbers of epicardial/subepicardial cells due to epicardial thickening, both the number of cells expressing *Cldn6* and the levels of *Cldn6* expression decreased at 7 d.p.ci. (Fig. 3f,g and Extended Data Fig. 6g–i). Accordingly, no cells were recovered from 7- and 14-d.p.ci. samples contributing to the *Cldn6*+ epicardial cell cluster, suggesting a dynamic response to injury, which we decided to explore further.

**CLDN6 localizes to cell clusters in the injury.** CLDN6 has a key role in the formation of embryonic epithelium and the development of endodermal tissues[54,55]. It is an oncofetal tight junction molecule that is expressed at high levels in stem cells and developing tissues but transcriptionally silenced in healthy adult tissues of mammals[56,57]. Immunohistochemical analyses in the post-metamorphic salamander heart showed that CLDN6 is expressed in the outermost epicardial layer and marks cell–cell junctions (Fig. 4a,b). Transmission electron microscopy (TEM) imaging confirmed that the outermost epicardial cells of the homeostatic epicardium are sealed via tight junctions at the epicardium–pericardial fluid interface (Fig. 4b). As expected, adherens junctions and desmosomes were located beneath the tight junctions (Fig. 4b). After cryoinjury (7 d.p.ci.), CLDN6 was present at reduced levels in the epicardial cells basal to the injury site and absent at the cell–cell borders (Fig. 4c,d). Providing further support for these data, TEM imaging revealed loss of tight junctions (Fig. 4d). Cells located subepicardially instead showed cytoplasmic CLDN6 expression, suggesting that EPDCs retain CLDN6 protein as they migrate (Fig. 4e). Subsequently, we found clusters of CLDN6+ cells in the injury forming honeycomb-like structures (Fig. 4f) connected via focal tight junctions rather than mature tight junction strands, suggesting a dynamic junctional remodelling accompanying migration of CLDN6+ EPDCs into the injury area (Fig. 4f). We did not detect expression of CLDN6 in the myocardium (Extended Data Fig. 7a).

**Targeting CLDN6+ epicardium and EPDCs.** We next wanted to ablate CLDN6+ cells to assess their relevance to regeneration. Certain members of the CLDN family proteins act as specific receptors of the *Clostridium perfringens* enterotoxin (CPE)[58–61]. Binding of CPE leads to the formation of an active pore, which subsequently enhances calcium influx and results in cell death[58]. Bulk and scRNA-seq data obtained from sham-operated and injured hearts showed expression of *Cldn6* and *Cldn7* (Extended Data Fig. 7b,c). Other CPE-sensitive *Cldn* receptors were not expressed (Extended Data Fig. 7b,c), allowing us to target CLDN6+ epicardium and EPDCs in the injured heart.

Although salamander CLDN6 is 68% identical to human CLDN6, with a high level of conservation of the key amino acids required for CPE binding (Extended Data Fig. 7d,e)[59], we first benchmarked the use of CPE in targeting salamander CLDN6+ cells. Transfection of HEK293T cells, which are normally non-responsive to CPE[62,63], with a *P. waltl Cldn6* expression construct sensitized the cells to CPE and resulted in cell death following treatment (Extended Data Fig. 7f). In contrast, transfection of a mutant form of the *P. waltl Cldn6* harbouring a deletion of the CPE binding domain did not sensitize the cells (Extended Data Fig. 7f), indicating that *P. waltl* CLDN6 is a specific receptor of CPE. To rule out off-target effects and distinguish cell type-specific effects of CPE from potential systemic toxicity, we generated previously well-characterized variants of the toxin: the binding-deficient CPE-Y306A/L315A and c-CPE that lacks the cytotoxicity domain (Fig. 5a)[64–66]. As expected, these variants did not show an effect on cell viability (Extended Data Fig. 7f). Next, we treated animals with wild-type CPE (wt-CPE), CPE-Y306A/L315A or c-CPE for 6 h at 7 d.p.ci. and performed a terminal deoxynucleotidyl transferase dUTP nick end labelling (TUNEL) assay to assess apoptosis across the injury area, epicardium, subepicardium and myocardium (Fig. 5b). Treatment with a low dose of wt-CPE (1.8 μg g⁻¹) resulted in a substantial increase in TUNEL+ cells, which were distinctly concentrated to the injury area, with little effect observed across the epicardium and subepicardium (Fig. 5c and Extended Data Fig. 8a,b). This suggested that CLDN6+ EPDCs in the injury area, rather than the epicardium itself, were preferentially targeted by wt-CPE at this dose. In comparison, treatment with a high dose of wt-CPE (10 μg g⁻¹) also caused cell death in the epicardium, confirming CLDN6 as a pan-epicardium marker (Extended Data Fig. 8c). Dying cells in the injury area were also confirmed to express CLDN6 (Extended Data Fig. 8d). No effect was observed across the myocardium, regardless of the dose injected, confirming the specificity of the toxin (Fig. 5c and Extended Data Fig. 8a–c). We observed a loss of CLDN6+ EPDCs in the injury area 24 h after treatment with wt-CPE (Extended Data Fig. 8e). In contrast to

**Fig. 4 | CLDN6+ cells organize in a honeycomb-like pattern in the injury area. a**, CLDN6 marks salamander epicardium. Overview images (top) and magnified views of the regions in the white boxes (bottom) of uninjured hearts stained for DAPI, CLDN6 and sarcomeric α-actinin (SαA). The arrowhead marks an epicardial cell. Scale bars, 50 μm (top) and 10 μm (bottom). **b**, Schematic depicting the region of interest (box) in an uninjured heart probed for CLDN6 expression and TEM imaging (top). Representative image showing DAPI, CLDN6 and pan-CK staining (middle); the arrows mark CLDN6+ cell–cell borders. Overview and close-up TEM images of homeostatic epicardium (bottom); desmosomes (arrow), adherens junctions (arrowhead) and tight junctions (asterisks) are shown. Scale bars, 20 μm (middle), 1 μm (bottom left) and 250 nm (bottom right). **c**, Epicardial CLDN6 expression is reduced following cryoinjury. Overview images (top) and magnified views of the region in the white box (i) (bottom) showing DAPI, CLDN6, SαA and pan-CK basal to the injury site at 7 d.p.ci. Scale bars, 50 μm (top) and 10 μm (bottom). **d**, Schematic depicting the region of interest in injured hearts probed for CLDN6 expression and analysed by TEM for tight junctions at 7 d.p.ci. (top). Representative images of CLDN6, MHC and pan-CK immunostaining (middle); the arrows mark the CLDN6⁻ cell–cell border. Overview (left) and magnified (right) TEM images of activated epicardium (bottom); desmosomes (arrows) and adherens junctions (arrowheads) are shown. Scale bars, 20 μm (middle), 1 μm (bottom left) and 250 nm (bottom right). **a,b,d**, The epicardium–myocardium border is marked by dashed lines. **e**, Subepicardial EPDCs express cytoplasmic CLDN6 7 d.p.ci. Representative images of DAPI, CLDN6, SαA and pan-CK immunostaining corresponding to the boxed area (ii) in panel **c**. The arrowheads mark CLDN6+ cells in the subepicardium. Scale bars, 10 μm. **f**, CLDN6+ cells form a honeycomb-like structure in the injury area at 7 d.p.ci. Representative images of CLDN6 and MHC immunostaining; low-magnification image showing the injured apex (top). High-magnification images of the outlined areas (i) and (ii) (middle). Overview (left) and magnified (right) TEM images showing focal tight junctions (marked by asterisks; bottom). Scale bars, 200 μm (top), 20 μm (middle), 1 μm (bottom left) and 250 nm (bottom right). **a–f**, The confocal images are representative of four animals and the TEM images are representative of three animals. IA, injury area; and peri, pericardial space.

wt-CPE, the binding-deficient negative control CPE-Y306A/L315A did not display toxicity, thereby confirming the CLDN dependence of the CPE effects (Fig. 5c). Similarly, c-CPE treatment also did not induce apoptosis (Fig. 5c).

To further confirm that CLDN6[+] cells are specifically ablated by wt-CPE, we combined lineage tracing of epicardial cells with wt-CPE treatment (Fig. 5d and Extended Data Fig. 8f). Under homeostatic conditions, wt-CPE treatment reduced the number

of CHERRY[+] epicardial cells by approximately 5.4-fold, which led to a decrease in the number of epicardium-derived cardiomyocytes of approximately fourfold (Extended Data Fig. 8g–j). The number of CHERRY[+] EPDCs in the injury area of cryoinjured hearts at 7 d.p.ci. was reduced approximately fivefold following wt-CPE treatment (Fig. 5e,f). Together, these experiments validate the use of wt-CPE to ablate CLDN6[+] cells for downstream functional experiments.

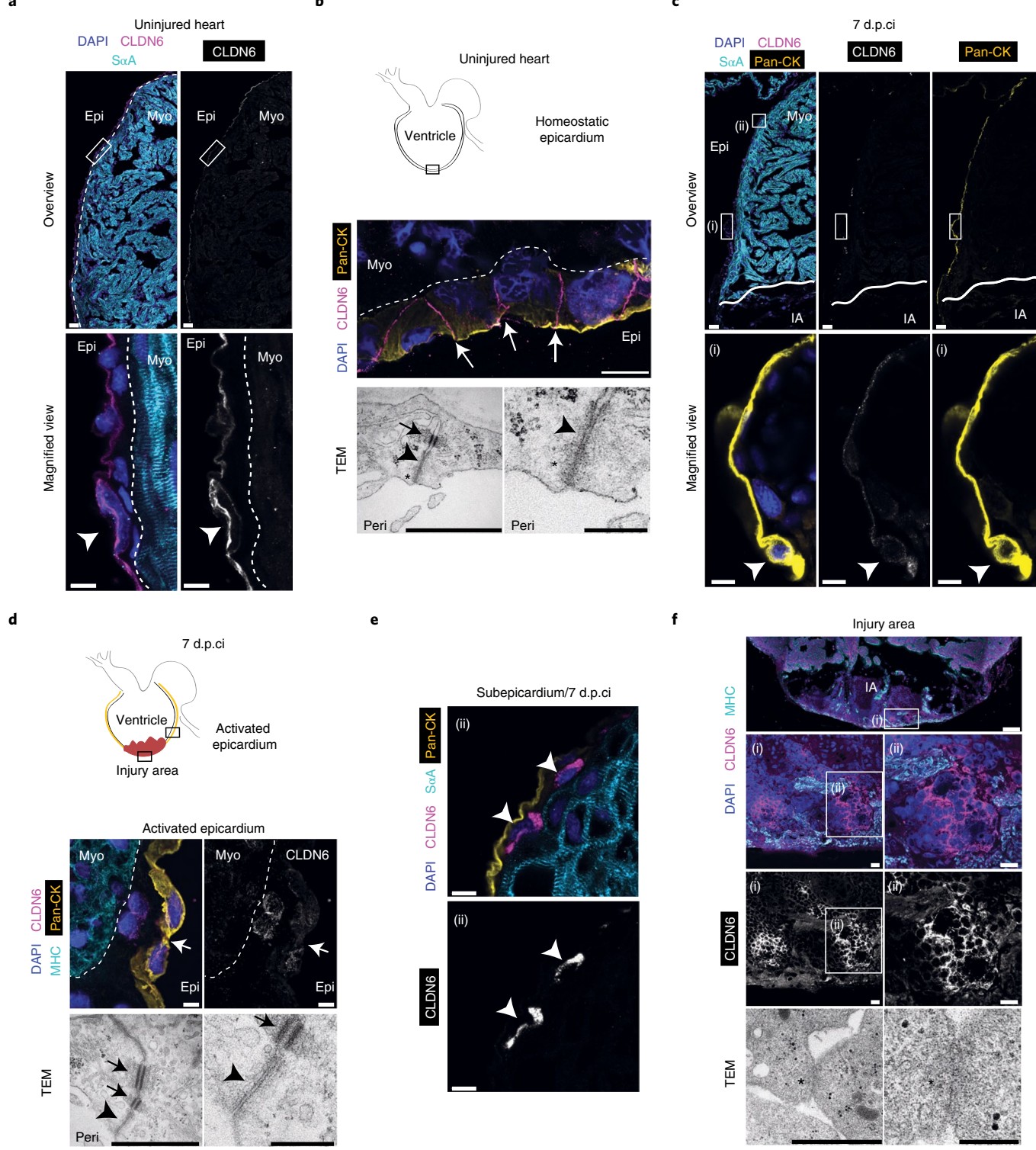

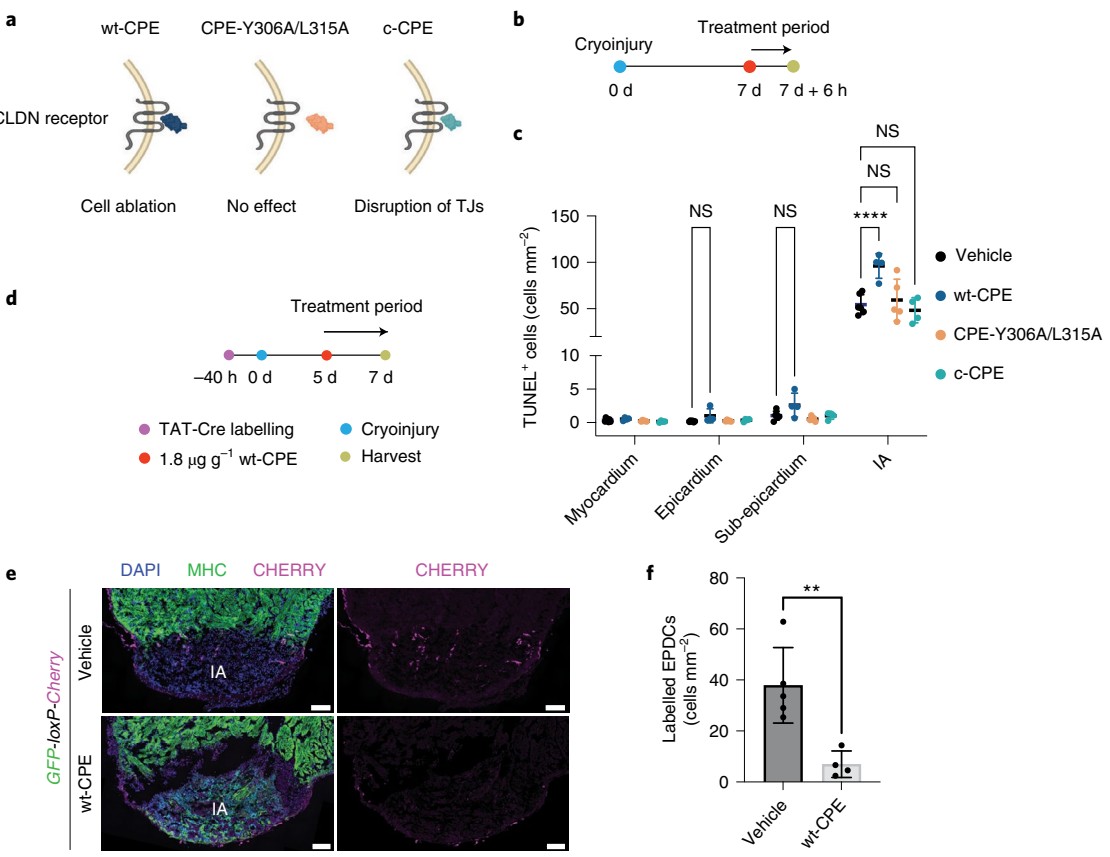

**Fig. 5 | CLDN6⁺ EPDCs are specifically ablated by wt-CPE. a**, Schematic showing the interaction between CPE variants and the CLDN receptor. TJs, tight junctions. **b**, Schematic of the experimental design for the TUNEL assay. **c**, Number of TUNEL⁺ cells following treatment with vehicle, wt-CPE, CPE-Y306A/L315A and c-CPE for 6 h. Data are the mean ± s.d. of $n = 7$ (vehicle), 4 (wt-CPE and c-CPE) and 5 (Y306A/315A) animals. Two-way ANOVA with Tukey's test; NS, not significant ($P > 0.05$) and ****$P < 0.0001$. **d**, Experimental design for assessing the effect of CPE treatment on lineage-traced EPDCs. **e**, CHERRY⁺ EPDCs are depleted by wt-CPE treatment. Representative images of DAPI, CHERRY and MHC immunostaining. Scale bars, 200 μm. **f**, Number of CHERRY⁺ EPDCs in the injury area at 7 d.p.ci. Data are the mean ± s.d. of $n = 4$ (vehicle) and 5 (wt-CPE) animals. Unpaired two-tailed Student's $t$-test; **$P = 0.0056$. IA, injury area.

**Ablation of CLDN6⁺ EPDCs impairs cardiac regeneration.** To ensure sustained depletion of CLDN6⁺ EPDCs, we established an optimized wt-CPE treatment regimen with minimal systemic toxicity to the animals (Fig. 6a and Extended Data Fig. 9a) and confirmed that the wt-CPE treatment did not impact the integrity of the myocardium in control hearts (Extended Data Fig. 9b). We next performed cryoinjuries and monitored the regeneration process in each animal up to 21 d.p.ci. using echocardiography (Fig. 6b, Supplementary Videos 1–4 and Extended Data Fig. 9c). Both vehicle- and CPE-Y306A/L315A-treated animals showed progressive regeneration over the course of 21 d, whereas the regeneration of wt-CPE-treated hearts was diminished (Fig. 6b,c). Histological assessment showed impaired regeneration of the wt-CPE-treated animals compared with the CPE-Y306A/L315A- and vehicle-treated animals at 21 d.p.ci. (Fig. 6d), highlighting the requirement of CLDN6⁺ EPDCs for efficient regeneration.

Next, we determined whether the wt-CPE treatment reduced the number of epicardium-derived cardiomyocytes. We injected TAT-Cre into *Cherry-loxP-H2B::YFP* reporter animals, treated them with either vehicle or wt-CPE and used echocardiography to measure the injury size and estimate the number of cardiomyocytes in the regenerate (Fig. 6e). We found that approximately 4% (percentage ± s.e.m., 4.27 ± 0.62; $n = 5$ animals) of the cardiomyocytes in the regenerate were derived from the epicardium in the vehicle-treated hearts at 21 d.p.ci. Given that the labelling efficiency of epicardial

cells was about 29%, we estimate that epicardium-derived cardiomyocytes account for approximately 15% of new cardiomyocytes at 21 d.p.ci. (Fig. 6f,g). In comparison, wt-CPE-treated hearts showed a substantial reduction in the number of epicardium-derived cardiomyocytes (percentage ± s.e.m., 1.45 ± 0.20; $n = 4$ animals), confirming the outcome of wt-CPE treatment (Fig. 6f,g). Together, these data show that the impaired regeneration, which at least in part is caused by the reduction of epicardium-derived cardiomyocytes, is not compensated by the contribution of new cardiomyocytes from elsewhere.

**Cardiogenic transition state revealed by scRNA-seq analyses.** To molecularly profile the conversion of epicardial cells to cardiomyocytes and determine the identity of the honeycomb-forming cells, we performed scRNA-seq on live cells isolated using fluorescence-activated cell sorting (FACS) from the injury area of vehicle- and wt-CPE-treated hearts at 7 d.p.ci. (Fig. 7a). To supplement the live-cell sort and enrich for EPDCs, we also sorted cells using an antibody to CLDN6 (Fig. 7a). Seven distinct cell clusters were identified through unbiased clustering and marker-gene expression (Fig. 7b, Extended Data Fig. 10a and Supplementary Table 1). By comparing clusters across each cell-isolation strategy, we identified Cluster 0 as an intermediate cell cluster that both disappeared following wt-CPE treatment and was enriched when cells were sorted with anti-CLDN6 (Fig. 7c). Notably, Cluster 0 showed

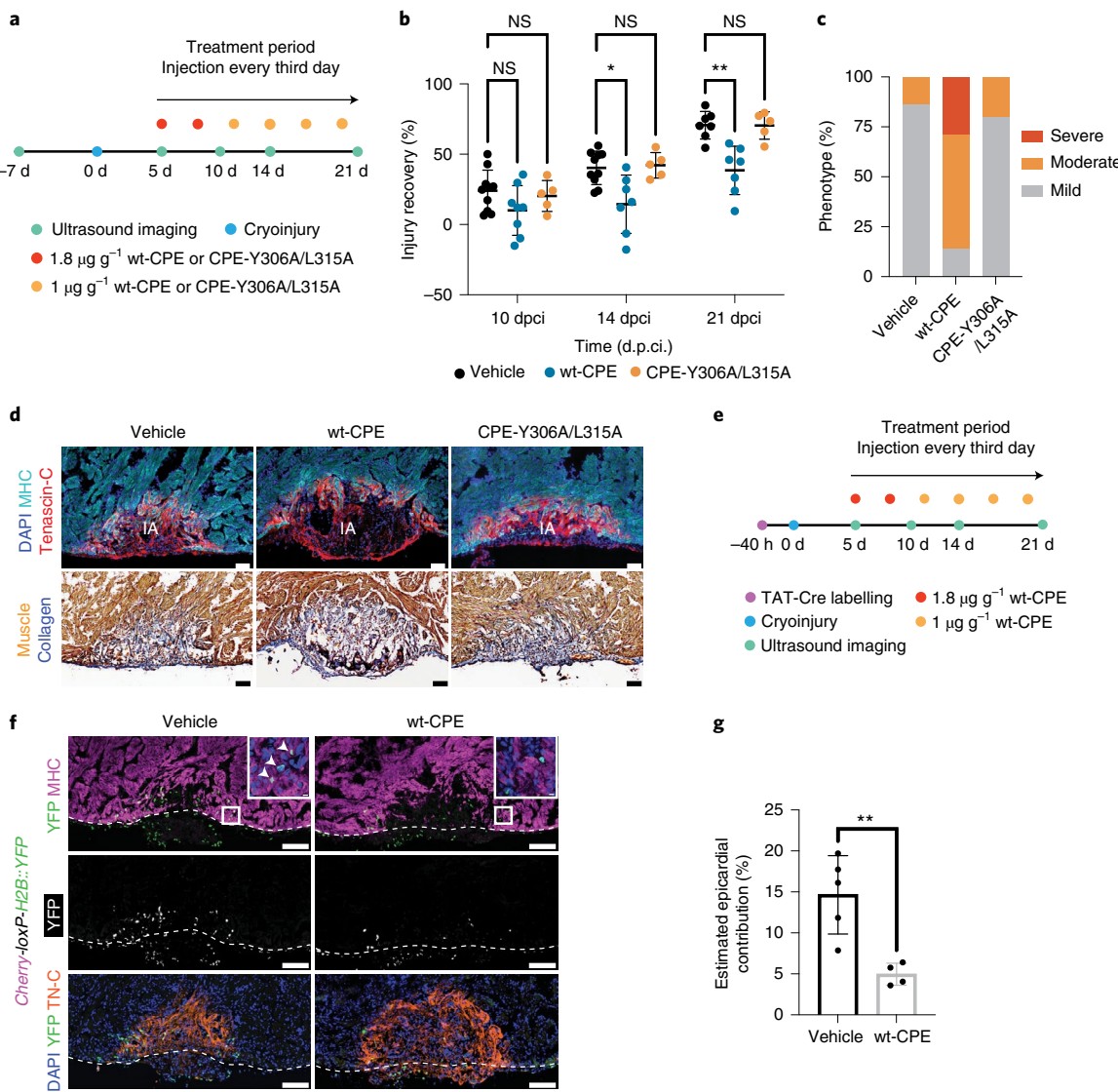

**Fig. 6 | Ablation of CLDN6⁺ EPDCs impairs cardiac regeneration. a**, Schematic of the treatment regimen to assess the impact of wt-CPE and CPE-Y306A/L315A on regeneration. **b**, Cardiac regeneration is impaired by wt-CPE treatment. Recovery from injury was monitored. Data are the mean ± s.d. of $n=10$ (vehicle), 8 (wt-CPE) and 5 (CPE-Y306A/L315A) animals. Two-way ANOVA with Tukey's test; 10-d.p.ci. vehicle versus wt-CPE, $P=0.52$; 14-d.p.ci. vehicle versus wt-CPE, *$P=0.0167$; 21-d.p.ci. vehicle versus wt-CPE, **$P=0.0034$; vehicle versus CPE-Y306A/L315A, $P>0.9999$; NS, not significant. **c**, Levels of regeneration at 21 d.p.ci. Score: mild, <45%; moderate, 45–75%; and severe, >75%, based on the remaining injury size. $\chi^2$ test; ****$P<0.0001$. **d**, Hearts treated with wt-CPE have larger injuries at 21 d.p.ci. compared with those treated with vehicle or CPE-Y306A/L315A. Representative images of DAPI, MHC and Tenascin-C immunostaining (top). AFOG staining of sections of vehicle-, wt-CPE- and CPE-Y306A/L315A-treated hearts (bottom). Scale bars, 100 μm. IA, injury area. **c**,**d**, $n=7$ (vehicle and wt-CPE) and 5 (CPE-Y306A/L315A) animals. **e**, Experimental design for assessing the impact of wt-CPE treatment on the epicardium-derived cardiomyocyte population. **f**, The epicardial contribution to cardiomyocytes is reduced by wt-CPE treatment. Representative images of DAPI, YFP, MHC and Tenascin-C immunostaining of sections of vehicle- and wt-CPE-treated hearts at 21 d.p.ci. The epicardium–myocardium border is marked by dashed lines. Insets: magnified views of the regions in the white boxes. The arrows show YFP⁺MHC⁺ cells in the regenerate. Scale bars, 200 μm (main images) and 10 μm (insets). **g**, Percentage of epicardium-derived cardiomyocytes in the regenerate at 21 d.p.ci. Unpaired two-tailed Student's $t$-test, **$P=0.006$. Data are the mean ± s.d. of $n=5$ (vehicle-treated) and 4 (wt-CPE-treated) animals.

expression of well-established cardiac transcription factors such as *Gata4*, *Gata6*, *Foxc1* and *Foxc2*, and expressed extracellular-matrix markers such as *Tenascin-X*, *Fibulin5* and *Collagen6* (Fig. 7d and Extended Data Fig. 10b). Co-immunostaining of CLDN6⁺ and isolectin-B4 excluded an endothelial cell identity for these cells (Extended Data Fig. 10c). Trajectory analysis of Cluster 0 suggested a differentiation path, where cells initially expressed genes related to EMT, such as *Twist1*, *Klf8* and *Fgfr2*, and subsequently upregulated the expression of cardiac muscle genes such as *Myl3*, *Myl4* and *Tnnc1* (Fig. 7e,f and Extended Data Fig. 10d). Based on the

expression of the early EMT marker *Snail1* (ref. [67]) and the absence of cardiomyocyte genes in the transitioning cells (Fig. 3d and Extended Data Table 1), we infer that they precede the intermediate cell state. These data suggest that injury induces transcriptional reprogramming of the epicardial cells into a cardiomyocyte fate via an intermediate CLDN6⁺ state. To further test this model, we combined lineage tracing of EPDCs in *GFP-loxP-Cherry* reporter animals with in situ hybridization against *Twist1*, *Gata4*, *Gata6* and *Myl3* at 7 d.p.ci. (Fig. 7g,h). We found that the *Cherry*⁺ EPDCs expressed *Twist1* and *Gata4* closer to the epicardium (Fig. 7g)

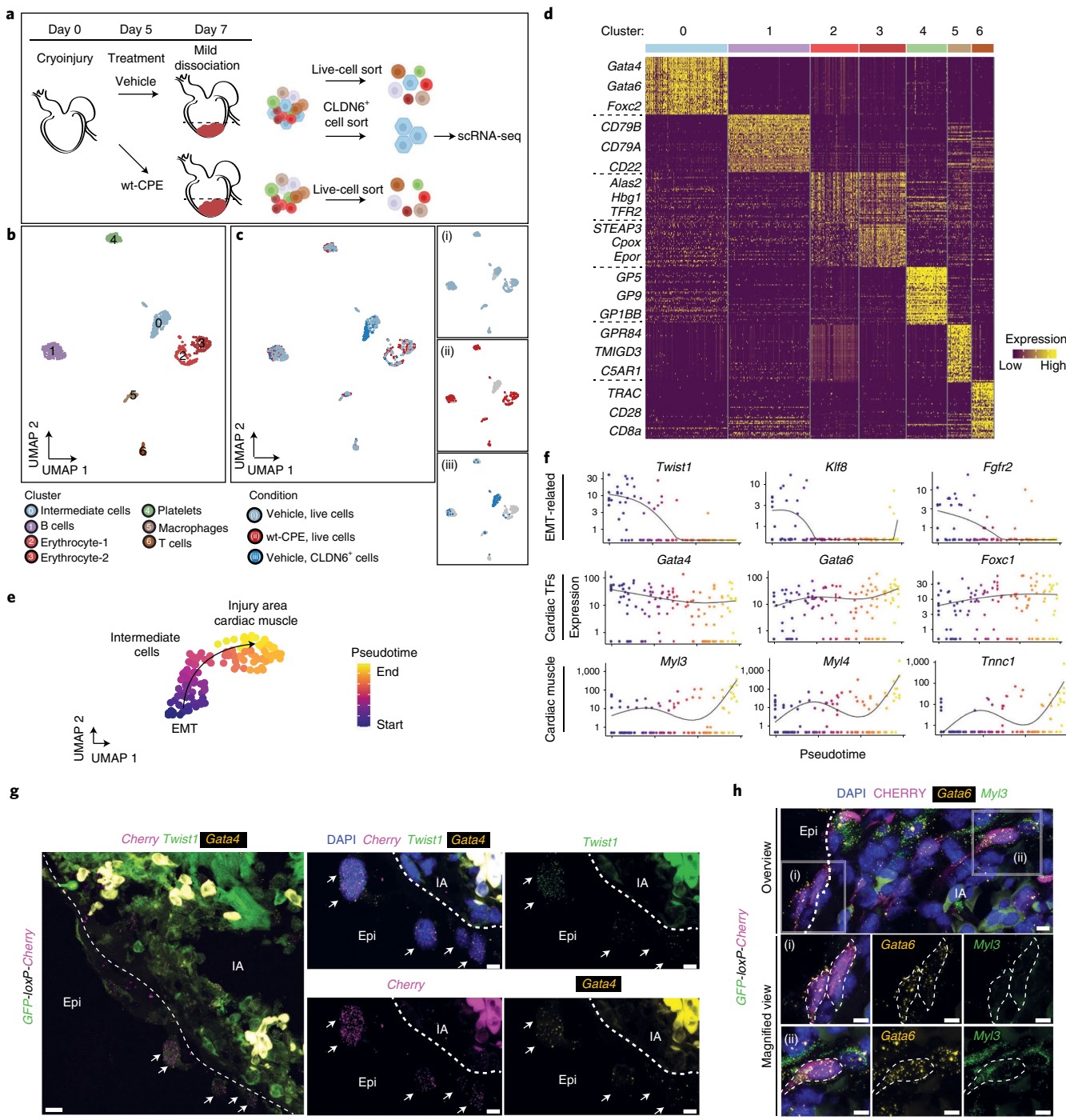

**Fig. 7 | scRNA-seq of CLDN6⁺ EPDCs reveal transcriptional reprogramming towards cardiomyocyte fate. a**, Experimental design for obtaining single cells from vehicle- and wt-CPE-treated hearts at 7 d.p.ci. The dashed lines show apex isolation. **b**, UMAP visualizing the scRNA-seq data from 681 single cells isolated at 7 d.p.ci. **c**, Cluster 0 represents CLDN6⁺ differentiation intermediates. UMAP showing the distribution of cells in relation to the treatment and FACS isolation strategy (left). UMAP showing live cells isolated from hearts treated with vehicle (i) or wt-CPE (ii) as well as CLDN6⁺ cells isolated from the vehicle-treated hearts (iii) (right). **d**, Heatmap showing specific marker-gene expression for the different cell clusters. **e**, Pseudotime trajectory of CLDN6⁺ differentiation intermediates. The arrow indicates the cardiomyocyte differentiation trajectory. **f**, Expression, and cell density plots of genes related to EMT (top), cardiogenesis (cardiac transcription factors, TFs; middle) and muscle cell function (bottom) across pseudotime. **g**, TAT-Cre recombinase-mediated lineage tracing in *GFP-loxP-Cherry* reporter salamanders combined with in situ hybridization shows that EPDCs express *Twist1* and *Gata4* as they migrate into the injury area at 7 d.p.ci. Overview image showing the injury area and the epicardial layer covering it (left). Magnified views (middle and right). Scale bars, 20 μm (overview; left) and 10 μm (magnified views; middle and right). The dashed lines separate the epicardium. The arrows mark cells expressing *Twist1* and *Gata4*. The bright autofluorescence in the injury area is unspecific staining of blood cells. **h**, TAT-Cre recombinase-mediated lineage tracing in tandem with in situ hybridization and immunostaining shows that EPDCs express *Myl3* after entering the injury area at 7 d.p.ci. Overview image showing a section of the injury area (top); the dashed line marks epicardium-IA border. Magnified views of (i) showing *Gata6⁺Myl3⁻* cells in the epicardial layer surrounding the injury area (middle). Magnified views of (ii) showing *Gata6⁺Myl3⁺* cells in the injury area (bottom). CHERRY+ cells are outlined by dashed lines. Scale bars, 10 μm. **g,h**, Data represent three animals. IA, injury area.

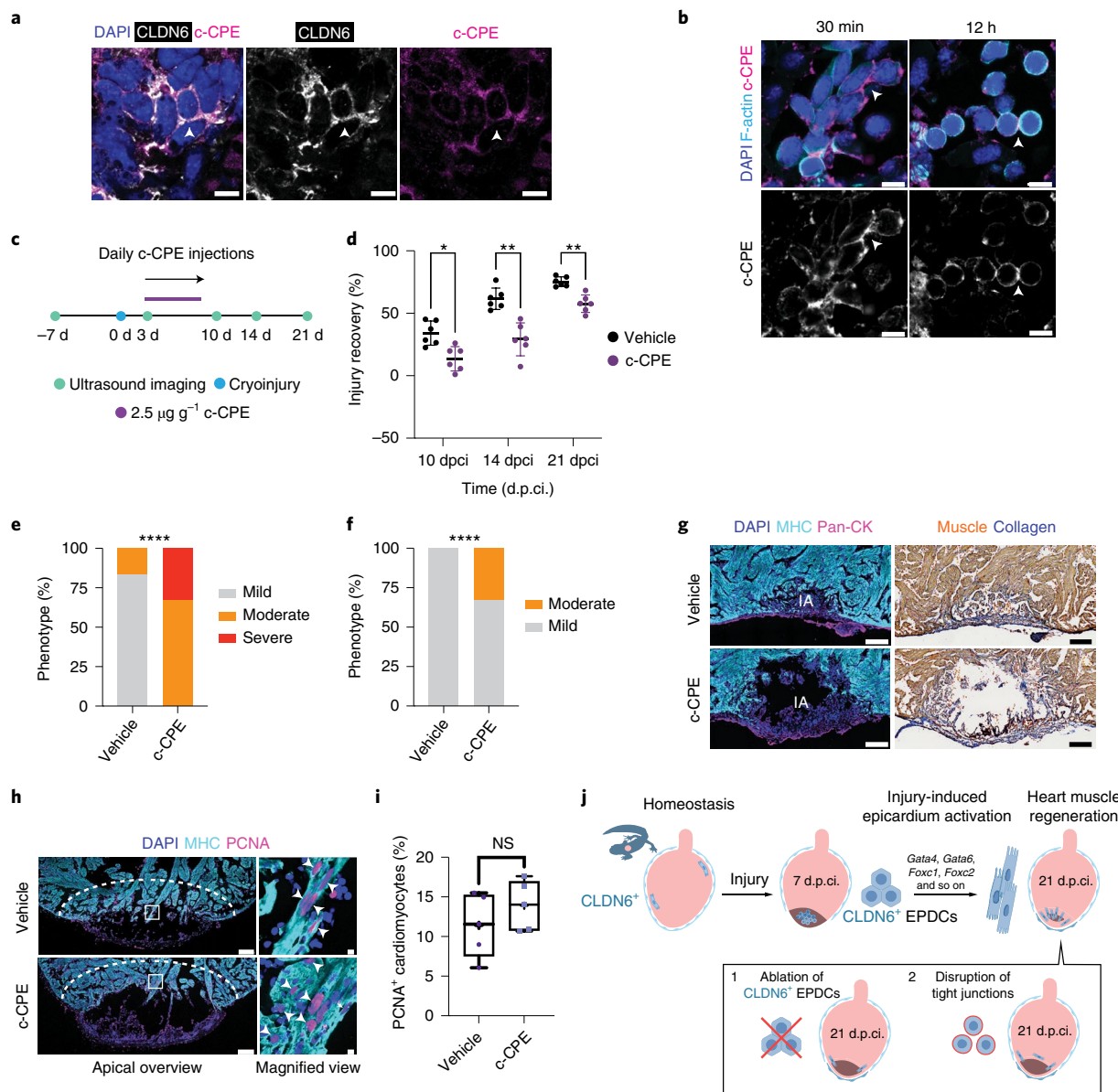

**Fig. 8 | Disruption of tight junctions inhibits regeneration. a**, c-CPE binds to CLDN6[+] cells in the injury area at 7 d.p.ci. Representative images of DAPI, CLDN6 and c-CPE immunostaining 30 min after c-CPE injection. The arrowheads mark a CLDN6[+]c-CPE[+] cell. Data represent three animals. **b**, Treatment with c-CPE disrupts the honeycomb-like structure. Representative images of DAPI, F-actin and c-CPE immunostaining at 7 d.p.ci. The arrowheads mark cell–cell contacts. Data represent three animals per time point. **a,b**, Scale bars, 10 μm. **c**, Schematic of the in vivo c-CPE treatment regimen. **d**, Recovery from injury was monitored. The injury area was calculated for each animal by echocardiography at the indicated time points and normalized to the pre-treatment injury size at 3 d.p.ci. Data are the mean ± s.d. Two-way ANOVA with Tukey's test; 10 d.p.ci., *$P = 0.0130$; 14 d.p.ci., **$P = 0.0024$; and 21 d.p.ci., **$P = 0.0022$. **e,f**, Levels of regeneration at 14 (**e**) and 21 d.p.ci. (**f**). Score: mild, <45%; moderate, 45–75%; and severe, >75%, based on the remaining injury size at 14 and 21 d.p.ci., respectively. $\chi^2$ test; ****$P < 0.0001$. **g**, The c-CPE-treated hearts have larger injuries at 21 d.p.ci. Representative images of DAPI, MHC and pan-CK immunostaining (left). AFOG staining (right). Scale bars, 200 μm. IA, injury area. **d–g**, $n = 6$ animals per treatment group. **h**, Border-zone cardiomyocyte proliferation is not hindered by c-CPE treatment. Representative images of DAPI, MHC and PCNA immunostaining at 10 d.p.ci. The dashed lines show the border-zone area. The boxed areas in the main images are magnified (right). The arrowheads mark PCNA[+] cardiomyocytes. Scale bars, 200 μm (overview; left) and 10 μm (magnified view; right). **i**, Percentage of PCNA[+] cardiomyocytes in the injury border zone from **h** at 10 d.p.ci. The box-and-whiskers plots show the mean (+), median (horizontal line), quartiles (boxes) and range (whiskers). Two-tailed Mann–Whitney test: NS, not significant ($P = 0.4206$); $n = 5$ animals per treatment group. **j**, Model showing the conversion of EPDCs into cardiomyocytes in response to injury.

and activated the expression of *Myl3* after entering the injury area (Fig. 7h). These observations corroborate the proposed differentiation trajectory inferred from the scRNA-seq analysis. Collectively, the results provide molecular evidence for injury-induced activation of epicardial cell-to-myocyte conversion.

**Disruption of tight junctions impedes cardiac regeneration.** To assess the importance of focal tight junctions connecting CLDN6[+] EPDCs, we took advantage of c-CPE, the non-toxic binding domain of CPE that disrupts tight junctions by temporarily displacing CLDNs (Fig. 5c)[68]. Fusion of recombinant c-CPE to Strep-Tag II

enabled visualization of c-CPE binding to CLDN6+ cells by immunostaining (Fig. 8a). Labelled cell clusters with clear cell geometry features were observed in the injury area of animals treated with c-CPE at 7 d.p.ci. for 30 min (Fig. 8a,b). This cellular architecture was lost by 12 h post injection, as evidenced by dispersion of the clusters and the labelled cells adopting a rounded shape (Fig. 8b). This indicates that c-CPE treatment is sufficient to disturb established cell contacts between EPDCs, thus affecting cell morphology and tissue organization.

To determine how the disruption of tight junctions and subsequent dispersion of CLDN6+ EPDC clusters may affect regeneration, cryoinjured salamanders were injected daily with c-CPE (3–10 d.p.ci.) and regeneration was monitored until 21 d.p.ci. by echocardiography (Fig. 8c and Supplementary Video 5). Hearts treated with c-CPE displayed impaired regeneration across the 21-d.p.ci. period, with the effect detectable as soon as 10 d.p.ci. Notably, the inhibitory effect persisted up to 21 d.p.ci., despite the treatment being terminated at 10 d.p.ci. (Fig. 8d–f), which was also confirmed at the histological level (Fig. 8g). These observations suggest a critical role for the focal tight junctions early in regeneration. Importantly, the c-CPE treatment had no effect on the regeneration of the epicardial cell layer, as assessed by pan-CK staining (Fig. 8g), and did not affect border-zone cardiomyocyte proliferation, as assessed by PCNA expression (Fig. 8h,i). Together, these data indicate that disruption of focal tight junctions between EPDCs is sufficient to delay cardiac regeneration without impinging on the epicardial integrity or cardiomyocyte proliferation.

## Discussion

Here we show that epicardium-derived intermediates migrate into the injured myocardium, create CLDN6+ honeycomb-like structures connected via focal tight junctions and become cardiomyocytes in salamanders (Fig. 8j).

The epicardium plays multiple essential roles in vertebrate heart regeneration. Epicardial cells secrete paracrine factors to neighbouring cells, including cardiomyocytes, that in the regenerative zebrafish re-enter the cell cycle and proliferate to replace lost cardiomyocytes[19]. Epicardial cells also produce extracellular-matrix components required for re-vascularization and muscle regeneration[69]. To what extent epicardial cells convert into other cell types during regeneration has remained an open question, mostly because of the lack of suitable lineage-tracing tools[25]. Although their contribution to fibroblasts, perivascular cells, smooth muscle cells and adipocytes is generally accepted, conversion to cardiomyocytes remains controversial[25]. Here we used a genetic marker-independent lineage-tracing strategy that demonstrates the conversion of epicardial cells to cardiomyocytes in salamanders. This occurs at a low rate during homeostasis and is accentuated in response to cryoinjury to regenerate cardiac muscle. Due to the relatively low labelling efficiency, it is not possible to fully quantify the contribution of epicardial cells to new cardiomyocytes but we estimate that at least 15% of the regenerated myocardium is derived from epicardium. This contribution is substantial, as indicated by observations that its prevention by ablation of the EPDCs impairs heart regeneration in salamanders. In addition to cardiomyocytes, we observed EPDCs coexpressing the mesenchymal marker vimentin or the smooth muscle cell/myofibroblast marker α-smooth muscle actin, indicating that epicardial cells also give rise to non-myocyte lineages. It will be important in the future to discern, both in quantitative and qualitative terms, the different roles the epicardium has in salamanders during homeostasis as well as after injury.

The specificity of TAT-Cre-mediated recombination is a key element of our study, for which we present two lines of evidence. First, nuclear Cre is detected only in epicardium. Small amounts of Cre protein diffusing into the myocardium are sequestered in the extracellular matrix, which precludes transduction of non-epicardial cells (Extended Data Figs. 1b and 2d)[37]. Second, we show that ablation of

lineage-labelled epicardial cells reduces the number of labelled cardiomyocytes during both homeostasis and regeneration (Fig. 6f,g and Extended Data Fig. 8e–j). It has been reported in other species that transient homotypic or heterotypic cell fusion could trigger cell-cycle re-entry of cardiomyocytes[70,71]. Theoretically, contribution from such a mechanism cannot be fully discounted in salamanders either. However, observations that regeneration is inhibited by the non-cytotoxic c-CPE in the absence of an effect on cardiomyocyte proliferation further supports the model that cellular contribution by the epicardium, rather than cell fusion between epicardial cells and cardiomyocytes, is essential for cardiac regeneration in salamanders (Fig. 8h,i).

The ablation studies using CPE rely on the expression of *Claudins* by epicardial cells and their progeny. The use of two different versions of the toxin, one ablating EPDCs and the other disrupting tight junctions, revealed that tight junctions per se are necessary for cardiac regeneration. These junctions are present both in the epicardial layer during homeostasis and in the EPDCs that invade the injury area. Tight junctions act as paracellular barriers for the passage of ions and solutes. They also function as mechanosensors bridging mechanical cues to the signalling platforms that regulate cytoskeletal organization and gene expression[72]. We observed that EPDCs form an intriguing honeycomb-like shape. Cells organized in a honeycomb-like pattern are widely found in natural contexts, ranging from retinal pigment epithelium[73] to endothelial cells during angiogenesis[74,75]. This is thought to give mechanical support to tissues[76]. Muscle cells in the heart are surrounded by collagen sheaths that are organized in a honeycomb-like network[77,78], which inspired the bioengineering of bilaminar scaffolds yielding electrically excitable grafts with multi-layered heart cells of neonatal rats[79]. It is not inconceivable that EPDCs migrating to the injury area provide support to regenerating cardiomyocytes regardless of their origin. In addition, the Hippo–YAP pathway has been shown to both translate mechanical forces into biochemical signals[80,81] and regulate differentiation of EPDCs in the developing mouse heart, potentially in response to mechanical tension[82]. Intriguingly, CLDN6 regulates the nuclear localization of YAP1 (refs. [83,84]) and renders lineage plasticity to hepatocellular carcinoma cells[84], indicating a link between fate switching and CLDN6.

Through lineage tracing, scRNA-seq and toxin-mediated ablation we reveal an epicardial cell-to-cardiomyocyte conversion trajectory, characterized by the expression of EMT markers, followed by signature genes expressed in cardiac muscle. Further studies will be needed for two principal goals. First, to identify putative epicardial subpopulations with cardiomyocyte potential. Second, to understand how to stimulate such a transition in species where this does not naturally occur, such as mammals. The present data indicate that epicardial plasticity serves as a basis for naturally occurring regeneration in a vertebrate, thereby highlighting the relevance of targeting the epicardium in mammals as a strategy complementary to stimulating cardiomyocyte proliferation.

## Online content

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

## Methods

**Animals.** All of the procedures in this study related to animal handling, care and treatment were performed according to the guidelines approved by the Jordbruksverket/Sweden under the ethical permit numbers 18190-18 and 5723-2019. *P. waltl* were bred and maintained in our aquatic animal facility[85]. The transgenic lines tgTol2(*CAG:loxP-GFP-loxP-Cherry*)[Simon] and tgTol2(*CAG:Nucbow*)[Simon] were described previously[38].

**Transgenesis.** Tol2-CAG:-loxP-mCherry-stop-loxP-h2bYFP plasmid and Tol2 transposase were injected into single-cell eggs to generate transgenic salamanders[86,87]. Only $F_1$ and $F_2$ CHERRY+ animals, in which the transgene is ubiquitously expressed, were used for experiments.

**Tissue processing and histological analysis.** Animals were anaesthetized in 0.1% MS222 (Sigma, A5040). Their hearts were excised, washed in 70% PBS with heparin (100 units ml$^{-1}$; Sigma, H4784) and fixed with 4% paraformaldehyde for 2 h at room temperature (RT) or overnight at 4 °C (Santa Cruz, CAS 30525-89-4) for immunostaining and rinsed three times before being exposed to 10, 20 and 30% sucrose (Sigma, S0389) solutions at 4 °C. The tissues were then equilibrated to O.C.T. compound (Tissue-Tek, 4583) by immersing them in a 1:1 mixture of 30% sucrose and O.C.T., followed by 100% O.C.T. and finally frozen. Sections (10–14 µm) were prepared on a cryostat at −14 °C (Thermo, Cryostar NX70) in the frontal plane and stored at −80 °C.

To obtain fresh frozen tissue sections, hearts were mounted in 7% tragacanth (Sigma, G1128), dipped in pre-cooled 2-metyhlbutane (Sigma, 277258) and frozen in liquid nitrogen.

**Staining procedures.** For immunofluorescence staining, tissue sections were fixed in 4% formaldehyde (Millipore, 1.00496.5000) for 5 min at RT and washed three times (5 min each) with PBS. The tissue was permeabilized with PBS containing 0.25% Triton X-100 (Sigma, T8787) for 10 min, followed by three washes of 5 min with PBS. The sections were blocked in 1% bovine serum albumin (Sigma, A7906), 10% normal goat serum (Invitrogen, 31873) and 0.1% Triton X-100 in PBS for 1 h at RT. The samples were then incubated overnight with primary antibodies in blocking solution at 4 °C. The following day, the sections were washed three times (10 min each) with PBS containing 0.1% Triton X-100 (PBS-T) and rinsed with PBS. The samples were incubated with secondary antibodies diluted in blocking solution for 2 h at RT. For nuclear staining, the samples were incubated with DAPI (2 µg ml$^{-1}$; Sigma, D9542) for 10 min, followed by five washes with PBS-T and one wash with PBS (5 min each). The sections were incubated with 0.1% Sudan black (Sigma, 199664) in 70% ethanol for 3 min and rinsed under water. The slides were mounted with DAKO fluorescent mounting medium (S3023, Agilent).

For the CLDN6 staining shown in Fig. 4b,d,f, fresh frozen tissue sections were fixed with pre-chilled methanol for 2 min and rehydrated in 75, 50 and 25% methanol in PBS (1 min each), followed by three washes in PBS (5 min each). The sections were blocked (1% bovine serum albumin and 10% normal goat serum in PBS) for 1 h at RT and incubated overnight with the primary antibodies at 4 °C. The following day, the slides were washed three times (10 min each) with PBS containing 0.1% Tween-20 and rinsed with PBS. Subsequently, the sections were incubated with the secondary antibodies for 2 h at RT, followed by incubation with DAPI for 10 min at RT. The slides were washed five times (5 min each) with PBS containing 0.1% Tween-20, rinsed with PBS and mounted. For the CLDN6 staining shown in Fig. 4a,c,e, the regular immunofluorescence protocol on fixed frozen tissue was followed.

For PCNA staining, the sections were incubated in citrate buffer (Sigma, C9999-100ML) at 90–95 °C for 15 min, allowed to cool at RT for 20 min and permeabilized in 0.2 % Triton X-100 for 15 min.

To detect proliferating cells, animals were injected with 0.1 mg g$^{-1}$ 5-ethynyl-2′-deoxyuridine (900584, Sigma), which was detected using a Click-IT EdU cell proliferation kit for imaging (C10340, Thermo Fisher).

For the AFOG staining, a dye solution was prepared by dissolving dye powder in distilled water to a final concentration of 0.5% aniline blue (Acros Organics, 401180250), 1% orange G (Sigma, O7252) and 1.5 % acid fuchsin (Sigma, F8129). The pH was set to 1.1 with hydrochloric acid (Sigma, H1758). Tissue sections were fixed in 10% neutral buffered formalin (Sigma, HT501128) for 10 min at RT and rinsed with PBS. The sections were incubated in Bouin's solution (Sigma, HT10132) for 2 h at 60 °C, followed by 1 h at RT. The slides were washed under water for 30 min and incubated for 5 min in 1% phosphomolybdic acid (Sigma, HT153), followed by a wash in distilled water for 5 min. Tissue was stained in AFOG solution for 8 min and rinsed in water for 5 min. The sections were rinsed twice in 95% ethanol and twice in 100% ethanol (5 min each). Finally, the slides were rinsed twice with xylene (Sigma, 214736) for 2 min, air dried and mounted in Organo/Limonene mounting medium (Sigma, O8015).

The following antibodies were used: guinea pig anti-CLDN6 (1:500; custom-made against the peptides TASQPRSDYPSKNYV and CPKKDDHYSARYTATA), rabbit anti-CLDN6 (1:50; Abcam, 107059), mouse anti-CLDN6 (1:30; Thermo Fisher, MA5-24076), mouse anti-MYH-1 (1:200; DSHB, MF-20), rabbit anti-cytokeratin (1:250; Abcam, 9377), rabbit anti-GFP (1:500; Life Technologies, A-6455), chicken anti-GFP (1:1,000; Abcam, ab13970),

mouse anti-Cre (1:500; US Biological, C7920), rabbit anti-Cre (1:500; Abcam, ab216262), rabbit anti-RFP (1:200; Rockland, 600-401-379), rat anti-RFP (1:200; Life Technologies, M11217), mouse anti-α-actinin (1:800; Sigma, A7811), rabbit anti-α-smooth muscle actin (1:100; Abcam, ab5694), isolectin GS-IB4 (1:200; Thermo Fisher, I21411 or I32450), chicken anti-vimentin (1:200; Millipore, AB5733) and Alexa Fluor 488 phalloidin (1:500; Thermo Fisher, A12379). Highly cross-adsorbed Alexa Fluor-conjugated secondary antibodies raised in goats were used at a 1:1,000 dilution. Specifically, anti-chicken 488 (A11039), anti-chicken 647 (A21449), anti-guinea pig 555 (A21435), anti-mouse IgG1 488 (A21121), anti-mouse IgG1 647 (A21240), anti-mouse IgG2b 647 (A21241), anti-mouse IgG2b 488 (A21141), anti-mouse IgG2b 568 (A21144), anti-mouse IgG2b 647 (A21242), anti-rabbit 488 (A11034), anti-rabbit 568 (A11011), anti-rabbit 647 (A21245) and anti-rat 568 (A11077). All secondary antibodies were obtained from Thermo Fisher.

**Nuclear Cre signal quantification.** High-magnification images were obtained using a Zeiss LSM900 confocal microscope. For each animal, at least 20 randomly selected areas (2,02 mm × 1,78 mm; scaled) capturing the epicardium–myocardium border were imaged. Macros were generated to quantify the mean intensity and raw integrated density of the nuclear Cre signal using ImageJ. Briefly, the pan-cytokeratin signal was used to define the epicardium. The DAPI signal was used to define cell nuclei. The nuclear Cre signal in a negative control area corresponding to the middle of the tissue was measured to determine the mean intensity threshold as 25 (average mean intensity + 3 × s.d.). False positives with a mean intensity of >25 and raw integrated density of <18,000 were excluded.

**Biotin permeability assay.** EZ-Link Sulfo-NHS-LC-Biotin (21335, Thermo Fisher) solution was prepared at a concentration of 1 mg ml$^{-1}$ in amphibian PBS (aPBS). Biotin (10 µl) was microinjected (FemtoJet 4i, Eppendorf) into the pericardial cavity of the animals and allowed to perfuse for 10 min. The biotin was quenched with 100 mM glycine in aPBS and hearts were fixed overnight. Sections were stained with the addition of Rhodamine Red-X-conjugated streptavidin (1:500 of 1 mg ml$^{-1}$ stock; Thermo Fisher, S6366) for 30 min at RT.

**Pericardial injections of the TAT-Cre recombinase.** TAT-Cre recombinase (70–100 µM; Millipore, SCR508) diluted in 10 µl amphibian HBSS (aHBSS) was microinjected (FemtoJet 4i, Eppendorf) into the pericardial sac of the reporter salamanders. The animals were kept in water at 25 °C. For labelling combined with the injury, the animals were subjected to cryoinjury at 40 h.p.i. After the injury the animals were kept in water at 18 °C and the temperature was raised to 25 °C at 10 d.p.ci.

**Cryoinjury.** Post-metamorphic salamanders (7–10 cm) were anesthetized by immersion in 0.1% MS222 (Sigma) and placed in a supine position. A skin incision was made and the pericardium was cut open. The heart was manoeuvred out of the pericardial cavity. Excess moisture was removed with a tissue. A liquid nitrogen-cooled copper filament (1.2 mm) was applied to the ventricle apex for 10 s. The heart was then placed back into the pericardial cavity. The pericardium was sealed with ethilon 8-0 black monofilament sutures (Ethicon) and the outer skin was sealed with Vicryl Rapid 6-0 (Ethicon). The animals were placed in 0.5% sulfamerazine solution (Sigma, S0800) on ice overnight and transferred to regular water the next day. The water temperature was raised to 25 °C at 10 d.p.ci.

**Heart dissociation and FACS.** We adopted a dissociation protocol biased towards non-myocyte cell populations of the heart[88]. Hearts were rinsed with ice-cold 70% aHBSS (Sigma, 55037C). The ventricles were minced into smaller pieces and collected in cold aHBSS. The tissue pieces were allowed to settle, rinsed once with aHBSS and incubated with 2 mg ml$^{-1}$ Collagenase/Dispase (Sigma, 10269638001) in aHBSS for 2 h at 27 °C. The tissue pieces were rinsed with aHBSS, resuspended in aPBS with 10% fetal bovine serum (FBS) and mechanically broken with the help of a pipette. Cells were passed through a 100-µm filter and spun down at 300$g$ for 5 min at 4 °C. The pellets were resuspended in 1 ml aPBS with 1% FBS. The cells were stained with Sytox Blue dead-cell stain (1:1,000; Thermo Fisher, S34857), calcein AM (1:250; Thermo Fisher, C1430) and Vybrant DyeCycle ruby (1:1,000; Thermo Fisher, V10273) and then sorted on a FACS Aria III system (BD Biosciences) using a 130-µm nozzle.

To establish a milder dissociation protocol to capture the cells in the injury area at 7 d.p.ci., the apical regions of the hearts were removed with a scalpel and minced into smaller pieces that were collected in aHBSS. The tissue pieces were rinsed once with cold aHBSS and incubated in digestion buffer containing 1.5 mg ml$^{-1}$ bovine serum albumin (Sigma, A7906), 3 mg ml$^{-1}$ glucose (Sigma, G6152) and 2 mg ml$^{-1}$ Collagenase/Dispase (Sigma, 10269638001) in aHBSS. The tubes were placed in a 27 °C waterbath for 30 min with shaking (90 r.p.m.). After 30 min, solution containing tissue pieces and dissociated cells was gently pipetted up and down without disturbing the larger tissue pieces and collected in a separate tube with FBS. Fresh digestion buffer was added to the remaining tissue pieces and the procedure was repeated.

To perform CLDN6 staining on isolated cells, 5 µg anti-CLDN6 (Abcam, 107059) or rabbit isotype control (Abcam, ab171870) was conjugated to Dylight

650 (Abcam, 201803). The cells were incubated with the conjugated antibodies for 1 h at 4 °C and washed twice with FACS buffer to remove the unbound antibodies. Sytox blue and vybrant orange staining was performed. The FACS data were analysed using the FACSDiva and FlowJo software.

**scRNA-seq.** The single-cell transcriptome data were generated at the Eukaryotic Single-cell Genomics facility of the Science for Life Laboratory in Stockholm, Sweden.

Single-cell libraries were sequenced on a HiSeq2500 system by Illumina. Single-end 43-base-pair reads were mapped to the *P. waltl* reference coding sequences consolidated in orthology groups using STAR (v.2.5.3a)[89]. Reads mapping uniquely to one orthology group were assigned to that group and a final matrix of unique read counts per orthology group was used for downstream analysis.

Analysis of the scRNA-seq was performed using Seurat package version 4.0.1 (ref. [90]). Genes expressed in fewer than three cells and cells with less than 200 expressed genes were omitted. Cells with total read counts lower than 5,000 were also discarded. After quality control, data from 2,908 cells were analysed. The average counts of individual cells were 155,888. Total read counts and cell-cycle regression were performed and normalized with the CellCycleScoring() and ScaleData() functions in Seurat. The FindVariableGene() function was used to identify 2,000 variable genes, which were subjected to subsequent analysis. Principal component analysis was performed, and the first 12 (Fig. 3) and 24 (Fig. 7) principal components were used for UMAP dimension reduction analysis. The UMAP was used for graphic-based clustering using the FindNeighbors and FindClusters functions.

Top markers of individual clusters were identified using the FindAllMarkers function. The top 300 marker genes with the highest log-transformed fold change and $P < 0.05$ were subjected to over-representation analysis using the protein analysis through evolutionary relationships (PANTHER) tool[91]. The analysis was conducted with the PANTHER GO-slim Molecular Function, PANTHER GO-slim Biological Process and Reactome pathway databases, using an over-representation test by Fisher's exact test with a false discovery rate cutoff of 5%. Features in the scRNA-seq data were used as the background.

Monocle 3 was used for single-cell trajectories and pseudotime analysis based on the UMAP generated in the Seurat analysis[92–95]. Genes that were differentially expressed across the pseudotime axis were identified using the graph_test() function. The corresponding heatmaps were plotted using the plot_pseudotime_heatmap function in Monocle 2.

Scatter, bar, dot and violin plots, and other data representation graphs were generated using the ggplot2 R package.

**RNAscope in situ hybridization.** Custom RNAscope probes against *Cldn6*, *Cldn7*, *Cldn15*, *Dkk2*, *Cd248*, *Twist1*, *Gata4*, *Gata6* and *Myl3* were designed and manufactured by Advanced Cell Diagnostics (ACD). Catalogue probe against *Cherry* was used to detect the CHERRY signal (ACD, 431201-C2 and 431201-C3). RNAscope fluorescent multiplex assay (ACD, 320850), RNAscope 2.5 HD duplex assay (ACD, 322430) and RNAscope 2.5 HD assay-RED (ACD, 322350) were performed according to the manufacturer's recommendations, with minor modifications: fixed frozen tissues were treated with protease III for 15 min. High-magnification images were acquired with a Zeiss LSM700 confocal microscope or Zeiss AxioScan Z1. The Zen Blue, CaseViewer and HALO software were used for visualization and quantification.

**TEM.** Transmission electron microscopy was performed at the TEM unit (EMil) of the Karolinska Institute. Salamander hearts were fixed with 2.5% glutaraldehyde in 0.1 M phosphate buffer, pH 7.4, and stored at 4 °C. Following the primary fixation, the hearts were rinsed with 0.1 M phosphate buffer and post-fixed in 2% osmium tetroxide in 0.1 M phosphate buffer, pH 7.4, at 4 °C for 2 h. The hearts were then stepwise ethanol dehydrated, followed by acetone and embedded in LX-112. Semi- and ultrathin sections were prepared using a Leica EM UC7 ultramicrotome. The ultrathin sections were contrasted with uranyl acetate, followed by lead citrate and examined in a Tecnai 12 Spirit Bio TWIN transmission electron microscope operated at 100 kV. Digital images were acquired using a 2kx2k Veleta CCD camera.

**Phylogenetic tree and sequence alignment.** Multiple sequence alignment of CLDN6 was performed using the T-Coffee programme with default settings using the EMBL-EBI[96,97]. A phylogenic tree was generated using the ClusterW2 package in the EMBL-EBI API[98]. The alignment and tree were visualized using Jalview and TreeDyn[99–101].

**Bulk RNA sequencing.** Sequencing libraries were prepared using the TruSeq stranded total RNA sample prep LS protocol. Reads were mapped and annotated as described earlier. Trimmed mean of M values normalization was performed using the edgeR package[102].

**Molecular cloning.** The *P. waltl Cldn6* sequence was retrieved from the *P. waltl* genome. Amino acids at positions 140–157 were removed to create the *PwCldn6Δ* sequence, where the ECL2 domain was removed. The T2A-H2B-EBFP2 sequence was inserted to the 5′ end of the stop codon in wild-type *P. waltl Cldn6* and the

*PwCldn6Δ* sequence, as a selection marker. The recombinant sequences were synthesized as double-stranded DNA fragments (IDT gblock) and inserted into a *piggyBac*-CAG expression plasmid using an infusion kit (Takara). Plasmids were transformed into One Shot Stbl3 chemically competent *Escherichia coli* (Thermo Fisher, C737303).

**Cell culture and transfection.** The HEK293T cell line was purchased from the American Type Culture Collection (CRL-3216). The cells were maintained in Dulbecco's modified eagle medium with 10% FBS (Life Technologies). The cells were transfected using Lipofectamine 2000 (Invitrogen, 11668-027).

**Cell viability assay.** HEK293T cells (10,000) were plated into each well of a 96-well plate coated with poly-L-lysine (Sigma, P5899). The cells were exposed to wt-CPE, Y306A/L315A or c-CPE diluted in culture medium (3 mg ml⁻¹) the following day. A Pierce LDH cytotoxicity assay kit was used to determine the levels of cytotoxicity (Thermo Scientific, 88953).

**Expression and purification of NH₂-His-tagged CPE.** The pET16b-10×HIS-CPE plasmid was generated by GenScript[103,104]. An overnight culture (10 ml) of BL21-Codon Plus (DE3)-RIL competent cells (Agilent Technologies, 230245) transformed with 60 ng pET16b-10×HIS-CPE was inoculated into 1 l of LB medium containing 100 mg ml⁻¹ carbenicillin and 25 mg ml⁻¹ chloramphenicol. The culture was cultured at 37 °C with vigorous shaking to an optical density at 600 nm of 0.5–0.6 and induced with 1 mM isopropyl-1-thio-β-D-galactopyranoside (Thermo Fisher, R1171) for 3 h. The cells were harvested by centrifugation at 4,000*g* for 20 min at 4 °C. The cell pellets were resuspended in native lysis buffer containing 50 mM NaH₂PO₄, 0.5 M NaCl, 1 mM imidazole, rLysozyme (60 KU g⁻¹ cell paste; Millipore, 71110), benzonase nuclease (250 units ml⁻¹; Sigma, E1014-25KU) and EDTA-free protease inhibitor (Thermo Fisher, A32965), pH 8.0, and kept on ice for 30 min, followed by sonication with Bandelin Sonopuls (Cycle 2, 6 min, 50% power output). The suspension was centrifuged at 26,500*g* for 30 min at 4 °C. The supernatant was applied to Ni-NTA agarose affinity resin for 1 h at 4 °C. Unbound proteins were washed away with a buffer containing 20 mM imidazole and the His-tagged wt-CPE was eluted with 250 mM imidazole. Elutions containing wt-CPE were pooled and concentrated using a Pierce protein concentrator (10 kDa molecular-weight cutoff; Pierce, 88517). Triton X-114-mediated endotoxin removal was performed[105] and high-affinity Triton-binding beads (Bio-Rad, 152-8920) were used to remove the residual TX-114. Zeba spin desalting columns (Thermo Fisher, 89890) were used to remove the residual salts. Purified wt-CPE protein was kept at −80 °C.

**Expression and purification of Strep-Tag II-tagged Y306A/L315A, CPE and c-CPE.** BL21-Codon Plus (DE3)-RIL competent cells (Agilent Technologies, 230245) were transformed with 50 ng Strep-Tag II–CPE_pET22b(+), Strep-Tag II–cCPE_pET22b(+) or Strep-Tag II–Y306A/L315A_pET22b(+), generated by GenScript. An overnight culture (10 ml) was used to inoculate 1 l of LB medium containing 100 mg ml⁻¹ carbenicillin and 25 mg ml⁻¹ chloramphenicol. When the absorbance at 600 nm reached 0.5–0.7 after incubation at 37 °C with vigorous shaking, the culture was induced with 1 mM isopropyl-1-thio-β-D-galactopyranoside (Thermo Fisher, R1171). After 3 h, the cells were harvested by centrifugation at 4,000*g* for 20 min at 4 °C. The pellets were resuspended in lysis buffer containing 100 mM Tris–HCl, 150 mM NaCl, 1 mM EDTA (iba, 2-1003-100), rLysozyme (60 KU g⁻¹ cell paste; Millipore, 71110), benzonase nuclease (250 units ml⁻¹; Sigma, E1014-25KU) and EDTA-free protease inhibitor (Thermo Fisher, A32955), pH 8.0, and kept on ice for 30 min, followed by sonication with Bandelin Sonopuls (Cycle 2, 6 min, 50% power output). After centrifugation of the lysed cells at 26,500*g* for 30 min at 4 °C, the supernatant was passed through a Strep-Tactin Sepharose column (iba, 2-1202-051). Unbound proteins were washed away with 100 mM Tris–HCl, 150 mM NaCl and 1 mM EDTA, pH 8.0 (IBA, 2-1003-100), and the Strep-Tag II-tagged protein was eluted with 2.5 mM desthiobiotin (IBA, 2-1000-025). The eluate was concentrated using a Pierce protein concentrator (10 kDa molecular-weight cutoff; Pierce, 88517) and endotoxins were removed by incubation with Triton X-114 (Sigma, X114-1L). Triton X-114 residuals were removed using high-affinity Triton-binding beads (Bio-Rad, 152-8920) and salt removal was performed with Zeba spin desalting columns (Thermo Fisher, 89890). Purified Strep-Tag II-tagged CPE, Y306A/L315A and c-CPE protein were kept at −80 °C.

**Treatment of salamanders with wt-CPE and its variants.** For experiments where epicardium was ablated following TAT-Cre labelling (Extended Data Fig. 8f–j), 200 ng Strep-Tag II-tagged wt-CPE in a volume of 5 µl was microinjected into the pericardial cavity of the animals.

**Echocardiography.** Serial echocardiography was performed using a Vevo 3100 imaging system (VisualSonics) equipped with a high-frequency transducer (MX700, 29-71 MHz). Animals were anesthetized with pH-adjusted MS222 (0.05%; pH 7.0–7.5; tricaine methanesulfonate, Sigma) and placed in the supine position (VisualSonics). B-mode parasternal long axis (PLAX) images were acquired by placing the transducer directly above the ventral side of the animal, parallel to the

midline of the chest. For each animal and time point, we first obtained B-mode images corresponding to the PLAX plane containing both the ventricle and outflow tract. Within the context of regeneration, additional planes containing the largest cross-sectional area of the injury were obtained for injury-size calculations when necessary. This was achieved by scanning the transducer along the left–right axis using the in-built micromanipulator. The focus depth, two-dimensional gain and image dimensions were optimized according to the manufacturer's recommendations and at least four cardiac cycles were recorded per imaging sequence. After the procedure, the animals were returned to water to resuscitate. Injury size was calculated by measuring the area of the whole ventricle (excluding outflow tract but including injury area) and the injury itself in Vevo Lab 3.2.0 using the 2D area function. The following formula was used: (injury size ÷ ventricle size) × 100.

**Quantification.** In each type of experiment, hearts from different animals were analysed in multiple independent experiments performed on different days. Six series with frontal cryosections, each containing representative regions of the heart were generated. At minimum, one of these series was immunostained and all sections were documented via high-throughput imaging to ensure fair sampling for quantification. Subsequently, at least three representative sections were selected for further quantification.

Quantification of epicardium-derived cardiomyocytes in Fig. 1i was performed by counting the number of recombined cells that were MHC⁺ and normalizing to the area of the section.

Quantification of nuclear Cre signal in Extended Data Figs. 1b and 3c was performed using ImageJ as described above; 698, 308, 800 epicardial cell nuclei, and 1,569, 1,719 and 3,846 myocardial cell nuclei were analysed at 30, 40 and 96 h.p.i., respectively (Extended Data Fig. 1b). Cell nuclei (3,105) in the injury area and border myocardium were analysed at 5 h.p.ci. (Extended Data Fig. 3c).

In Fig. 2g, the number of epicardium-derived cardiomyocytes was normalized to the injury area on day 3 as a proxy to the starting injury size. Serial echocardiography was used to measure the injury sizes at 3 and 21 d.p.ci. for each animal. This led to the calculation of a regeneration ratio by dividing the injury size at 3 d.p.ci. by the injury size at 21 d.p.ci. Tenascin-C staining on cryosections obtained on day 21 closely matched the injury sizes calculated via echocardiography at this time point. Therefore, measuring the Tenascin-C⁺ area and multiplying it by the calculated regeneration ratio allowed us to estimate the starting injury size for each animal. For the sham-operated hearts, the area corresponding to 25% of the ventricle size was measured.

To analyse the clone size for the experiments in Extended Data Fig. 4, an overview image of the apex was taken, followed by a z-stack image of the clone. Cells were considered clonally related only if they were nearby and showed identical colours.

Quantification of RNAscope data in Extended Data Fig. 6g–i was performed using the HALO software employing custom parameters for cell detection. Analysis was extended to the epicardium–subepicardium, defined as the region surrounding the myocardium.

Quantification of TUNEL⁺DAPI⁺ nuclei in Fig. 5c and Extended Data Fig. 8c was stratified to specific regions of the heart. Epicardium was defined as the pan-CK⁺ layer, whereas subepicardium was defined as the area between the pan-CK⁺ layer and myocardium. All counts were normalized to the area of each respective compartment. In the case of epicardium and subepicardium, the counts were normalized to the area of the myocardium as a proxy.

Quantification of labelled EPDCs per mm² in Fig. 5f was done by counting the number of CHERRY⁺ cells in the injury area and normalizing the count to the size of the injury. Labelled cells on the epicardial layer covering the injury area were not included.

Quantification of the percentage of epicardium-derived cardiomyocytes in the regenerate in Fig. 6g was done by calculating the starting injury sizes as described in the previous section for Fig. 2f and counting the total number of cardiomyocytes in this area.

Quantification of PCNA⁺ cardiomyocytes within the border-zone myocardium in Fig. 8i was calculated by first counting the number of MHC⁺DAPI⁺ nuclei within a distance of 250 µm from the injury border and then counting the number of PCNA⁺ cells in this population. All counts were normalized to the size of the counting area to account for variations in animal-heart size.

To calculate the injury-recovery percentages in Figs. 6b and 8d, the injury sizes were measured based on serial echocardiography images and normalized to the pre-treatment injury size at 5 and 3 d.p.ci., respectively, for each animal.

For experiments involving CPE-variant treatments, salamanders were pre-screened for normal heart function via echocardiography. The injury sizes at 5 (wt-CPE-related experiments) or 3 d.p.ci. (c-CPE-related experiments) were measured to ensure consistent tissue damage and animals were allocated randomly into treatment groups. Sample sizes were not pre-determined based on statistical power calculations but were based on our experience with this type of in vivo experiments. For assays in which variability is high, we typically used $n \geq 8$ salamanders and repeated the assay multiple times to ensure reproducibility.

Quantification of the percentage of vimentin⁺ or MHC⁺ EPDCs in Extended Data Fig. 3l was done by counting the number of YFP⁺ EPDCs in the regenerating

area that coexpressed the respective markers. Five sections were randomly selected for each animal to perform the quantification.

Quantification of labelled epicardial cells in Extended Data Fig. 8i was performed by counting the number of CHERRY⁺pan-CK⁺ cells in the epicardial layer and normalizing the count to the area of the ventricle.

Quantification of labelled myocytes in Extended Data Fig. 8j was performed by counting the number of CHERRY⁺MHC⁺ cells in the myocardial layer and normalizing the count to the area of the ventricle. Ten sections were randomly selected for each animal to perform the quantification.

**Statistics and reproducibility.** No statistical methods were used to pre-determine the sample sizes. The sample sizes were empirically estimated on the basis of pilot experiments and previously performed experiments with a similar set-up to provide sufficient sample sizes for statistical analyses. Experiments were repeated multiple times. The number of biological and technical replicates are reported in the legend of each figure. Statistical analyses were performed using GraphPad Prism 9. Two-parameter comparisons of samples from in vivo studies were performed using a two-tailed unpaired Student's $t$-test. Analyses of in vivo studies with multiple parameters were performed using a two-way ANOVA with Tukey's post-hoc test. Statistical significance was defined as $*P < 0.05$, $**P < 0.01$, $***P < 0.001$ and $****P < 0.0001$ for all figures.

**Reporting Summary.** Further information on research design is available in the Nature Research Reporting Summary linked to this article.

## Data availability

Sequencing data that support the findings of this study have been deposited at the Gene Expression Omnibus under the accession code GSE180914. The PANTHER GO-SLIM database and REACTOME pathway database are publicly available (www.pantherdb.org/panther/goSlim.jsp, https://reactome.org/). All other data supporting the findings of this study are available from the corresponding authors on reasonable request. Source data are provided with this paper.

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

## Acknowledgements

We thank S. Kagioglou for technical help; E. Balado, C. Donaldson, E. Arvidsson, A. Karakatsanis and A. Giannakakis for their help with raising animals; T. Davit-Béal for the donation of adult *P. waltl*; A. Becker and A. Ngezahayo for sharing the Strep-Tag II–c-CPE plasmid and protein purification protocol; L. Haag and K. Hultenby for their help with the interpretation of the TEM images; G. Palano and E. Hansson for helping with the establishment of the cell-dissociation protocol; E. Tanaka for scientific discussions leading to the use of TAT-Cre for lineage tracing; F. Salomons for helping with microscopy and automated image analysis training; and the members of the Chien and Simon laboratories for rigorous scientific discussions. Analysis of *P. waltl* gene expression was enabled by resources provided by the Swedish National Infrastructure for Computing at the Uppsala Multidisciplinary Center for Advanced Computational Science, partially funded by the Swedish Research Council through grant agreement no. 2018-05973. Some illustrations (Figs. 2c, 5a and 7a) were created with BioRender.com. This work was supported by a Swedish Research Council Distinguished Professor Grant (Dnr 541-2013-8351; K.R.C.), the Swedish Heart Lung Foundation (grant no. 20140623; K.R.C.), the Wallenberg Foundation (KAW Dnr 2013.0028; K.R.C.), AstraZeneca Pharmaceuticals (ICMC) (K.R.C.), the Karolinska Institutet (K.R.C), the Swedish Research Council (grant no. 2018-02443; A.S.), Cancerfonden (grant no. 19 0417 Pj 01 Hx; A.S), Stiftelsen Olle Engkvist (A.S.), SFO Stem Cells and Regenerative Medicine (A.S.), a EMBO long-term fellowship (grant no. ALTF-729-2015; E.E), a NIH Ruth Kirschstein postdoctoral fellowship (grant no. F32GM117806; A.E.) and the German Research Foundation (grant no. GO3220/1-1; A.G.).

## Author contributions

Conceptualization: N.W. and K.R.C. initiated the salamander heart regeneration project, with input from A.S. E.E. A.S. and K.R.C. conceived and co-designed the current CLDN6+ EPDC study. Methodology: E.E., N.W., A.J.A. and H.W. Formal analysis: E.E., C.Y.T.Y., Y.-L.T., A.E. and A.G. Investigation: E.E., C.Y.T.Y., T.S., C.H.U. and J.S. Resources: A.J.A. and H.W. Writing of the original draft: E.E., C.Y.T.Y. and A.S. Writing—review and editing: E.E., C.Y.T.Y, A.S. and K.R.C. Visualization: E.E., C.Y.T.Y. and Y.-L.T. Funding acquisition: A.S. and K.R.C.

## Funding

## Competing interests

K.R.C. is a scientific founder and equity holder in Moderna Therapeutics and Procella Therapeutics, and chair of the External Science Panel for AstraZeneca. The remaining authors declare no competing interests.

## Additional information

**Extended data** is available for this paper at https://doi.org/10.1038/s41556-022-00902-2.

**Correspondence and requests for materials** should be addressed to Elif Eroglu, András Simon or Kenneth R. Chien.

**a**

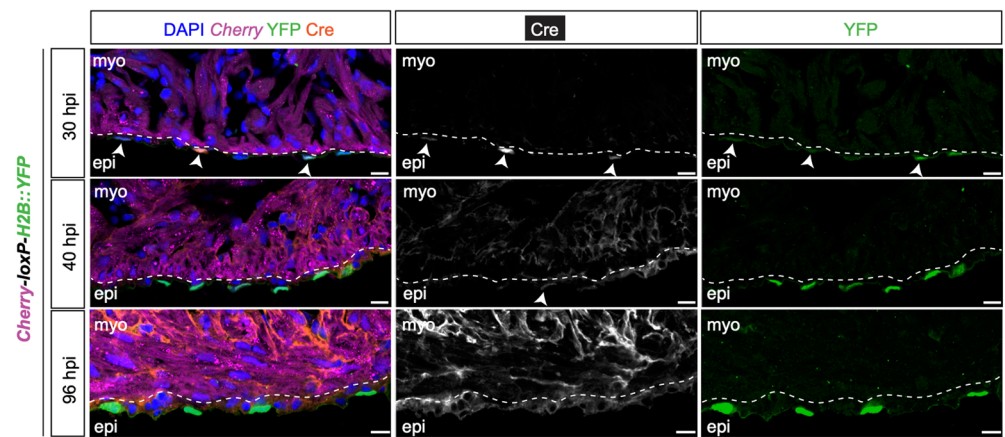

**b**

Epicardium | Myocardium

30 hpi | 40 hpi | 96 hpi

Raw Integrated Density of Cre signal per nucleus

Mean Intensity of Cre signal per nucleus

**Extended Data Fig. 1 | See next page for caption.**

**Extended Data Fig. 1 | TAT-Cre recombinase transduces epicardial cells. a**, Nuclear Cre expression is detectable 30-h post injection in the epicardial layer. Representative immunostaining showing DAPI, Cherry, YFP and Cre at 30-, 40- and 96-h post TAT-Cre recombinase injection (h.p.i.) into the pericardial sac of *Cherry-loxP-H2B::YFP*. Arrowheads mark nuclear Cre+ epicardial cells. Dashed line marks epi-myo border. Scale bars, 20 μm. Data shown represent 9 animals at 30 h.p.i., 6 animals at 40 h.p.i., 5 animals at 96 h.p.i. **b**, TAT-Cre recombinase does not transduce myocardial cell nuclei. Quantification of the number of Cre+ nuclei in the epicardium and myocardium at 30-, 40- and 96-h.p.i. Dashed line at mean intensity of 25 depicts the threshold separating negative versus positive nuclei. Note that scale is adjusted based on nuclear Cre signal in the epicardium at 30-h.p.i. Insets show data plotted with different scale to provide magnified view. See methods for details of the quantification.

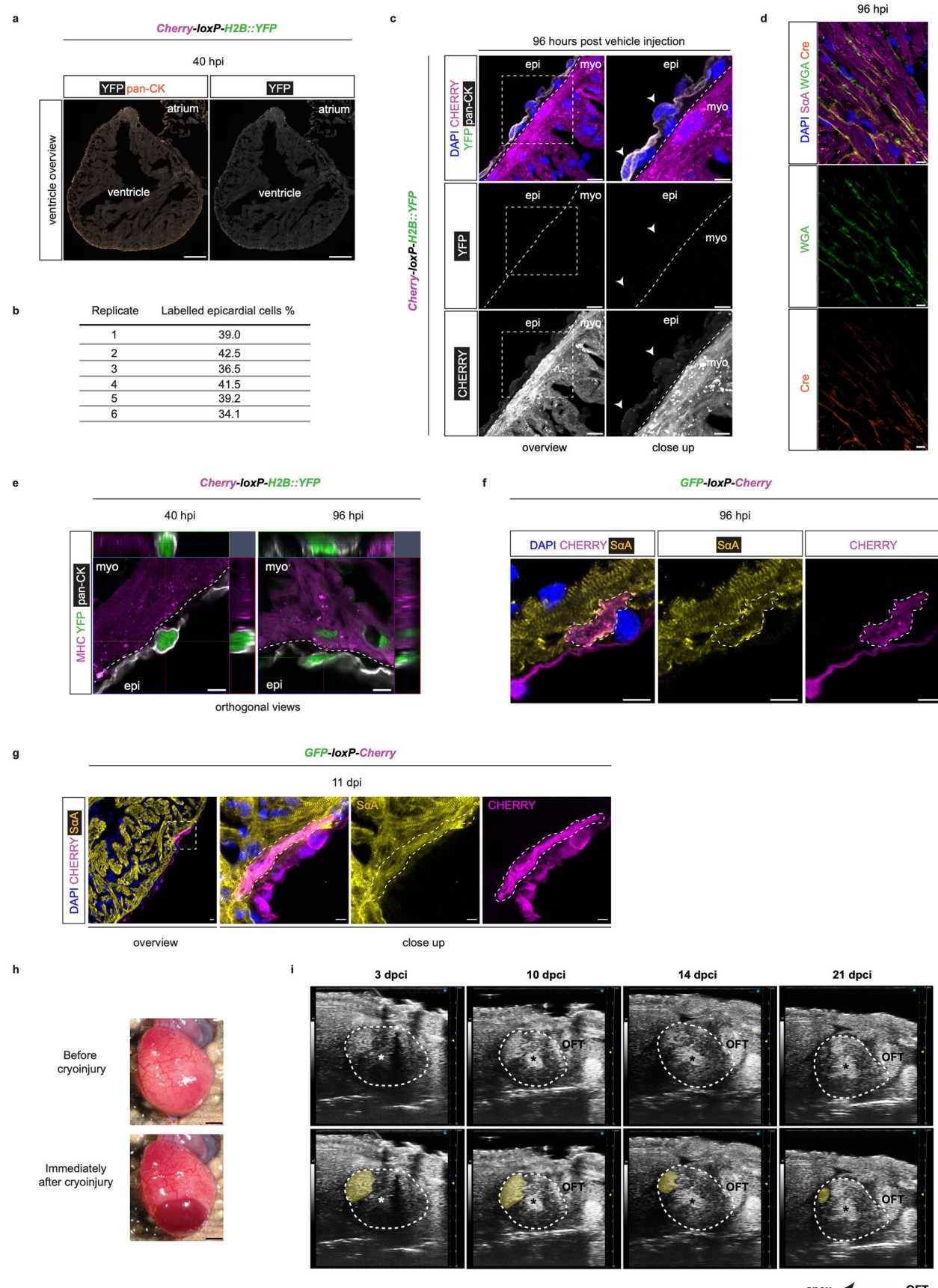

**Extended Data Fig. 2 | See next page for caption.**

**Extended Data Fig. 2 | TAT-Cre recombinase selectively labels epicardial cells. a**, TAT-Cre recombinase induces epicardium specific recombination. Overview image of a ventricle immunostained for YFP and pan-CK 40-h post TAT-Cre injection (h.p.i.). Scale bars, 500 µm. **b**, Table showing the percent labelled epicardial cells at 40-h.p.i. Pan-CK⁺ cells were counted to determine the number of epicardial cells. **c**, Vehicle injection does not result in recombination. Representative immunostaining for DAPI, CHERRY, YFP and pan-CK 96-h post vehicle injection. Dashed line marks epi-myo border. Boxed area is magnified. Arrowheads mark epicardial cells. Scale bars, 20 µm (left), 10 µm (right). **d**, Cre is localized to the myocardial extracellular matrix at 96-h.p.i. Representative immunostaining showing DAPI, SαA, wheat-germ agglutinin (WGA) and Cre at 96-h post TAT-Cre recombinase injection into the pericardial sac of *Cherry-loxP-H2B::YFP*. Scale bars, 10 µm. **e**, Representative immunostaining for MHC, YFP and pan-CK on TAT-Cre recombinase injected *Cherry-loxP-H2B::YFP* heart sections at 40- and 96-h.p.i. Orthogonal views of YFP⁺ cells in the epicardium (at 40-h.p.i. and 96-h.p.i.) and myocardium (at 96-h.p.i.). Dashed line shows epi-myo border. Scale bars, 10 µm. **f**, At 96-h.p.i., epicardium-derived cardiomyocytes are morphologically immature. Representative immunostaining for DAPI, CHERRY and SαA on TAT-Cre recombinase injected *GFP-loxP-Cherry* heart section at 96-h.p.i. CHERRY⁺ epicardium-derived cardiomyocyte is outlined. Scale bars, 10 µm. **g**, At 11-dpi, epicardium-derived cardiomyocytes show sarcomeric organization. Representative immunostaining for DAPI, CHERRY and SαA on TAT-Cre recombinase injected *GFP-loxP-Cherry* heart section at 11-dpi. CHERRY⁺ epicardium-derived cardiomyocyte is outlined. Boxed area shown in the overview image is magnified. Scale bars, 20 µm (overview), 10 µm (close-up). **h**, Representative images of the ventricle before and immediately after cryoinjury, showing the damage to the apex. Scale bars, 1 mm. **i**, Representative still images of longitudinal axis views of the regenerating salamander heart at 3-, 10-, 14- and 21-d.p.ci. obtained by ultrasound imaging. Ventricle is outlined by dashed lines and injury area is pseudocolored in the bottom panels. Asterisk marks the lumen. OFT, outflow tract. **a**, **b**, **c**, **d**, **e**, **f**, **g**, **i**, Data shown represent 6 animals (**a**, **b**, **d**), 3 animals (**c**, **e**), 4 animals (**f**, **g**) and 16 animals (**i**).

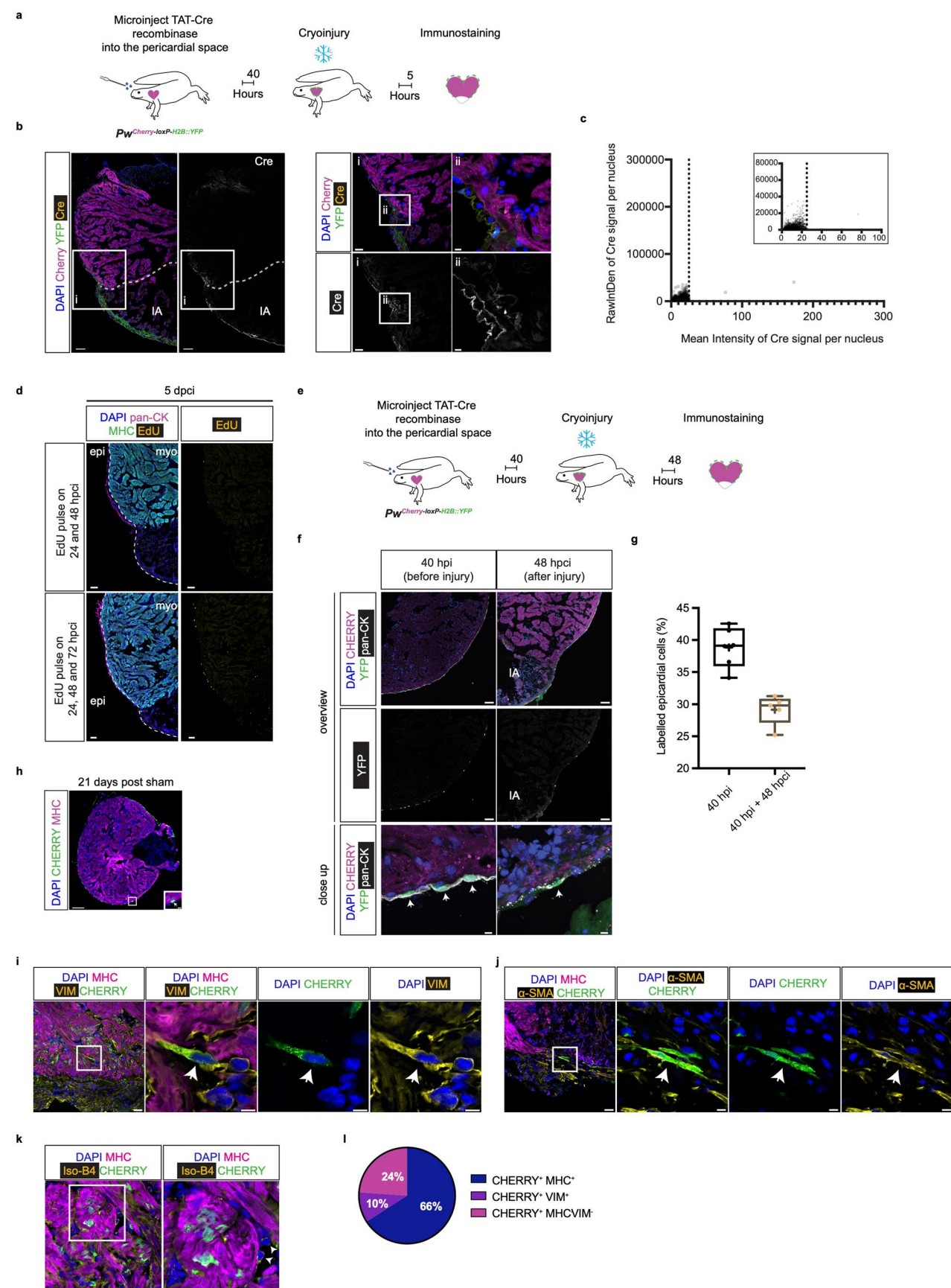

**Extended Data Fig. 3 | See next page for caption.**

**Extended Data Fig. 3 | Salamander epicardial cells are multipotent. a**, Scheme showing the experimental design. **b**, Representative overview and close up images showing immunostaining for DAPI, YFP and Cre. Injury area (IA) is delineated by dashed line. Boxed areas are magnified. Scale bars, 100 μm (overview), 50 μm (i), 10 μm (ii). **c**, Quantification of Cre⁺ nuclei in the injury area and myocardium 5-hpci. 3105 cells were analysed, n = 3 animals. Dashed line shows the threshold. **d**, Representative immunostainings of DAPI, pan-CK, MHC and EdU at 5-d.p.ci. Dashed lines separate epicardium and myocardium. Scale bars, 100 μm. **e**, Schematic showing the experimental design. **f**, Arrows mark YFP⁺ epicardial cells. Scale bars, 100 μm (overview), 10 μm (close-up). Data shown represent 6 animals (40 h.p.i.) and 5 animals (48 hpci). **g**, Quantification of labelled epicardial cell frequency (percentage of YFP⁺, pan-CK⁺ cells) before (n = 6 animals) and after cryoinjury (n = 5 animals). Box and whiskers plot shows mean (+), median, quartiles (boxes), and range (whiskers). **h**, Representative immunostaining on TAT-Cre recombinase injected *GFP-loxP-Cherry* heart sections, showing DAPI, CHERRY and MHC at 21-days post sham operation. Boxed area is shown in the inset, arrow marking a labelled cardiomyocyte. Scale bars, 200 μm (overview), 20 μm (inset). **i-l**, Epicardium-derived fibroblasts/myofibroblasts are found in the injured area at 21-d.p.ci. Experimental scheme as shown in Fig. 2c. **i**, Representative immunostainings showing DAPI, MHC, Vimentin (Vim) and CHERRY at 21-d.p.ci. Boxed area is magnified. Arrow marks CHERRY⁺, Vim⁺ cell. Scale bars, 20 μm (overview), 10 μm (close up). **j**, Representative immunostainings showing DAPI, MHC, α-Smooth muscle actin (α-SMA) and CHERRY at 21-d.p.ci. Boxed area is magnified. Arrow marks CHERRY⁺, α-SMA⁺ cell. Scale bars, 50 μm (overview), 10 μm (close up). **k**, Representative immunostainings showing DAPI, MHC, Isolectin-B4 and CHERRY at 21-d.p.ci. Boxed area is magnified. Arrows show CHERRY⁻, Isolectin-B4⁺ cells. Scale bars, 20 μm (overview), 10 μm (close up). **l**, Percentage of epicardium-derived cells at 21-d.p.ci. **b**, **d**, **h**, **i**, **j**, **k,l**, Data shown represent 3 animals (**b**, **d**, **l**) and 6 animals (**h**, **i**, **j**, **k**).

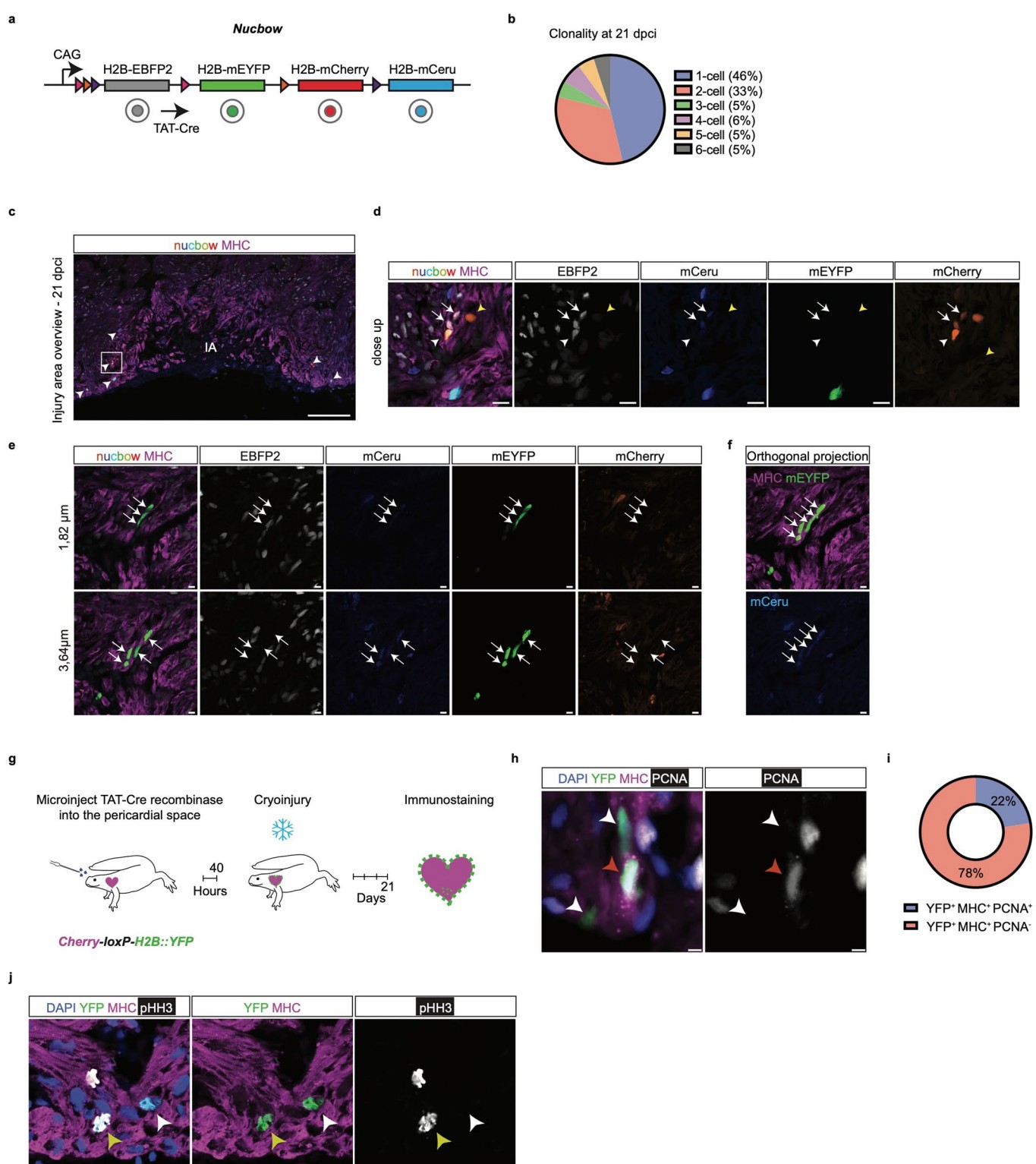

**Extended Data Fig. 4 | See next page for caption.**

**Extended Data Fig. 4 | EPDCs expand clonally. a**, Schematic drawing showing the *Nucbow* reporter construct[106]. **b**, Pie chart showing the abundance of 1- to 6-cell epicardium-derived cardiomyocyte clones at 21-d.p.ci. n = 4 animals, 80 clones. **c**, Representative image of heart apex showing multi-colour labelling induced by TAT-Cre recombinase in *Nucbow* reporter. EBFP2 (white), mEYFP (green), mCherry (orange), mCerulean (blue) and MHC (magenta). Arrows mark clones of various colours and sizes. IA, injury area. Boxed area is shown in panel d. Scale bar, 200 µm. **d**, Representative image showing a two-cell clone (arrows) expressing EBFP2, mCerulean and mCherry. Note that adjacent labelled cells (white and yellow arrowheads) do not show same colour combinations and therefore do not belong to the same clone. Scale bar, 10 µm. **e**, Representative images showing two different Z-layers of a five-cell clone expressing mEYFP and mCeru. Arrows mark mEYFP+, mCeru+ cells belonging to the clone. See Extended Data Fig. 4f for orthogonal projection. Scale bars, 10 µm. **f**, Orthogonal projection of the image shown in panel e. Scale bars, 10 µm. **g**, Scheme depicting the experimental design to assess the proliferative potential of epicardium-derived cardiomyocytes. **h**, Epicardium-derived cardiomyocytes are proliferative. Representative immunostaining for DAPI, YFP, MHC and PCNA at 21-d.p.ci. Arrowheads mark epicardium-derived YFP+ cardiomyocytes that are PCNA− (white) and PCNA+ (red). Scale bar, 10 µm. **i**, Quantification of PCNA+ epicardium-derived cardiomyocytes percentage. **j**, Representative immunostaining for DAPI, YFP, MHC and Phospho-Histone H3 (pH3) at 21-d.p.ci. Arrowheads mark epicardium-derived YFP+ cardiomyocytes that are pH3− (white) and pH3+ (yellow). Scale bars, 10 µm. (**c**, **d**, **e**, **f**, **h**, **i**, **j**) Data shown represent 4 animals.

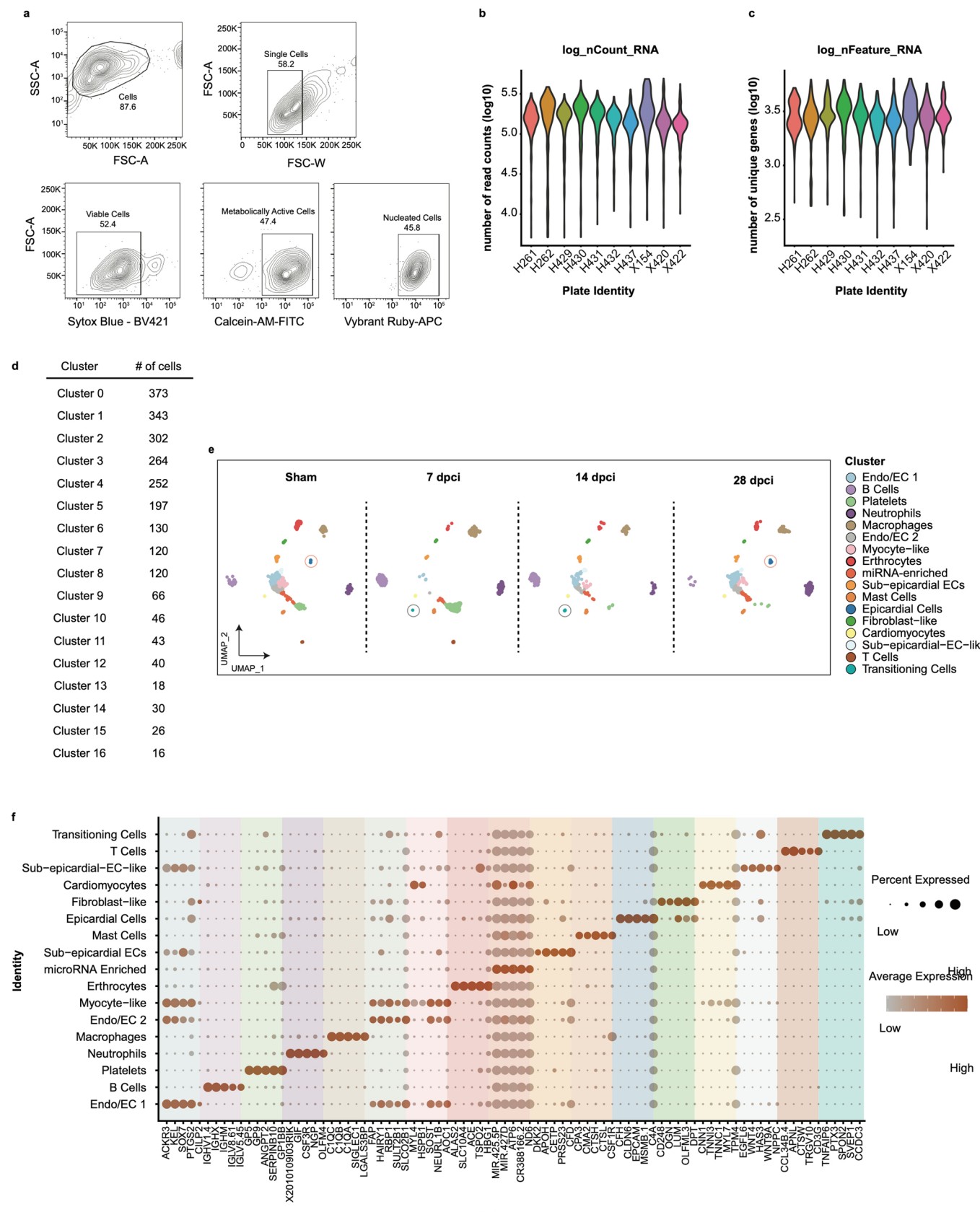

**Extended Data Fig. 5 |** See next page for caption.

**Extended Data Fig. 5 | Quality control of scRNA-seq data. a**, FACS plots showing the gating strategy to obtain live, metabolically active and nucleated cells. Percentage of cells after each gate in comparison to all cells is displayed. **b**, Violin plot of log10 of total read counts of each individual cell in different plates obtained from single-cell sequencing. See Supplementary Table S1 for information on plate identity. **c**, Violin plot of log10 of the number of unique genes of each individual cell in different plates obtained from single-cell sequencing. See Supplementary Table S1 for information on plate identity. **d**, Table showing the number of cells recovered in each cell cluster. **e**, UMAP plots showing the distribution of recovered single-cells according to time point. Epicardial cells and transitioning cells are circled to highlight dynamic recovery. **f**, Dot plot showing the relative expression of marker genes for the 17 clusters identified in the scRNA-seq data as presented in Fig. 3b.

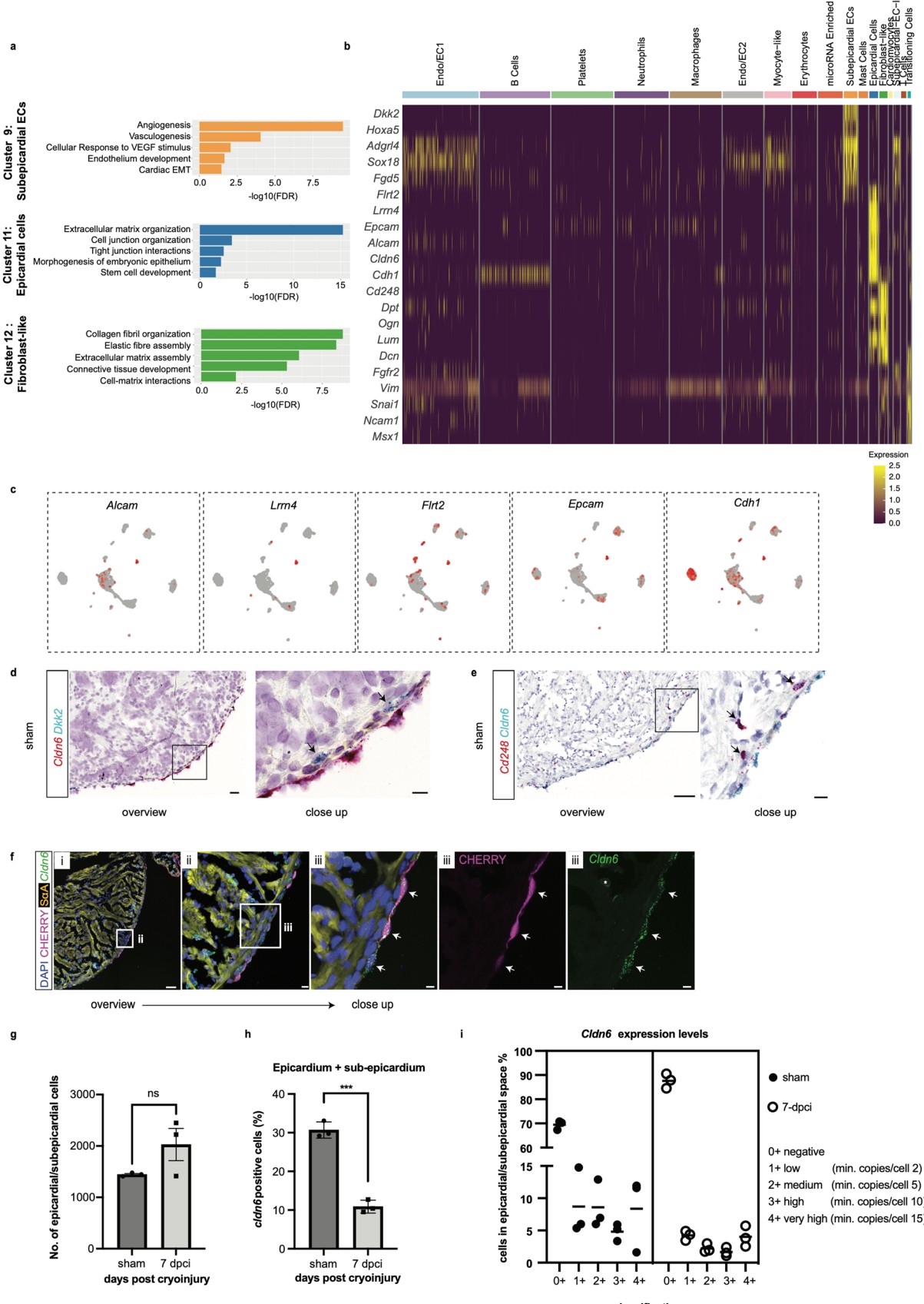

**Extended Data Fig. 6 | See next page for caption.**

**Extended Data Fig. 6 | Cluster 11 represents the epicardium-proper. a**, Representative GO terms upregulated in candidate epicardium/epicardium-derived cell clusters. **b**, Heatmap showing select marker gene expression presented in Fig. 3d across all cell clusters. **c**, UMAP showing the expression of epithelial/ mesothelial markers expressed in cluster 11. **d**, *Dkk2* expressing cells are located to the subepicardium (marked by arrows). Representative images showing the expression of *Cldn6* (red) and *Dkk2* (cyan) in the sham operated hearts. Scale bars, overview; 50 μm close-up; 20 μm. Data shown represent 3 animals. **e**, *Cd248* expressing cells are in the subepicardium and myocardial interstitium (marked by arrows). Representative images showing the expression of *Cd248* (red) and *Cldn6* (cyan) in the sham operated hearts. Scale bars, overview; 100 μm close-up; 20 μm. Data shown represent 3 animals. **f**, Epicardial cells labelled by the TAT-Cre recombinase are *Cldn6*⁺. Representative images showing *in situ* hybridization combined with immunostaining for *Cldn6* mRNA, DAPI, CHERRY and SαA on sections from TAT-Cre recombinase injected *GFP-loxP-Cherry*. Arrows mark *Cldn6*⁺ epicardial cells labelled by TAT-Cre induced recombination. Scale bars, 100 μm (i), 20 μm (ii), 10 μm (iii). Data shown represent 4 animals. **g**, Epicardium/sub-epicardium thickens in response to the injury. Quantification of epicardial/subepicardial cell numbers of the sham operated (n = 3 animals) and injured hearts (7-d.p.ci.) (n = 3 animals). Student's t-test, unpaired, two-tailed. ns p = 0.1377. Data are represented as mean values ± sem. **h**, Cryoinjury reduces the number of cells expressing *Cldn6* mRNA in the epicardium. Quantification of cells expressing *Cldn6* mRNA in the epicardial and sub-epicardial space of the sham operated and injured hearts (n = 3 animals per group). Student's t-test, unpaired, two-tailed, ***p = 0.0002, Data are represented as mean values ± SD. **i**, *Cldn6* mRNA expression levels decrease in response to the injury. Quantification of *Cldn6* mRNA expression levels in the epicardium/subepicardium cells of the sham operated (n = 3 animals) and injured hearts (7 d.p.ci.) (n = 3 animals).

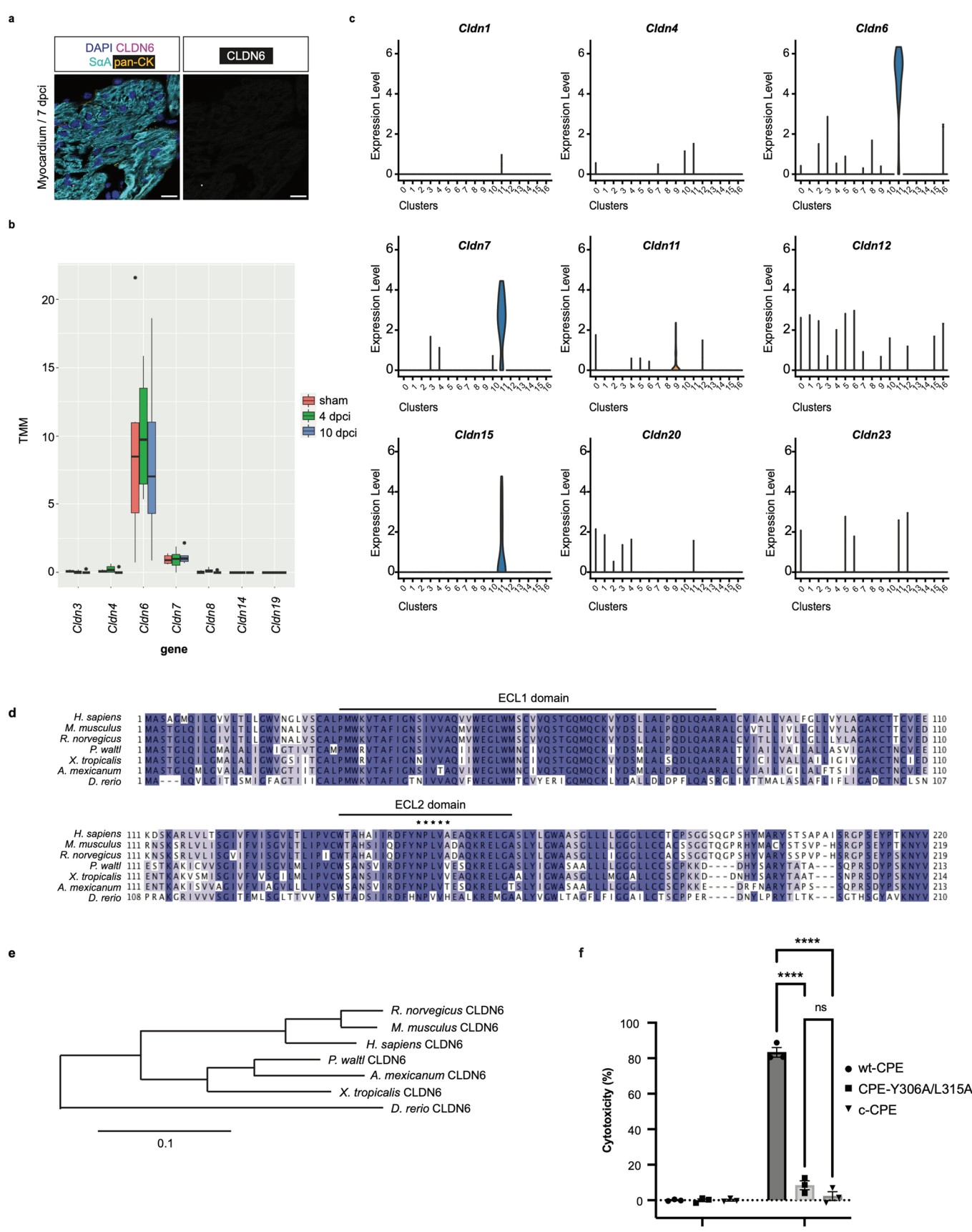

**Extended Data Fig. 7 | See next page for caption.**

**Extended Data Fig. 7 | CPE-sensitive *Cldn* expression is restricted to the epicardial cells. a**, CLDN6 is absent in the myocardium. Representative immunostaining showing DAPI, CLDN6, SαA and pan-CK at 7-d.p.ci. See Fig. 4c for overview image. Scale bar, 10 μm. Data shown represent 4 animals. **b**, *Cldn6* and *Cldn7* are the only CPE sensitive *Cldn* gene family members expressed in the salamander heart. Expression of CPE sensitive *Cldn* genes in the salamander heart as determined by bulk RNA sequencing of the sham operated and injured hearts (4-d.p.ci. and 10-d.p.ci., n = 3 hearts per time point). Box and whiskers plot shows median, quartiles (boxes), and range (whiskers). TMM, Trimmed mean of M values. Note that *Cldn9* gene is not annotated in the *Pw* genome and therefore not shown despite being a CPE-sensitive CLDN. **c**, Violin plots showing the distribution of expression for the *Cldn* gene family members. *Cldn6*, *Cldn7* and *Cldn15* are expressed in the epicardial cluster 11. *Cldn11*, a nonclassical CLDN family member that is not a CPE receptor, is expressed in cluster 9[107]. **d**, CLDN6 protein is well conserved. Amino acid sequence alignment of *Pw* CLDN6 with sequences from other vertebrates, *H. sapiens*, *M. musculus*, *R. norvegicus*, *A. mexicanum*, *X. leavis* and *D. rerio*. Extracellular loop 1 (ECL1) and ECL2 domains are highlighted[108]. Asterisk shows the key motif NP(V/L)(V/L)(P/A) within the ECL2 domain required for the CLDN-CPE interaction[109]. **e**, A phylogenetic tree of vertebrate CLDN6 protein. The scale bar indicates evolutionary distance. **f**, Salamander *Cldn6* sensitizes HEK293T cells to wt-CPE. Quantification of cytotoxicity % in response to treatment of sensitized HEK293T cells with wt-CPE, CPE-Y306A/L315A or c-CPE for 3 hrs. Two-way ANOVA, Tukey's multiple comparisons test. ****p < 0.0001, error bars represent sem. n = 10,000 HEK293T cells examined in 3 independent experiments for each condition.

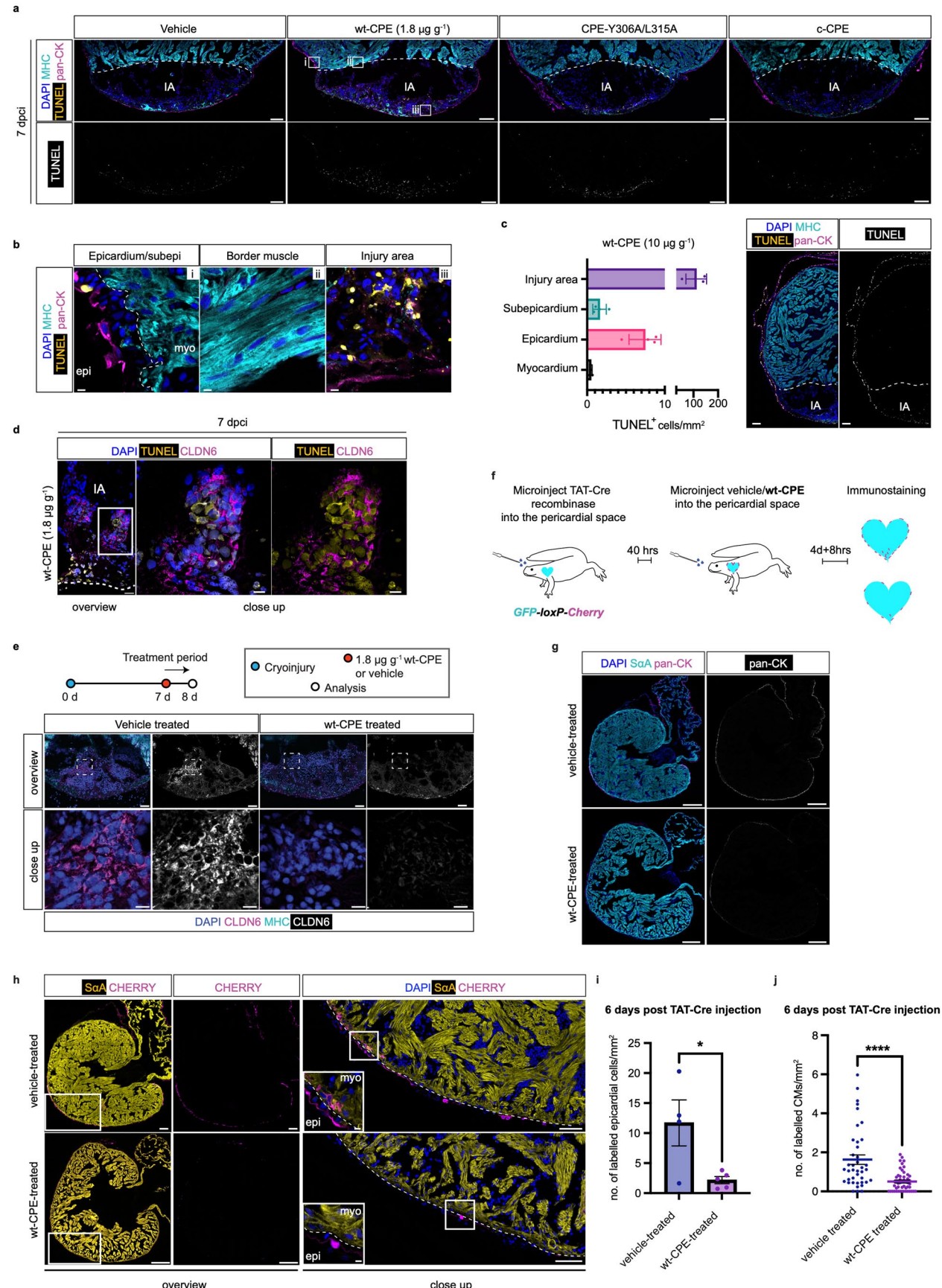

**Extended Data Fig. 8 | See next page for caption.**

**Extended Data Fig. 8 | wt-CPE treatment ablates CLDN6$^+$ epicardium and/or EPDCs in a dose-dependent manner. a**, wt-CPE treatment causes cell death. Representative immunostainings showing TUNEL$^+$ cells following treatment for 6 h. Shown are overview images of apex. Healthy myocardium is separated by dashed lines. Boxed areas are magnified in panel b. IA, injury area. Scale bars, 200 μm. **b**, Representative close-up images. Scale bars, 10 μm. **c**, (Left) Quantification of TUNEL$^+$ cells upon high dose wt-CPE treatment for 6 hrs. Error bars represent ± SD. (Right) Representative images of a ventricle section immunostained for DAPI, MHC, TUNEL and pan-CK. Scale bars, 200 μm. **d**, Representative immunostainings showing CLDN6$^+$, TUNEL$^+$ EPDCs in the injury area, following wt-CPE treatment for 6 hrs. Scale bars, 50 μm (overview), 20 μm (close up). **e**, wt-CPE treatment ablates CLDN6$^+$ cells in the injured area. Scheme showing the experimental design. Representative immunostaining of DAPI, CLDN6 and MHC. Scale bars, 100 μm (overview), 20 μm (close-up). **f**, Scheme showing the experimental design to assess the impact of epicardial cell ablation under homeostasis. **g**, wt-CPE treatment disrupts the epicardial layer. Representative immunostainings for DAPI, SαA and pan-CK. Scale bars, 500 μm. **h and i**, wt-CPE treatment reduces the number of epicardial cells. **h**, Overview and close up images showing DAPI, SαA and CHERRY immunostaining 4 days and 8 h post vehicle or wt-CPE injection. Scale bars, 500 μm (overview), 250 μm (close up), 10 μm (insets). **i**, Quantification of CHERRY$^+$ epicardial cells 104 h post wt-CPE treatment. vehicle; n = 4 animals, wt-CPE; n = 5 animals. Student's t-test, unpaired, two-tailed. *p = 0.0275. Error bars represent ± sem. **j**, Epicardial cell ablation reduces the number of CHERRY$^+$ cardiomyocytes. Quantification of CHERRY$^+$, SαA$^+$ cardiomyocytes 104 h post wt-CPE treatment. 10 sections for each animal are plotted. Vehicle; n = 4 animals, wt-CPE; n = 5 animals. Student's t-test, unpaired, two-tailed. ****p < 0.0001. (**a**, **b**, **c**, **d**, **e**, **g**, **h**) Data shown represent vehicle n = 7 animals, wt-CPE n = 4 animals, Y306A/315A n = 5 animals, c-CPE n = 4 animals (**a**), 4 animals (**b**, **c**), 3 animals (**d**), 3 animals for each group (**e**), 4 animals for the vehicle group and 5 animals for the wt-CPE injected group (**g**, **h**).

**a**

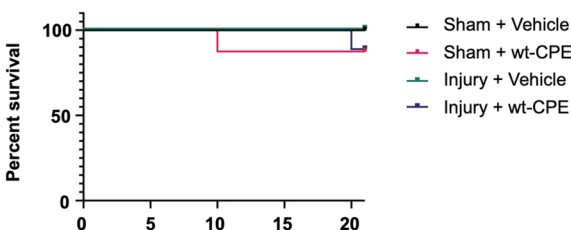

**b**

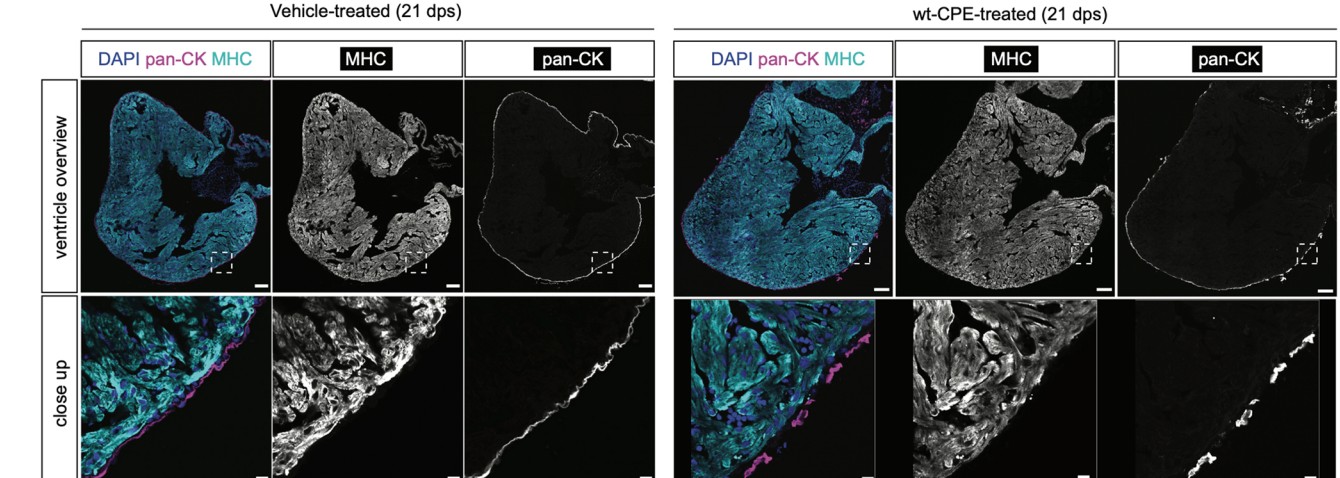

**c**

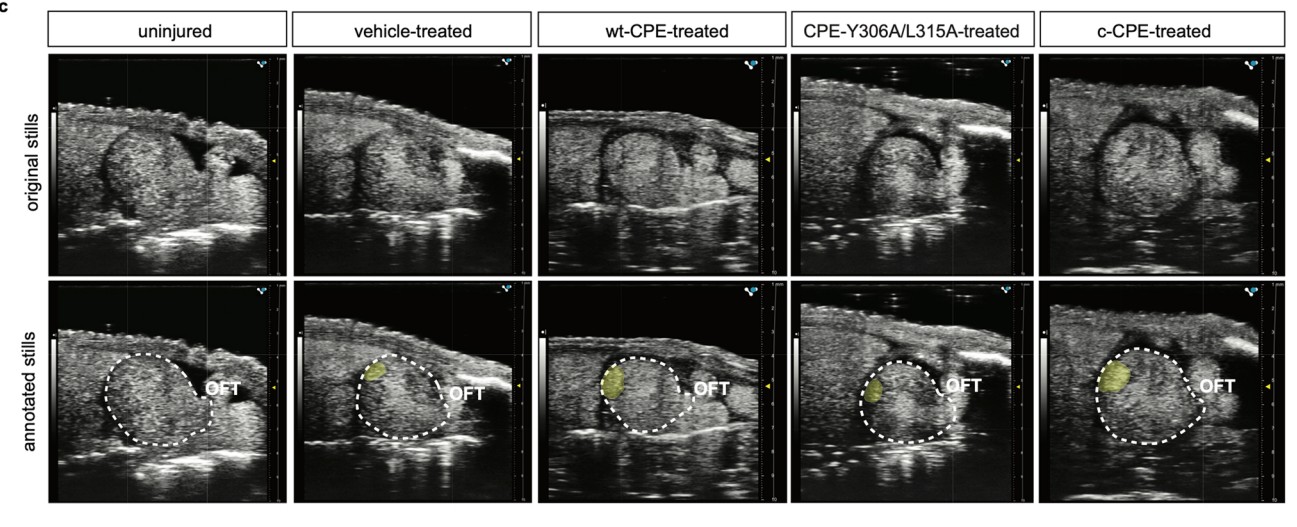

**Extended Data Fig. 9 | wt-CPE treatment does not affect healthy myocardium. a**, Survival curve showing the effect of wt-CPE treatment over the course of 21 days for the following treatment groups: sham + vehicle (n = 4 animals), sham + wt-CPE (n = 8 animals), injured + vehicle (n = 9 animals) and injured + wt-CPE (n = 9 animals). Vehicle or wt-CPE injection scheme as depicted in Fig. 6a. **b**, Immunofluorescence staining for MHC shows unaffected muscle morphology on sham operated, wt-CPE-treated hearts after 21 days. Overview and close-up images of vehicle or wt-CPE-treated heart sections stained for DAPI, pan-CK and MHC. wt-CPE treatment had a minor long-term effect on epicardial integrity as shown by pan-CK staining, while muscle tissue was unaffected. Scale bars, 200 μm (overview), 10 μm (close-up). Data shown represent 6 animals for each group. **c**, Representative still images of the Supplementary Movies. Ventricles are outlined by dashed lines. The injury area is pseudocolored in the bottom panels. Data shown represent 8 animals for each group.

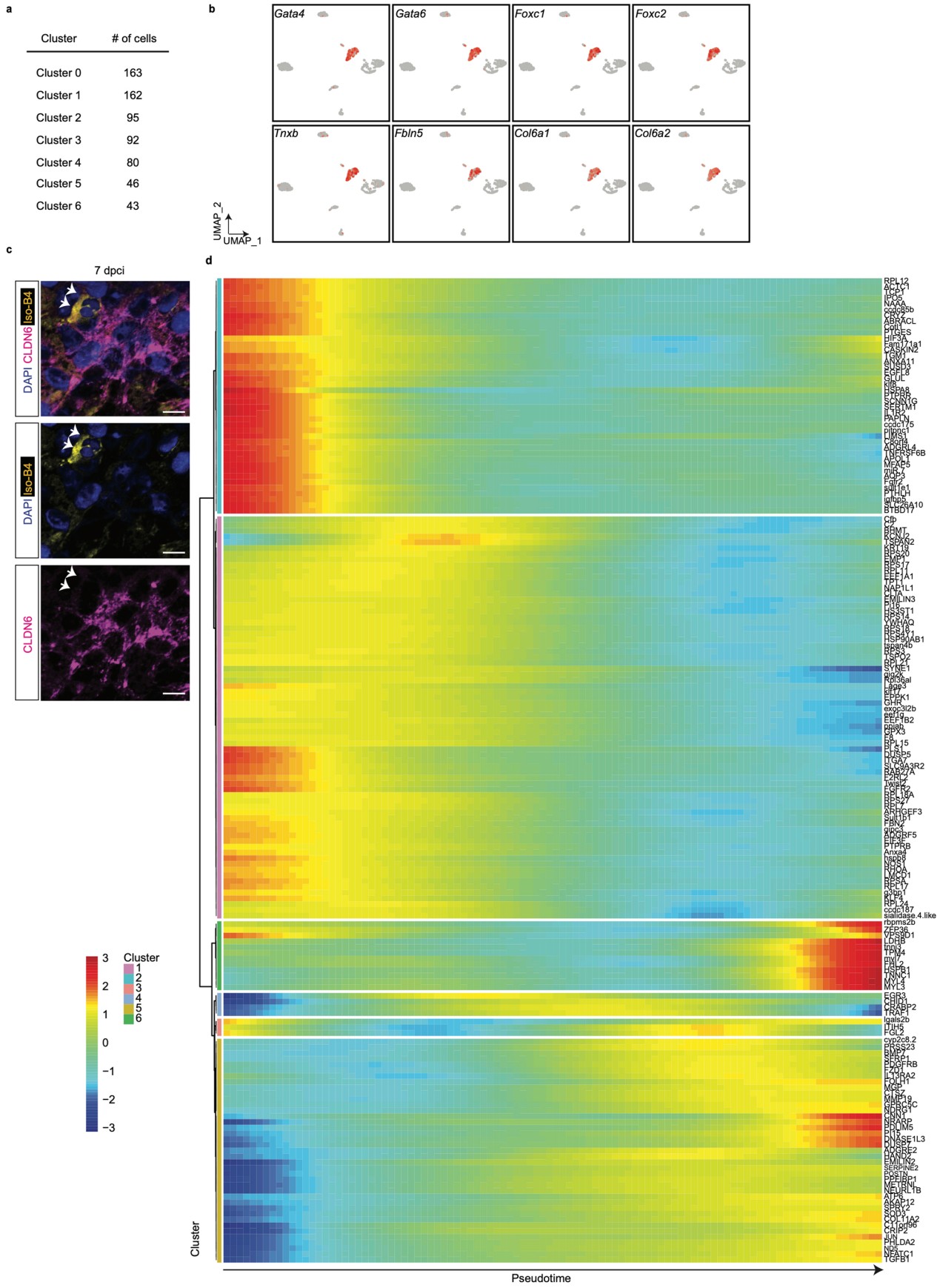

**Extended Data Fig. 10 | See next page for caption.**

**Extended Data Fig. 10 | Cardiogenic transcription programme is activated in the epicardium-derived intermediate cells. a**, Table showing the number of cells recovered in each cluster. **b**, UMAP showing the expression of selected genes expressed in the intermediate cells. **c**, Epicardium-derived CLDN6+ intermediates do not express endothelial cell marker Isolectin-B4. Representative immunostaining showing DAPI, CLDN6 and Isolectin-B4. Arrows mark Isolectin-B4+ CLDN6− cells. Scale bars, 10 μm. Data shown represent 3 animals. **d**, Heatmap showing differentially expressed genes along the pseudotime. Note that "cluster" refers to the hierarchical clustering of differentially expressed genes and not the UMAP clusters.

# Reporting Summary

## Statistics

For all statistical analyses, confirm that the following items are present in the figure legend, table legend, main text, or Methods section.

| n/a | Confirmed | |
|---|---|---|
| ☐ | ☒ | The exact sample size ($n$) for each experimental group/condition, given as a discrete number and unit of measurement |
| ☐ | ☒ | A statement on whether measurements were taken from distinct samples or whether the same sample was measured repeatedly |
| ☐ | ☒ | The statistical test(s) used AND whether they are one- or two-sided<br>*Only common tests should be described solely by name; describe more complex techniques in the Methods section.* |
| ☒ | ☐ | A description of all covariates tested |
| ☐ | ☒ | A description of any assumptions or corrections, such as tests of normality and adjustment for multiple comparisons |
| ☐ | ☒ | A full description of the statistical parameters including central tendency (e.g. means) or other basic estimates (e.g. regression coefficient) AND variation (e.g. standard deviation) or associated estimates of uncertainty (e.g. confidence intervals) |
| ☐ | ☒ | For null hypothesis testing, the test statistic (e.g. $F$, $t$, $r$) with confidence intervals, effect sizes, degrees of freedom and $P$ value noted<br>*Give P values as exact values whenever suitable.* |
| ☒ | ☐ | For Bayesian analysis, information on the choice of priors and Markov chain Monte Carlo settings |
| ☒ | ☐ | For hierarchical and complex designs, identification of the appropriate level for tests and full reporting of outcomes |
| ☒ | ☐ | Estimates of effect sizes (e.g. Cohen's $d$, Pearson's $r$), indicating how they were calculated |

*Our web collection on statistics for biologists contains articles on many of the points above.*

## Software and code

Policy information about availability of computer code

| | |
|---|---|
| Data collection | Images were acquired using Zen Blue (Carl Zeiss, v3.1). Echocardiography data was collected using the Vevo Lab 3.2.0. FACS data was collected using FACSDiva (BD Biosciences, v8.0.3). |
| Data analysis | -Sc-RNA-seq reads were mapped using STAR (v.2.5.3a).<br>-Single cell RNA sequencing analysis was performed using Seurat package version 4.0.1. All functions and parameters were described in detail in the Methods and Supplementary Information. All scripts are available upon request.<br>-Top 300 marker genes with the highest log fold change and a p-value < 0.05 were subjected to overrepresentation analysis using the Protein ANalysis THrough Evolutionary Relationships (PANTHER, Version 15) tool.<br>-Multiple sequence alignment of CLDN6 was performed with T-Coffee in default setting using the EMBL-EBI.<br>-Phylogenic tree was generated using the ClusterW2 package in the EMBL-EBI API. The alignment and tree were visualized using Jalview and TreeDyn.<br>-Monocle 3 (version 0.2.3.0) was used for single-cell trajectories and pseudo-time analysis, based on the UMAP generated in the Seurat analysis.<br>-ggplot2 R package (version 3.3.5) was used to generate scatter plots, bar plots, dot plots, violin plots and other data representation graphs.<br>-Statistical tests were performed using Graphpad Prism 9.1.2.<br>-Echocardiography data was analyzed using Vevo Lab 3.2.0.<br>-Images were analyzed using Zen Blue (Carl Zeiss, v3.1) and CaseViewer (v2.2.1).<br>-FACS data was analyzed using FACSDiva (BD Biosciences, v8.0.3) and FlowJo (v10.7.1).<br>-HALO (V3.0.311.267) software was used to quantify the RNAscope data.<br>-Image J (Fiji, version 2.0.0-rc-69/1.52p) was used to quantify the amount of Cre signal in cell nuclei. |

For manuscripts utilizing custom algorithms or software that are central to the research but not yet described in published literature, software must be made available to editors and reviewers. We strongly encourage code deposition in a community repository (e.g. GitHub). See the Nature Portfolio guidelines for submitting code & software for further information.

## Data

Policy information about availability of data

All manuscripts must include a data availability statement. This statement should provide the following information, where applicable:
- Accession codes, unique identifiers, or web links for publicly available datasets
- A description of any restrictions on data availability
- For clinical datasets or third party data, please ensure that the statement adheres to our policy

Sequencing data that support the findings of this study have been deposited at the Gene Expression Omnibus (GEO) under the accession code GSE180914. http://PANTHER GO-SLIM database and REACTOME pathway database are publicly available (www.pantherdb.org/panther/goSlim.jsp, https://reactome.org/). Source data are provided with this study. All other data supporting the findings of this study are available from the corresponding authors on reasonable request.

# Field-specific reporting

Please select the one below that is the best fit for your research. If you are not sure, read the appropriate sections before making your selection.

☒ Life sciences          ☐ Behavioural & social sciences          ☐ Ecological, evolutionary & environmental sciences

For a reference copy of the document with all sections, see nature.com/documents/nr-reporting-summary-flat.pdf

# Life sciences study design

All studies must disclose on these points even when the disclosure is negative.

| | |
|---|---|
| Sample size | No statistical methods were used to predetermine sample sizes. For both experiments involving animals and/or cell lines sample sizes were empirically estimated on the basis of pilot experiments and previously performed experiments with similar setup to provide sufficient sample sizes for statistical analysis. For experiments where animal to animal variation was high we typically employed n > 8 animals. For experiments where animal to animal variation was low we typically employed n < 8 animals. |
| Data exclusions | For the analysis of the scRNA-seq data: Genes expressed in less than three cells and cells with less than 200 expressed genes were omitted. Cells with total read counts lower than 5000 were also discarded.<br>For in vivo experiments no data were excluded, but sometimes an animal died during the course of the experiment, presumably due to the side effects of toxin treatment. In those instances data obtained in the interim analyses were included even if it was not possible to perform an end-point analysis. |
| Replication | The experimental results were obtained from at least three-independent experiments. All attempts were successful. |
| Randomization | For the studies relating to the CPE toxin and its variants: all animals were pre-screened for heart function via echocardiography to exclude any inherent functional defects. Before the animals were distributed into treatment groups, echocardiography was performed again to measure the injury sizes. If the injury sizes were too large (larger than 28%) or too small (smaller than 17% ) the animals were excluded. Animals with similar sized injuries were distributed into the treatment groups randomly. For experiments other than those involving animals, samples were randomly allocated to experimental groups. |
| Blinding | All experiments relating to the CPE and its variants were performed by two investigators. Echocardiography data was collected and analyzed blindly by the investigators. |

# Reporting for specific materials, systems and methods

We require information from authors about some types of materials, experimental systems and methods used in many studies. Here, indicate whether each material, system or method listed is relevant to your study. If you are not sure if a list item applies to your research, read the appropriate section before selecting a response.

## Materials & experimental systems

| n/a | Involved in the study |
|---|---|
| ☐ | ☒ Antibodies |
| ☐ | ☒ Eukaryotic cell lines |
| ☒ | ☐ Palaeontology and archaeology |
| ☐ | ☒ Animals and other organisms |
| ☒ | ☐ Human research participants |
| ☒ | ☐ Clinical data |
| ☒ | ☐ Dual use research of concern |

## Methods

| n/a | Involved in the study |
|---|---|
| ☒ | ☐ ChIP-seq |
| ☐ | ☒ Flow cytometry |
| ☒ | ☐ MRI-based neuroimaging |

# Antibodies

| | |
|---|---|
| Antibodies used | -guinea pig anti-Cldn6 (1:500, custom made against the peptides TASQPRSDYPSKNYV and CPKKDDHYSARYTATA)<br>-rabbit anti-Cldn6 (1:50, 107059, abcam)<br>-mouse anti-CLDN6 antibody (1:30, MA5-24076, clone 342927, Thermo Fisher)<br>-mouse anti-MYH-1 (1:200, MF-20 concentrate, Myeloma Strain: P3U-1, DSHB)<br>- chicken anti-Vimentin (1:200, AB5733, Millipore)<br>- rabbit anti-α-smooth muscle actin (1:100, ab5694, Abcam)<br>-rabbit anti-Cre recombinase (1:500, ab216262, abcam)<br>-rabbit anti-cytokeratin (1:250, 9377, abcam)<br>-rabbit anti-GFP (1:500, A-6455, Life Technologies)<br>-chicken anti-GFP (1:1000, ab13970, abcam)<br>-mouse anti-Cre recombinase (1:500, C7920, clone 2Q2151, US biological)<br>-rabbit anti-RFP (1:200, 600-401-379, Rockland)<br>-rat anti-RFP (1:200, M11217, clone 16D7, Life Technologies)<br>-mouse anti-α-actinin (1:800, A7811, clone EA-53, Sigma)<br>-Alexa-Fluor 488 Phalloidin (1:500, A12379, Thermo Fisher)<br>-rabbit isotype control (1:100, abcam, ab171870)<br>-Isolectin GS-IB4, Alexa Fluor 488 or Alexa Fluor 647 Conjugate (1:200, I21411 or I32450  Thermo Fisher)<br>-Highly cross-adsorbed Alexa Fluor conjugated secondary antibodies (1:1000, raised in goats, Thermo Fisher) were used. Specifically, anti-chicken 488 (A11039, lot no. 2079383), anti-chicken 647 (A21449, lot no. 1698677), anti-guinea pig 555 (A21435, lot no. 1711692), anti-mouse IgG1 488 (A21121, lot no. 2983196), anti-mouse IgG1 647 (A21240, lot no. 2092265), anti-mouse IgG2b 647 (A21241, lot no.2056280), anti-mouse IgG2b 488 (A21141, lot no. 2228625), anti-mouse IgG2b 568 (A21144, lot no. 2349089), anti-mouse IgG2b 647 (A21242, lot no. 2155295), anti-rabbit 488 (A11034, lot no. 2069632), anti-rabbit 568 (A11011, lot no. 2277758), anti-rabbit 647 (A21245, lot no. 2018272), anti-rat 568 (A11077, lot no. 1692966) |
| Validation | All antibodies are commercially available and validated by the manufacturer except for guinea pig anti-Claudin 6 (Chien Lab). Protein sequence conservation has been considered while choosing antibodies to be used on Pleurodeles waltl tissues. The subcellular localization of all the proteins analyzed in this study has been previously reported. This was used to validate the specificity of the antibody.<br><br>Guinea pig and rabbit anti-Cldn6 antibodies were validated via western blot on HEK293T cell extracts obtained from cells transfected with wild type Pleurodeles waltl Cldn6 or a deletion mutant.<br><br>Anti-MYH-1 (MF-20 concentrate, DSHB) has been validated on salamander tissue in Mercer et al., Developmental Biology, 2013. Anti-Cre recombinase (C7920, US biological) has been validated on Pleurodeles tissue in Joven et al., Development, 2018.<br><br>-rabbit anti-Cldn6 (https://www.citeab.com/antibodies/723509-ab107059-anti-claudin-6-antibody)<br>-mouse anti-CLDN6 antibody (https://www.thermofisher.com/antibody/product/Claudin-6-Antibody-clone-342927-Monoclonal/MA5-24076)<br>-mouse anti-MYH-1 (MF-20 concentrate) (https://dshb.biology.uiowa.edu/MF-20)<br>-chicken anti-Vimentin (https://www.merckmillipore.com/SE/en/product/Anti-Vimentin-Antibody,MM_NF-AB5733)<br>- rabbit anti-α-smooth muscle actin (https://www.abcam.com/alpha-smooth-muscle-actin-antibody-ab5694.html)<br>-rabbit anti-Cre recombinase (https://www.abcam.com/cre-recombinase-antibody-ab216262.html)<br>-rabbit anti-cytokeratin (https://www.abcam.com/wide-spectrum-cytokeratin-antibody-ab9377.html)<br>-rabbit anti-GFP (https://www.thermofisher.com/antibody/product/GFP-Antibody-Polyclonal/A-6455)<br>-chicken anti-GFP (https://www.abcam.com/gfp-antibody-ab13970.html)<br>-mouse anti-Cre recombinase (https://www.usbio.net/antibodies/C7920/Cre-Recombinase)<br>-rabbit anti-RFP (https://www.rockland.com/categories/primary-antibodies/rfp-antibody-pre-adsorbed-600-401-379/)<br>-rat anti-RFP (https://www.thermofisher.com/antibody/product/mCherry-Antibody-clone-16D7-Monoclonal/M11217)<br>-mouse anti-α-actinin  (https://www.sigmaaldrich.com/SE/en/product/sigma/a7811)<br>-Alexa-Fluor 488 Phalloidin (https://www.thermofisher.com/order/catalog/product/A12379)<br>-rabbit isotype control (https://www.abcam.com/rabbit-igg-polyclonal-isotype-control-chip-grade-ab171870.html)<br>-Isolectin GS-IB4, Alexa Fluor 488 Conjugate (https://www.fishersci.com/shop/products/molecular-probes-alexa-fluor-isolectin-gs-ib-sub-4-sub-from-i-griffonia-simplicifolia-i-alexa-fluor-488-conjugate/I21411)<br>-Isolectin GS-IB4, Alexa Fluor 647 Conjugate (https://www.fishersci.com/shop/products/molecular-probes-alexa-fluor-isolectin-gs-ib-sub-4-sub-from-i-griffonia-simplicifolia-i-alexa-fluor-647-conjugate/I32450) |

# Eukaryotic cell lines

Policy information about cell lines

| | |
|---|---|
| Cell line source(s) | HEK293T cells were obtained from the ATCC. |
| Authentication | The cell line was obtained from ATCC and was not revalidated. |
| Mycoplasma contamination | We perform routine mycoplasma checks in the lab but this cell line was not tested for mycoplasma contamination. |
| Commonly misidentified lines<br>(See ICLAC register) | We have checked the ICLAC register and the cell line used in our studies are not on the list of misidentified cell lines. |

# Animals and other organisms

Policy information about studies involving animals; ARRIVE guidelines recommended for reporting animal research

| | |
|---|---|
| Laboratory animals | Post-metamorphic (up to a year old) male/female wild type or transgenic Pleurodeles waltl were used for the salamander experiments. Animals had mixed genetic background. <br> Transgenic lines are as listed below: <br> tgTol2(CAG:loxP-GFP-loxP-Cherry)Simon <br> tgTol2(CAG:loxP-Cherry-loxP-H2B::YFP)Simon <br> tgTol2(CAG:Nucbow)Simon |
| Wild animals | The study did not involve wild animals. |
| Field-collected samples | The study did not involve samples collected from the field. |
| Ethics oversight | All the procedures related to animal handling, care and the treatment in this study were performed according to the guidelines approved by the Jordbruksverket/Sweden under the ethical permit numbers 18190-18 and 5723-2019. |

Note that full information on the approval of the study protocol must also be provided in the manuscript.

# Flow Cytometry

## Plots

Confirm that:

☒ The axis labels state the marker and fluorochrome used (e.g. CD4-FITC).

☒ The axis scales are clearly visible. Include numbers along axes only for bottom left plot of group (a 'group' is an analysis of identical markers).

☒ All plots are contour plots with outliers or pseudocolor plots.

☒ A numerical value for number of cells or percentage (with statistics) is provided.

## Methodology

| | |
|---|---|
| Sample preparation | To perform ventricle dissociations, hearts were collected and rinsed with ice cold amphibian HBSS (aHBSS, 70%) (Sigma, 55037C). Atria and outflow track were removed from the ventricle with the help of scissors. Using a sterile scalpel, ventricles were minced into smaller pieces and collected in aHBSS in an Eppendorf tube on ice. Tissue pieces were allowed to settle, rinsed once with aHBSS and incubated with 2 mg ml-1 Collagenase/Dispase (Sigma, 10269638001) in aHBSS for 2 hours at 27° C with frequent gentle tapping. After 2 hours, tissue pieces were carefully rinsed with aHBSS, resuspended in aPBS with 10% FBS and mechanically broken with the help of a pipette. Cells were passed through a 100 μm filter and spun down at 300g, 4° C for 5 minutes. The pellets were resuspended in 1 ml aPBS with 1% FBS. Cell viability and number was assessed via automated cell counter (TC20™ Bio-Rad). Cells were stained with Sytox™ Blue dead cell stain (1:1000, S34857, ThermoFisher), Calcein AM (1:250, C1430, ThermoFisher) and Vybrant™ DyeCycle™ Ruby (1:1000, V10273, ThermoFisher) according to the manufacturer's protocols. Cells were sorted on a FACS Aria III system (BD Biosciences) using a 130 μm nozzle to accommodate larger salamander cell size. <br><br> In order to establish a milder dissociation protocol to capture the cells in the injury area at 7 dpci, apical regions of the hearts were removed with a scalpel and minced into smaller pieces that were collected in aHBSS in a flat-bottom borosilicate glass jar. Tissue pieces were rinsed once with cold aHBSS and incubated in digestion buffer containing 1.5 mg ml-1 bovine serum albumin (Sigma, A7906), 3 mg ml-1 glucose (Sigma, G6152), 2 mg ml-1 Collagenase/Dispase (Sigma, 10269638001) in aHBSS. Tubes were shaken in a 27°C waterbath for 30 minutes with 90 rpm shaking. After 30 minutes, solution containing tissue pieces and dissociated cells was gently pipetted up and down without disturbing sizeable tissue pieces and collected in a separate tube with FBS. Fresh digestion buffer was added to the remaining tissue pieces and the procedure was repeated after another 30 minutes. <br><br> To perform CLDN6 antibody staining on isolated cells, 5 ug CLDN6 antibody (abcam, 107059) or rabbit isotype control (abcam, ab171870) was conjugated to Dylight 650 using Dylight 650 Fast Conjugation Kit (abcam, 201803). Cells were incubated with the conjugated antibodies for an hour at 4°C and washed two times with FACS buffer to remove unbound antibodies. Sytox blue and vybrant orange staining was performed subsequently to stain for live, nucleated cells. |
| Instrument | FACS Aria III system (BD Biosciences), 130 um nozzle |
| Software | BD FACSDiva Software |
| Cell population abundance | Cells were collected on miscroscope slides to assess morphology and to make sure no doublets were present. |

Gating strategy

Cells were first gated to exclude debris (FSC-A vs SSC-A plot), then gated to select for singlets (FSC-W vs FSC-A plot). To select for live, metabolically active and nucleated cells: Dead cells were excluded by gating for Sytox Blue negative cells (Sytox Blue vs FSC-A plot). Then metabolically active cells were determined to be Calcein-AM positive (Calcein-AM vs FSC-A plot) and lastly nucleated cells were selected to be Vybrant Ruby positive (Vybrant Ruby vs FSC-A).

Please refer to Extended Data Fig. 5a for a figure exemplifying the gating strategy described above.

☒ Tick this box to confirm that a figure exemplifying the gating strategy is provided in the Supplementary Information.

nature portfolio | reporting summary

March 2021

5