## [Peer Review File · Nature Cell Biology]

Peer Review Information

Journal: Nature Cell Biology

Manuscript Title: Epicardium-derived cells organize through tight junctions to replenish cardiac muscle in salamanders

Corresponding author name(s): Elif Eroglu, András Simon, Kenneth R. Chien

Reviewer Comments & Decisions:

Decision Letter, initial version:
--

Subject: Decision on Nature Cell Biology submission NCB-C46093

Message:

*Please delete the link to your author homepage if you wish to forward this email to co-authors.

Dear Professor Chien,

Your manuscript, "Epicardium-derived cells organize through tight junctions to replenish cardiac muscle in salamanders", has now been seen by 3 referees, who are experts in regeneration in salamander (referee 1) and cardiac reprogramming and heart regeneration (referees 2 and 3). As you will see from their comments (attached below) they find this work of potential interest, but have raised substantial concerns, which in our view would need to be addressed with considerable revisions before we can consider publication in Nature Cell Biology.

Nature Cell Biology editors discuss the referee reports in detail within the editorial team, including the chief editor, to identify key referee points that should be addressed with priority, and requests that are overruled as being beyond the scope of the current study. To guide the scope of the revisions, I have listed these points below. We are committed to providing a fair and constructive peer-review process, so please feel free to contact me if you would like to discuss any of the referee comments further.

In particular, it would be essential to:

a) address the concerns regarding the Tat-Cre, as noted by:

Referee 1:

There is also concern that the method of delivery for Cre is not specific enough to rule out the possibility that some internal (cardiomyocyte or other) cells have taken up the compound and undergone recombination.

Referee 2:

1) A recombinant Tat-Cre fusion protein was used for cell lineage tracing of epicardial cells. While strong nuclear Cre was observed predominantly in epicardial cells 30 hours after injection, there seems to still be Cre protein present later. There is not a clear demonstration of no nuclear Cre expression in myocytes after 30 hours and it seems possible that some of these cells could be recombined without going through the epicardial intermediate at baseline.

2) There are no studies demonstrating the localization of Cre protein expression after injury. Since the Tat-Cre fusion protein is not intrinsically epicardial specific, additional demonstration of its specific localization in the injury setting is needed to support claims of epicardial origin of recombined cardiomyocytes.

Referee 3:

- The authors provide several control experiments to prove that Cre recombination is occurring only in the outer epicardial layer, and that therefore if recombined cells are seen elsewhere they have to be derived from those epicardial cells. However, I would like to see in the discussion section a paragraph of (although unlikely) alternative possibilities: There are reports on cell fusion occurring after heart injury, maybe small amounts of Cre (not to be detected by immunostaining) could pass to cardiomyocytes? e.g. doi: 10.1038/s41467-017-01555-8; doi: 10.1096/fj.201902105R.)

b) demonstrate the toxin-mediated cell depletion is specific to CLDN6+ cells, as noted by referee 1:

In the toxin experiments, the authors have not provided sufficient evidence that other cells might be killed, especially since CLDN6 expression is incompletely characterized.

c) All other referee concerns pertaining to strengthening existing data, providing controls, methodological details, clarifications and textual changes, should also be addressed.

d) Finally please pay close attention to our guidelines on statistical and methodological reporting (listed below) as failure to do so may delay the reconsideration of the revised manuscript. In particular please provide:

We would be happy to consider a revised manuscript that would satisfactorily address these points, unless a similar paper is published elsewhere, or is accepted for publication in Nature Cell Biology in the meantime.

For your information, the original handling editor Christine Weber has sadly left the journal and I will handle the revised manuscript when it is submitted.

- ensure that it conforms to our format instructions and publication policies (see below and <https://www.nature.com/nature/for-authors>).
- provide a point-by-point rebuttal to the full referee reports verbatim, as provided at the end of this letter.
- provide the completed Reporting Summary (found here <https://www.nature.com/documents/nr-reporting-summary.pdf>). This is essential for reconsideration of the manuscript will be available to editors and referees in the event of peer review. For more information see <http://www.nature.com/authors/policies/availability.html> or contact me.

When submitting the revised version of your manuscript, please pay close attention to our [href="https://www.nature.com/nature-research/editorial-policies/image-integrity">Digital Image Integrity Guidelines](https://www.nature.com/nature-research/editorial-policies/image-integrity). and to the following points below:

Nature Cell Biology is committed to improving transparency in authorship. As part of our efforts in this direction, we are now requesting that all authors identified as 'corresponding author' on published papers create and link their Open Researcher and Contributor Identifier (ORCID) with their account on the Manuscript Tracking System (MTS), prior to acceptance. ORCID helps the scientific community achieve unambiguous attribution of all scholarly contributions. You can create and link your ORCID from the home page of the MTS by clicking on 'Modify my Springer Nature account'. For more information please visit www.springernature.com/orcid.

This journal strongly supports public availability of data. Please place the data used in your paper into a public data repository, or alternatively, present the data as Supplementary Information. If data can only be shared on request, please explain why in your Data Availability Statement, and also in the correspondence with your editor. Please note that for some data types, deposition in a public repository is mandatory - more information on our data deposition policies and available repositories appears below.

[REDACTED]

We would like to receive a revised submission within six months.

We hope that you will find our referees' comments, and editorial guidance helpful. Please do not hesitate to contact me if there is anything you would like to discuss.

Best wishes,

Jie Wang

Jie Wang, PhD
Senior Editor
Nature Cell Biology

Tel: +44 (0) 207 843 4924
email: jie.wang@nature.com

Reviewers' Comments:

Reviewer #1:

Remarks to the Author:

A. Summary of the key results

The manuscript entitled “Epicardium-derived cells organize through tight junctions to replenish cardiac muscle in salamanders” by Eroglu et al. describes the contribution of CLDN6(+) epicardium-derived cells in cardiac regeneration. Genetic lineage tracing studies along with single cell transcriptomic analysis revealed that upon injury epicardial cells expressing tight junction protein CLDN6 migrate into the site of injury as an intermediate cell state where they later transdifferentiate into cardiomyocytes. Moreover, pharmacological ablation of CLDN6(+) cells and disruption of tight junctions demonstrated impairment of cardiac regeneration impede cardiac regeneration, respectively, supporting the authors’ findings.

B. Originality and significance

The manuscript is original and shows significant findings in the putative role of the epicardium in cardiac regeneration upon injury as previous studies have not provided evidence in salamanders. This finding would be particularly interesting because in the two main heart regeneration models studied to date—neonatal mouse and zebrafish—pre-existing cardiomyocytes have been demonstrated to be the sole source of new cardiomyocytes during regeneration. This work, properly supported with additional experiments, therefore stands to provide an example from nature of an alternative source of cardiomyocytes for regeneration (albeit still representing perhaps only 15% of cells used to produce cardiomyocytes). This finding could therefore have implications in evolutionary biology, i.e., how regenerative mechanisms are conserved and/or permutated, as well as future implications for regenerative medicine approaches.

C. Data & methodology

The authors utilized immunochemical, genetic, and transcriptomic analyses to demonstrate and validate migration of epicardial-derived cells into the myocardium for regeneration. Although the majority of the data supports their findings, some immunostainings show questionable results relating to when and where the core CLDN6 marker is expressed. There is also concern that the method of delivery for Cre is not specific enough to rule out the possibility that some internal (cardiomyocyte or other) cells have taken up the compound and undergone recombination. In the toxin experiments, the authors have not provided sufficient evidence that other cells might be killed, especially since CLDN6 expression is incompletely characterized. They also present cryoinjury in newt as a method for creating heart damage that leads to regeneration; this method presumably mimics heart injury in humans a bit better than apical resection. While outside the scope of this work, determining whether epicardial cells are one source of cardiomyocytes during regeneration in the resection context would also be interesting.

D. Appropriate use of statistics and treatment of uncertainties

The authors used appropriate statistics for data presentation

E. Conclusions

The manuscript will have significant contribution to the field as extensive work was done to support the finding that epicardial cells transdifferentiate into myocardial cell lineage through an EMT transition for cardiac regeneration. While it is an interesting manuscript, the use of scRNA-seq was somewhat disappointing given the effort and the amount of transcriptomics data which could be acquired to better leverage and define the contribution of epicardial cell in cardiac regeneration. However, there are additional concerns that need to be addressed (see below) to support the main conclusions.

Additionally, CLDN6 appears to have been mostly treated as a marker of a cell population of interest, and one that was druggable as a means of doing cell ablation without genetics. An opportunity to investigate the cell biology of tight junctions and how they mechanistically impact heart regeneration has been missed. If CLDN6 is truly a marker of this cell population (although it's currently unclear if it marks all epicardium or epicardium specifically at the injury site), then its function could be important for regeneration per se. They'd like to make this claim with the c-CPE experiments (which supposedly do not kill CLDN6-expressing cells but instead transiently disrupts CLDNs), but the question remains what is it about tight junctions forming that modulates the next steps of heart regeneration?

F. Suggested general improvements and specific/technical issues:

General set-up:

- Some confusing statements in the intro and later in the results section about the status of an epicardium marker:

- The authors imply that there is no genetic marker for epicardium, but never state this directly -“The innate epicardial fate in mammals following injury has remained unresolved due to the paucity of tools for cell type specific tracing”
- The authors never explicitly state why they pursue a non-genetic marker strategy; this is likely because there is no marker currently available for newts, but the authors should clarify this and their reasoning
- Additional confusion later in Figure 1 when they use pan-ck as an epicardium marker. Upon additional research it became clear that pan-ck is a polyclonal antibody – and therefore unsuitable for lineage tracing, but this is not stated in the text, and will be confusing to anybody who is not already familiar with these reagents.

Figure 1 and related experiments:

- In extended data 1E, the authors show the TAT-Cre diffusing past cells, yet they state it is no longer permeable. This wouldn't be an issue if it was obvious that the Cre only entered the epicardium, but the authors mention the presence of labeled myocardial cells at all time points. They show that the number of these cells increases over time, and also show that TAT-Cre diffuses deeper into myocardial tissue over time.
- The authors state that these observations demonstrate that the epicardium is due to low level homeostatic contribution from the epicardium to the myocardium, but it is unclear how they are able to be absolutely sure that their reporter system is not leaky, and that the low frequency of labeled myocardial cells are not the result of TAT-Cre leaking past the myocardium.
- The authors present decisive visual evidence that the TAT-Cre diffuses deeper into the myocardium over time. They state that this demonstrates the TAT-Cre is no longer active during the time points when it is diffusing into the myocardium. Yet, the data shows that the number of labeled cardiomyocytes increases during this time period. Perhaps the authors have additional data, or reasoning that allows them to authoritatively rule out reporter leakage, but this is not clear in the text as written.
- The relative frequency of labeled epicardium and myocardium strongly suggests that the reporter mostly targets the epicardium, and the lineage tracing suggests that this tissue does contribute to the myocardium during regeneration, but it's not clear if this smaller “homeostatic” contribution is not the result of reporter leakage.
- Panel 1B (MINOR): Declaration of the color scale is a little confusing; it's not immediately obvious that pan-ck is white, and the 'merge' color looks more like the biotin color. Also DAPI is mentioned in the

middle and right images but is not labeled on the merge image even though it is present. The figure would benefit from the authors making it more obvious that pan-ck is white, possibly by placing on a black background, and declare all of the colors present in each column. Eg: [DAPI, pan-ck, biotin MHC]; [DAPI, pan-ck]; [DAPI, biotin], just as they have done with all other panels.

- Panels 1D, 1G, 2C, 2H (MINOR): Larger schematics, particularly the hearts and the colored text that outlines the reporter, would greatly benefit the reader's ability to understand the experimental design.
- The authors show that spontaneous recombination did not occur when vehicle was injected (Extended Data Fig. 1b). However, the data does not provide evidence that YFP(-) cells are Cherry(+). Additional staining of Cherry should be used to support the data.
- Fig. 1f shows YFP stainings of cells undergone epicardial-to-myocardial transition. However, the upper right panel shows lack of YFP staining in the myocardium. Increase in YFP brightness should strengthen the signal. Additionally, DAPI staining in Extended Data Fig. 1e is absent.

Figure 2 and related experiments:

- In Fig. 2a, distinguishing whether the large white spaces in the myocardial space of 7dpci and 14dpci are a part of the necrotic region is difficult. 14dpci looks larger than 7dpci. It is also confusing whether wound size shown in Fig 2a is similar to that of Extended Data Fig. 2a as scale bar is absent in the Extended Data figure.
- Use of 7 dpci is prevalent in regenerative analyses. Including 7dpci staining in Fig. 2 would be valuable.
- Soon after cryoinjury, there is a clear reduction in epicardial cells as shown in Extended Data Fig. 2d and e). However, it is not clear whether the epicardial cells replenish first and contribute to myocardial cell regeneration or whether replenishing and regeneration occurs simultaneously. Providing additional images of these process would further support this study.

Figure 3 and related experiments:

- The authors address that the epicardial-derived myocardial cells expand by utilizing Nucbow reporter animals. Although Extended Data Fig. 3d shows staining of same color clones, additional images of 5- and 6-cell clones, that are cleanly delineated as clones, would further strengthen the data.
- The rationale behind performing scRNA-seq using live cells collected from sham operated and different stages of regenerating heart is confusing. It would be valuable to include scRNA-seq datasets performed separately and compare the alterations in cell clusters as well as gene expression.

- The authors describe three clusters which express embryonic epicardial genes *wt1*, *tcf2* and *tbx15* as potential epicardial/epicardium-derived cell clusters. However, the corresponding reference to these genes are missing. Moreover, after defining the three clusters, the authors do not further discuss the significance in their identification. Performing scRNA-seq separately as commented above might provide further insights. In fact, in Figure 3A-B, the authors never actually mention which three clusters are potentially epicardial, even though they say this in the text. This should be mentioned in the text or the legend or in some visual manner on the figure itself.
 - Fig. 3b shows 15 clusters whereas Extended Fig. 4e display 16 clusters, which is inconsistent.
 - The authors describe that there is a decrease in *cldn6* expression at 7dpci with epicardial thickening shown in Fig. 3f, g and Extended Data Fig. 5e-g. However, Fig. 3f, g does not clearly demonstrate the thickened epicardium. Segmenting between the epicardium and myocardium using dotted line might be helpful.
 - The authors do not explain why they used the marker chosen in Figure 3H.
- Figures 4-5:
- The very top panel of Fig 4d shows inferred localization of CLDN6 and MHC from immunohistochemistry. Wider view of the section is necessary to provide distinction between staining patterns observed at injured compared to uninjured sites. This is important to advance the authors' claims about mechanism and about effects of experimental manipulations. If CLDN6 is a true pan-epicardium reporter, then it should be present in ALL epicardium cells not just at the injury site. If this is the case, the toxin SHOULD cause damage across the whole epicardium, not just the injury site, considering how it is administered.
 - Arrowhead is missing in Extended Data Fig. 5d. Moreover, the histology seems very different than that of Extended Data Fig. 5c. Better quality of images of *cd248* and *cldn6* in situ hybridizations are necessary for clear visualization.
 - The data trends suggest that both epicardium and sup-epicardium are affected beyond the injury site, but this result is deemed insignificant. The authors contend that this is the expected result, when it is unclear to the reader why we should expect the toxin to only have an impact at the injury site. Without this clarification, this result seems to undermine the author's claim that CLDN6 is a true epicardium marker – unless the increased epicardial and sub-epicardial cell death outside of the injury zone is in-fact significant. These issues raise questions without explanations. One possibility is that everything is boosted at the injury site anyway, and the error bars are bigger; the second possibility is that there is a jump in the epicardium and the subepicardium, which is deemed insignificant even though these distributions are much tighter. Given these results, why do the authors try to make the hard claim that

the effect is restricted to the epicardium at the injury site? A more accurate interpretation of their results should be presented.

Extended figures:

- The authors demonstrate that treatment with wt-CPE showed increased TUNEL+ cells in injury area effecting epicardium-derived CLDN6+ cells. However, Extended Data Fig. 7a and b does not show CLDN6 and TUNEL co-staining which is invaluable for supporting the reduction of CHERRY+ epicardium-derived cells in response to wt-CPE.
- Extended Data Fig. 8c is confusing and unclear. According to color scheme, Cluster 6 which are T-cells show expression of muscle genes including MYL3, MYL4, and TCNN1 with pseudotime. Further clarification is required for pseudotime data analysis.
- The authors identified that during cardiac homeostasis, low levels of epicardial cell conversion into myocardiocytes. It is mentioned in the discussion that epicardial cells contribute to different cell lineages including fibroblasts, perivascular cells, smooth muscle cells, and adipocytes. Thus, further discussion is necessary in regards to whether myocardial progenitor cells function as multipotent cells.

Supplemental videos:

- The differences between conditions in the supplemental videos are not readily apparent to the untrained eye. This could be rectified by visually highlighting and emphasizing the relevant features. Perhaps they could present a repeat of the content of each video twice, once with annotations and once without. A supplemental figure with annotated static images to complement the videos may also suffice.

DISCUSSION:

- The authors reiterate that epicardial cells support regeneration in many ways unrelated to their proliferation. Actually, this might undermine one of their conclusions, because the toxin could also be disrupting these other processes, not just direct cellular contributions. They should definitely acknowledge this possibility, especially for the manipulations where they actually kill the cells. They should also acknowledge the limitations of the study.

G. References

Appropriate references were used.

H. Clarity and context

Although some parts of the manuscript were confusing, its overall message was clear.

Reviewer #2:

Remarks to the Author:

The study by Eroglu et al examines epicardial activation and cardiac regenerative mechanisms in the salamander *Pleurodeles waltl*. The authors employ novel and innovative approaches to trace cell lineages and perform gene expression profiling at single cell resolution after cardiac injury. Derivation of a small population of cardiac myocytes from epicardially labeled cells was observed at homeostasis and after injury. In addition Claudin-6 and tight junctions of epicardial cells were found to be important in the regenerative response.

While there is strong novelty and potential interest in these mechanistic studies of salamander heart regeneration, there are some limitations in the approach and interpretations that are narrowly focused on epicardial cell conversion into cardiomyocytes.

Comments.

1) A recombinant Tat-Cre fusion protein was used for cell lineage tracing of epicardial cells. While strong nuclear Cre was observed predominantly in epicardial cells 30 hours after injection, there seems to still be Cre protein present later. There is not a clear demonstration of no nuclear Cre expression in myocytes after 30 hours and it seems possible that some of these cells could be recombined without going through the epicardial intermediate at baseline.

2) There are no studies demonstrating the localization of Cre protein expression after injury. Since the Tat-Cre fusion protein is not intrinsically epicardial specific, additional demonstration of its specific localization in the injury setting is needed to support claims of epicardial origin of recombined cardiomyocytes.

3) Were other cell types derived from epicardial recombined lineages observed? Under homeostatic conditions, were cardiac fibroblasts found to arise from the epicardial lineage? Additional information is needed on the epicardial-derived lineages after injury and in the NucBow experiments.

4) What is the identity of the “honeycomb” like cells in the subepicardial injury area? Are they endothelial? Do they express specific ECM proteins? Are they part of the scar?

5) A cluster of cells expressing *gata4/6* and *Foxc1/2* is identified as a potential source of cardiomyocytes. However, these markers are also expressed in endothelial cell lineages, and more definitive markers of

cardiomyocytes (Nkx2.5, Mef2, Tbx5) were not observed in this cluster. Potential endothelial fates of this population should be considered.

6) It is almost certain that the impaired regenerative response in the CPE injected hearts is due to more than the loss of the small population of epicardial-derived cardiomyocytes. Vascularization and fibroblast contributions to the scar ECM and resolution also should be monitored.

7) The observed epicardial-derived cardiomyocyte population is a small minority of new myocytes. Are these cells fundamentally different from the majority of new cardiomyocytes after regeneration? Also is there evidence that the majority of new cardiomyocytes come from existing cardiomyocytes?

Minor comments:

1) While some investigators have reported conversion of epicardial cells to cardiomyocytes in mammals, this area is still controversial and I do not think that the field has reached a consensus as stated in line 54 of the introduction.

Reviewer #3:

Remarks to the Author:

This manuscript described the differentiation of epicardial cells towards cardiomyocytes in the regenerating salamander species *Pleurodeles waltl*.

While humans and most mammals cannot efficiently regenerate cardiac muscle after injury, other vertebrate species have been described with a high regenerative capacity and the ability to replace lost cardiomyocytes with new myocardium. The cellular mechanism underlying this regenerative capacity seems to be predominantly in the ability of cardiomyocytes from the adult heart to reenter cell cycle, divide and give rise to daughter cells that then replenish the lost myocardium.

Here the authors describe an alternative mechanism of myocardium regeneration in the newt *Pleurodeles waltl*, in which cells within the epicardial layer divide and give rise to cardiomyocytes. This process has also been described previously to occur upon Thymosin beta 4 administration in the mouse (doi: 10.1038/nature05383), but the article has been widely disputed in the field. Finding a similar mechanism occurring naturally in an animal species of high regenerative capacity is therefore very surprising and will be of great interest to the community.

To identify conversion of epicardial cells into cardiomyocytes the authors use a nuclear-tagged Cre protein into the pericardial cavity. They show several control experiments to rule out leakiness of the flexed reporter line and unwanted Cre uptake of non-epicardial cells. They find that some epicardial cells give rise to underlying cardiomyocytes. This occurs during homeostasis but is enhanced upon ventricular cryoinjury. They also perform a rigorous quantification of this event and add multicolor fate mapping to study clonality.

They next try to understand more on epicardial gene signature and genes involved in this process. For this they make use of scRNAseq and identify several cell types in the newt heart, describing also new markers for epicardial cells. They identify Claudin6 as a new epicardial cell marker. Using a specific toxin (CPE) blocking claudin6 they could show that this protein is not only an epicardial marker but is also playing an important role in allowing the conversion of epicardial cells to cardiomyocytes and supporting heart regeneration. They confirm that the role of this protein is related to the establishment of tight junctions through experiments using toxin variants. Methodology is sound and of high-standard: I particularly like the elegant methods to assess amount of cryoinjured area through longitudinal ultrasound measurements.

The data are well-presented and conclusions based on robust data.

Suggested improvements:

Introduction:

- I would recommend also citing the following paper when talking about previous results studying the fate of epicardial cells during heart regeneration in the zebrafish *Dev Biol.* 2012 Oct 15;370(2):173-86. doi: 10.1016/j.ydbio.2012.07.007, since here authors did not solely rely on genetic lineage tracing but included transplantations assays.

Results:

- I found it surprising that the authors observe a decrease in epicardial cell numbers labelled in response to injury. What has been reported in other species is that epicardial cells become very proliferative in response to cardiac injury. Is the explanation that they leave the epicardium and contribute to cardiomyocytes in the newt? Please provide some discussion.
- The authors provide several control experiments to prove that Cre recombination is occurring only in the outer epicardial layer, and that therefore if recombined cells are seen elsewhere they have to be derived from those epicardial cells. However, I would like to see in the discussion section a paragraph of (although unlikely) alternative possibilities: There are reports on cell fusion occurring after heart injury, maybe small amounts of Cre (not to be detected by immunostaining) could pass to cardiomyocytes? e.g. doi: 10.1038/s41467-017-01555-8; doi: 10.1096/fj.201902105R.)

- Authors show how many epicardial-derived cardiomyocytes are PCNA-positive but: how many non-epicardial-derived cardiomyocytes are PCNA positive? Understanding the ration would be interesting. Also, is BrdU labelling possible in the newt? I would prefer to see this kind of labelling as PCNA also labels senescent cells “stuck in the cell cycle”.
- Statement on line 265. “show that the impaired regeneration caused by the reduction of epicardium-derived cardiomyocytes”. This needs to be reformulated. The authors show that regeneration is incomplete upon wt-CPE treatment and that there is less number of Cherry1 cardiomyocytes. This shows that both effects of wt-CPE correlate but not necessarily that the sole reason that the heart does not regeneration is due to the lack of cherry+ cells. Indeed the authors discuss this later on in lines 349-350.
- Figure 6: For images shown in panel h or single channels need to be shown. I could not see well the myl3;gata6 stained Cherry cells.
- Figure 7: CPE bind to Claudin6. Can the authors show a co-labelling of CPE with Claudin 6 for example in Figure 7a?
- Extended Data Fig 2: In panel b show also still images without the yellow highlight. In the legends authors refer to data not shown in epicardial cells proliferation at 48 hpci. Is it allowed to refer to non-shown data?? I strongly recommend to show all data.

REFERENCES – are limited to a total of 70 for Articles, Resources, Technical Reports; and 40 for Letters. This includes references in the main text and Methods combined. References must be numbered sequentially as they appear in the main text, tables and figure legends and Methods and must follow the precise style of Nature Cell Biology references. References only cited in the Methods should be numbered consecutively following the last reference cited in the main text. References only associated with Supplementary Information (e.g. in supplementary legends) do not count toward the total reference limit and do not need to be cited in numerical continuity with references in the main text.

Only published papers can be cited, and each publication cited should be included in the numbered reference list, which should include the manuscript titles. Footnotes are not permitted.

Methods should be written concisely, but should contain all elements necessary to allow interpretation and replication of the results. As a guideline, Methods sections typically do not exceed 3,000 words. The Methods should be divided into subsections listing reagents and techniques. When citing previous methods, accurate references should be provided and any alterations should be noted. Information must be provided about: antibody dilutions, company names, catalogue numbers and clone numbers for monoclonal antibodies; sequences of RNAi and cDNA probes/primers or company names and catalogue numbers if reagents are commercial; cell line names, sources and information on cell line identity and authentication. Animal studies and experiments involving human subjects must be reported in detail, identifying the committees approving the protocols. For studies involving human subjects/samples, a statement must be included confirming that informed consent was obtained. Statistical analyses and information on the reproducibility of experimental results should be provided in a section titled “Statistics and Reproducibility”.

All Nature Cell Biology manuscripts submitted on or after March 21 2016 must include a Data availability statement as a separate section after Methods but before references, under the heading “Data Availability”. . For Springer Nature policies on data availability see <http://www.nature.com/authors/policies/availability.html>; for more information on this particular policy see <http://www.nature.com/authors/policies/data/data-availability-statements-data-citations.pdf>. The Data availability statement should include:

- Accession codes for primary datasets (generated during the study under consideration and designated as “primary accessions”) and secondary datasets (published datasets reanalysed during the study under consideration, designated as “referenced accessions”). For primary accessions data should be made public to coincide with publication of the manuscript. A list of data types for which submission to community-endorsed public repositories is mandated (including sequence, structure, microarray, deep sequencing data) can be found here <http://www.nature.com/authors/policies/availability.html#data>.
- Unique identifiers (accession codes, DOIs or other unique persistent identifier) and hyperlinks for datasets deposited in an approved repository, but for which data deposition is not mandated (see here for details <http://www.nature.com/sdata/data-policies/repositories>).

- At a minimum, please include a statement confirming that all relevant data are available from the authors, and/or are included with the manuscript (e.g. as source data or supplementary information), listing which data are included (e.g. by figure panels and data types) and mentioning any restrictions on availability.
- If a dataset has a Digital Object Identifier (DOI) as its unique identifier, we strongly encourage including this in the Reference list and citing the dataset in the Methods.

We recommend that you upload the step-by-step protocols used in this manuscript to the Protocol Exchange. More details can found at www.nature.com/protocolexchange/about.

All imaging data should be accompanied by scale bars, which should be defined in the legend. Cropped images of gels/blots are acceptable, but need to be accompanied by size markers, and to retain visible background signal within the linear range (i.e. should not be saturated). The boundaries of panels with low background have to be demarked with black lines. Splicing of panels should only be considered if unavoidable, and must be clearly marked on the figure, and noted in the legend with a statement on whether the samples were obtained and processed simultaneously. Quantitative comparisons between samples on different gels/blots are discouraged; if this is unavoidable, it should only be performed for samples derived from the same experiment with gels/blots were processed in parallel, which needs to be stated in the legend.

Figures should be provided at approximately the size that they are to be printed at (single column is 86 mm, double column is 170 mm) and should not exceed an A4 page (8.5 x 11"). Reduction to the scale that will be used on the page is not necessary, but multi-panel figures should be sized so that the whole figure can be reduced by the same amount at the smallest size at which essential details in each panel are visible. In the interest of our colour-blind readers we ask that you avoid using red and green for contrast in figures. Replacing red with magenta and green with turquoise are two possible colour-safe alternatives. Lines with widths of less than 1 point should be avoided. Sans serif typefaces, such as

Helvetica (preferred) or Arial should be used. All text that forms part of a figure should be rewritable and removable.

The total number of Supplementary Figures (not including the “unprocessed scans” Supplementary Figure) should not exceed the number of main display items (figures and/or tables (see our Guide to Authors and March 2012 editorial <http://www.nature.com/ncb/authors/submit/index.html#suppinfo>; <http://www.nature.com/ncb/journal/v14/n3/index.html#ed>). No restrictions apply to Supplementary Tables or Videos, but we advise authors to be selective in including supplemental data.

GUIDELINES FOR EXPERIMENTAL AND STATISTICAL REPORTING

REPORTING REQUIREMENTS – We are trying to improve the quality of methods and statistics reporting in our papers. To that end, we are now asking authors to complete a reporting summary that collects information on experimental design and reagents. The Reporting Summary can be found here <https://www.nature.com/documents/nr-reporting-summary.pdf> If you would like to reference the guidance text as you complete the template, please access these flattened versions at <http://www.nature.com/authors/policies/availability.html>.

We strongly recommend the presentation of source data for graphical and statistical analyses as a separate Supplementary Table, and request that source data for all independent repeats are provided when representative experiments of multiple independent repeats, or averages of two independent experiments are presented. This supplementary table should be in Excel format, with data for different figures provided as different sheets within a single Excel file. It should be labelled and numbered as one of the supplementary tables, titled “Statistics Source Data”, and mentioned in all relevant figure legends.

Author Rebuttal to Initial comments

Point-by-Point response to the reviewers' comments:

We thank the reviewers for their positive and insightful comments, which helped us to significantly improve the manuscript. Below, please find a detailed point-to-point reply to every issue raised.

Reviewer #1:

Remarks to the Author:

A. Summary of the key results

The manuscript entitled “Epicardium-derived cells organize through tight junctions to replenish cardiac muscle in salamanders” by Eroglu et al. describes the contribution of CLDN6(+) epicardium-derived cells in cardiac regeneration. Genetic lineage tracing studies along with single cell transcriptomic analysis revealed that upon injury epicardial cells expressing tight junction protein CLDN6 migrate into the site of injury as an intermediate cell state where they later transdifferentiate into cardiomyocytes. Moreover, pharmacological ablation of CLDN6(+) cells and disruption of tight junctions demonstrated impairment of cardiac regeneration impede cardiac regeneration, respectively, supporting the authors’ findings.

B. Originality and significance

The manuscript is original and shows significant findings in the putative role of the epicardium in cardiac regeneration upon injury as previous studies have not provided evidence in salamanders. This finding would be particularly interesting because in the two main heart regeneration models studied to date—neonatal mouse and zebrafish—pre-existing cardiomyocytes have been demonstrated to be the sole source of new cardiomyocytes during regeneration. This work, properly supported with additional experiments, therefore stands to provide an example from nature of an alternative source of cardiomyocytes for regeneration (albeit still representing perhaps only 15% of cells used to produce

cardiomyocytes). This finding could therefore have implications in evolutionary biology, i.e., how regenerative mechanisms are conserved and/or permutated, as well as future implications for regenerative medicine approaches.

C. Data & methodology

The authors utilized immunochemical, genetic, and transcriptomic analyses to demonstrate and validate migration of epicardial-derived cells into the myocardium for regeneration. Although the majority of the data supports their findings, some immunostainings show questionable results relating to when and where the core CLDN6 marker is expressed. There is also concern that the method of delivery for Cre is not specific enough to rule out the possibility that some internal (cardiomyocyte or other) cells have taken up the compound and undergone recombination. In the toxin experiments, the authors have not provided sufficient evidence that other cells might be killed, especially since CLDN6 expression is incompletely characterized. They also present cryoinjury in newt as a method for creating heart damage that leads to regeneration; this method presumably mimics heart injury in humans a bit better than apical resection. While outside the scope of this work, determining whether epicardial cells are one source of cardiomyocytes during regeneration in the resection context would also be interesting.

D. Appropriate use of statistics and treatment of

uncertaintiesThe authors used appropriate statistics for

data presentation

E. Conclusions

The manuscript will have significant contribution to the field as extensive work was done to support the

finding that epicardial cells transdifferentiate into myocardial cell lineage through an EMT transition for cardiac regeneration. While it is an interesting manuscript, the use of scRNA-seq was somewhat disappointing given the effort and the amount of transcriptomics data which could be acquired to better leverage and define the contribution of epicardial cell in cardiac regeneration. However, there are additional concerns that need to be addressed (see below) to support the main conclusions. Additionally, CLDN6 appears to have been mostly treated as a marker of a cell population of interest, and one that was druggable as a means of doing cell ablation without genetics. An opportunity to investigate the cell biology of tight junctions and how they mechanistically impact heart regeneration has been missed. If CLDN6 is truly a marker of this cell population (although it's currently unclear if it marks all epicardium or epicardium specifically at the injury site), then its function could be important for regeneration per se. They'd like to make this claim with the c-CPE experiments (which supposedly do not kill CLDN6-expressing cells but instead transiently disrupts CLDNs), but the question remains what is it about tight junctions forming that modulates the next steps of heart regeneration?

We thank the reviewer for the positive comments on the importance of our findings and for raising important concerns, that we have addressed in a point-by-point reply as outlined below.

F. Suggested general improvements and specific/technical issues:

General set-up:

- *Some confusing statements in the intro and later in the results section about the status of an epicardium marker:*
- *The authors imply that there is no genetic marker for epicardium, but never state this directly - "The innate epicardial fate in mammals following injury has remained unresolved due to the paucity of*

tools for cell type specific tracing”.

- *The authors never explicitly state why they pursue a non-genetic marker strategy; this is likely because there is no marker currently available for newts, but the authors should clarify this and their reasoning.*

- *Additional confusion later in Figure 1 when they use pan-ck as an epicardium marker. Upon additional research it became clear that pan-ck is a polyclonal antibody – and therefore unsuitable for lineage tracing, but this is not stated in the text, and will be confusing to anybody who is not already familiar with these reagents.*

In the revised manuscript, we made the following changes to clarify the current state of epicardial markers and choice of tools to study epicardium in salamanders.

1) We modified the following sentence in the introduction as follows (new text in bold). "The innate epicardial fate in mammals following injury has remained unresolved due to the paucity of tools for cell type specific tracing, **as current genetic markers also label epicardial derivatives and/or non-epicardial cell types**^{23–26}." (Line 53-54)

2) We modified the following sentence to the introduction as follows (new text in bold): "However, the role of epicardium has not been investigated further than its upregulation of regeneration specific-matrix proteins and **genetic markers of the salamander epicardium have not been identified**³²." (Line 62)

3) We added the following information regarding the use of pan-cytokeratin as an epicardium marker "... (pan-CK)-positive epicardial cell layer **detected with a polyclonal antibody that recognizes a broad spectrum of keratins, ...**". (Line 88)

Figure 1 and related experiments:

- *In extended data 1E, the authors show the TAT-Cre diffusing past cells, yet they state it is no longer permeable. This wouldn't be an issue if it was obvious that the Cre only entered the epicardium, but the authors mention the presence of labeled myocardial cells at all time points. They show that the number of these cells increases over time, and also show that TAT-Cre diffuses deeper into myocardial tissue over time.*
- *The authors state that these observations demonstrate that the epicardium is due to low level homeostatic contribution from the epicardium to the myocardium, but it is unclear how they are able to be absolutely sure that their reporter system is not leaky, and that the low frequency of labeled myocardial cells are not the result of TAT-Cre leaking past the myocardium.*
- *The authors present decisive visual evidence that the TAT-Cre diffuses deeper into the myocardium over time. They state that this demonstrates the TAT-Cre is no longer active during the time points when it is diffusing into the myocardium. Yet, the data shows that the number of labeled cardiomyocytes increases during this time period. Perhaps the authors have additional data, or reasoning that allows them to authoritatively rule out reporter leakage, but this is not clear in the text as written.*
- *The relative frequency of labeled epicardium and myocardium strongly suggests that the reporter mostly targets the epicardium, and the lineage tracing suggests that this tissue does contribute to the myocardium during regeneration, but it's not clear if this smaller "homeostatic" contribution is not the result of reporter leakage.*

We agree with the reviewer that the data on epicardium specific Cre-mediated recombination is a critical

point. We have therefore performed new experiments and provided additional explanations to data presented in the original version of the manuscript. In the revised version, we clarify the distinction between nuclear localization and extracellular localization of Cre. Importantly, nuclear Cre is only found in the epicardial cells while Cre is exclusively found in the extracellular space in the myocardium. This is in line with previous observations of TAT-eGFP fusion protein binding to the extracellular matrix upon injection into muscle (Caron et al., 2001, Molecular Therapy). In fact, the transduction domain of the TAT protein is a heparin binding protein that interacts with heparan sulfate, proteoglycans of the cell surface and extracellular matrix (Rusnati et al., 1998, J. Biol. Chem). This affinity between the transduction domain of TAT and extracellular matrix surrounding the muscles interferes with the intracellular transduction process (Caron et al., 2001, Molecular Therapy). The following three experiments in the revised version further shows that nuclear Cre is exclusive to the epicardium. 1) Quantification of the Cre signal in the epicardial and myocardial cell nuclei at multiple time-points, (2) Co-immunostaining of Cre with wheat germ agglutinin (WGA) that marks the myocardial cell membranes and extracellular matrix, (3) Utilization of CPE to ablate TAT-Cre labelled epicardial cells under homeostasis to assess the impact on number of labelled cardiomyocytes.

- (1) Quantification of Cre signal in the epicardial and myocardial cell nuclei at multiple time-points.

Automated image quantification using ImageJ showed that at 30-hours post TAT-Cre injection (30-hpi) 25% of epicardial cell nuclei were Cre positive. Percentage of Cre⁺ epicardial cell nuclei decreased to 6.32% and 0.64% at 40- and 96-hpi respectively. In stark contrast to the epicardial nuclei, we did not detect any Cre expressing nuclei in the myocardium among the 7134 cell nuclei we analyzed (1569 nuclei at 30-hpi, 1719 nuclei at 40-hpi, 3846 nuclei at 96-hpi) (**Extended Data Fig. 1b**). These findings are in line with qualitative observations we made in the original manuscript and complement those quantitatively to show that TAT-Cre does not transduce myocardial cells.

- (2) Co-immunostaining for Cre and wheat germ agglutinin (WGA) that marks the myocardial cell

membranes and extracellular matrix. Co-immunostaining of Cre with wheat germ agglutinin at 96-hpi confirmed the localization of Cre to the extracellular matrix surrounding the cardiac muscle

(Extended Data Fig. 2d).

(3) CPE-mediated ablation of Tat-Cre labelled epicardial cells under homeostasis. To further test whether epicardial cells are indeed a source of labelled cardiomyocytes under homeostasis, we performed a CPE-mediated loss-of-function experiment. As a first step, we extended our understanding of the cellular target of the CPE, a point also raised by the reviewer. In the original manuscript, we reported the use of CPE to ablate epicardium-derived intermediate cells in the injury area (**Fig. 5c and Extended Data Fig. 9a, b**). We now provide additional data showing that epicardium is also a target of CPE toxin, but only at a much higher dose (**Extended Data Fig. 9c**). While we cannot fully explain the dose dependent sensitivity of epicardium versus epicardium-derived intermediate cells, we provide data describing reduced CLDN6 expression in post-injury epicardium (**Fig. 4a, c**), that might be responsible for reduced sensitivity to the toxin. Critically, the effect of CPE is confined to the epicardium and its immediate derivatives, and we did not observe any myocyte death at any dose we injected (**Fig. 5c and Extended Data Fig. 9a-c**). While high dose CPE was effective at ablating the epicardium, animals did not tolerate intraperitoneal injection. Therefore, we implemented microinjection of the toxin into the pericardial cavity for the experiment described below that further proves that Cre-mediated recombination is restricted to the epicardium.

(4) First, we labelled epicardial cells of $Pw^{GFP-loxP-Cherry}$ reporter animals with TAT-Cre recombinase. Second, 40 hours post TAT-Cre injection, we microinjected 200 ng of wt-CPE to the pericardial cavity (**Extended Data Fig. 10a**). This resulted in a ~5.4-fold reduction of CHERRY⁺ epicardial cells as assessed

4 days and 8 hours later (6 days post TAT-Cre injection). In conjunction with this reduction, we observed a ~4-fold decrease in the number of labelled cardiomyocytes, confirming that epicardium gives rise to cardiomyocytes under homeostasis (**Extended Data Fig. 10b-e**).

Taken together: TAT-Cre transduces epicardial cells with decreasing efficiency over the course of 96 hours. We find no indication of myocardial presence of nuclear Cre signal. Lastly, depleting the labelled epicardial cells results in significantly less labelled myocytes. We believe these additional data provide sufficient additional support for the conclusion that epicardial cells give rise to cardiomyocytes.

These data have been incorporated in **Extended Data Fig. 1, 2, 9 and 10** and are presented in the results part of the manuscript. We also added the following paragraph to the Discussion:

“The specificity of the Cre-mediated recombination is a key element of the present study for which we present two major lines of evidence. First, nuclear Cre is detected only in epicardium. Small amount of Cre protein diffusing into the myocardium is sequestered in the extracellular matrix, which precludes transduction of non-epicardial cells (**Extended Data Fig. 2d**)³⁶. Second, by functional assays we show that partial ablation of lineage labelled epicardial cells reduces the number of cardiomyocytes carrying the label both under homeostasis, and during regeneration (**Fig. 5l, m and Extended Data Fig. 10a-e**). It has been reported in other species that transient homotypic or heterotypic cell fusion could trigger cell cycle re-entry of cardiomyocytes^{68,69}. Theoretically, contribution from such a mechanism cannot fully be discounted in salamanders either. However, the fact that regeneration is inhibited by the non-cytotoxic c- CPE in the absence of an effect on cardiomyocyte proliferation further supports the model that the cellular contribution by the epicardium rather than cell fusion between epicardial cells and myocytes is essential for cardiac regeneration in salamanders (**Fig. 7h, i**).” (**Page 16, 17**)

- **Panel 1B (MINOR):** Declaration of the color scale is a little confusing; it's not immediately obvious that pan-ck is white, and the 'merge' color looks more like the biotin color. Also, DAPI is mentioned in the middle and right images but is not labeled on the merge image even though it is present. The figure would benefit from the authors making it more obvious that pan-ck is white, possibly by placing on a black background, and declare all of the colors present in each column. Eg: [DAPI, pan-ck, biotin MHC]; [DAPI, pan-ck]; [DAPI, biotin], just as they have done with all other panels.

Following the reviewer's suggestion, we have placed a black background behind all white labels and declared all of the colors present in Fig.1b.

- **Panels 1D, 1G, 2C, 2H (MINOR):** Larger schematics, particularly the hearts and the colored text that outlines the reporter, would greatly benefit the reader's ability to understand the experimental design. Following the reviewer's suggestion, we now enlarged the drawings and the text size to improve reader's ability to understand the experimental design.

- **The authors show that spontaneous recombination did not occur when vehicle was injected (Extended Data Fig. 1b). However, the data does not provide evidence that YFP (-) cells are Cherry (+). Additional staining of Cherry should be used to support the data.**

Following the reviewer's suggestion, we did immunostaining against CHERRY to show that YFP⁺ epicardial cells are CHERRY⁺ (Extended Data Fig. 2c). While one can see CHERRY signal (especially the dotted appearance), we would like to point out that the expression level of the transgene is much higher in cardiomyocytes which makes it difficult to appreciate the epicardial CHERRY signal.

- **Fig. 1f shows YFP stainings of cells undergone epicardial-to-myocardial transition. However, the upper right panel shows lack of YFP staining in the myocardium. Increase in YFP brightness should**

strengthen the signal. Additionally, DAPI staining in Extended Data Fig. 1e is absent.

In the revised version of the manuscript, we increased the YFP brightness in **Fig. 1f** (top right panel) and corrected the label in Extended Data Fig. 1e, which is now **Extended Data Fig. 2e**.

• In Fig. 2a, distinguishing whether the large white spaces in the myocardial space of 7dpci and 14dpci are a part of the necrotic region is difficult. 14dpci looks larger than 7dpci.

We thank the reviewer for pointing out the difficulty in distinguishing injured versus uninjured areas of the heart. In the revised version of the manuscript, we inserted dashed lines to clearly separate the injured area from the uninjured area for each time-point, which should also help to highlight the reduced injury size at 14-dpci compared to 7-dpci.

It is also confusing whether wound size shown in Fig. 2a is similar to that of Extended Data Fig. 2a as scale bar is absent in the Extended Data figure.

We thank the reviewer for pointing this out and added the scale bar information to the Extended Data Fig. 2a (in the revised manuscript **Extended Data Fig. 3a**) legend.

• Use of 7 dpci is prevalent in regenerative analyses. Including 7dpci staining in Fig. 2 would be valuable.

We agree with the reviewer's observation that 7-dpci is a key time-point for the events described in the manuscript and immunostainings of multiple targets at this time-point appears in a multitude of figures (**Fig. 3f, g, Fig. 4c, f, Fig. 5e and Extended Data Fig. 9a, d**). In the revised manuscript, we label these panels more clearly as 7-dpci.

• Soon after cryoinjury, there is a clear reduction in epicardial cells as shown in Extended Data Fig. 2d and e). However, it is not clear whether the epicardial cells replenish first and contribute to myocardial cell regeneration or whether replenishing and regeneration occurs simultaneously.

Providing additional images of these process would further support this study.

In the revised manuscript, we show that epicardial cells start proliferating around 3-dpci (**Extended Data Fig. 4a**). Assessed by pan-CK expression at 7-dpci, epicardium is still not fully recovered (**Extended Data Fig. 9a**), suggesting that epicardial replenishment and myocyte regeneration temporarily overlap with each other. Follow up studies will be necessary for describing cellular dynamics in detail.

The authors address that the epicardial-derived myocardial cells expand by utilizing Nucbow reporter animals. Although Extended Data Fig. 3d shows staining of same color clones, additional images of 5- and 6-cell clones, that are cleanly delineated as clones, would further strengthen the data.

We agree with the reviewer that showing larger clones would strengthen the data. Therefore, in the revised version of the manuscript we show a 5-cell clone that is mEYFP⁺ mCerulean⁺ MHC⁺ (**Extended Data Fig.5e, f**).

• The rationale behind performing scRNA-seq using live cells collected from sham operated and differentstages of regenerating heart is confusing. It would be valuable to include scRNA-seq datasets performedseparately and compare the alterations in cell clusters as well as gene expression.

We thank the reviewer for suggesting to extend our analysis of the scRNA-seq data with an emphasis on changes occurring during regeneration, which helped us to significantly improve the value of this dataset for the manuscript. In the dataset, cells at each time-point were collected and sequenced separately and we apologize that this was not readily perceivable from the original presentation of the data. To improve this, we now provide UMAP plots in **Extended Data Fig. 6e** showing cellular contributions separately for each time-point, which allows appreciation of the dynamic changes in cluster contributions across time. As a direct effect of the injury, endothelial/endocardial cells, myocyte-like cells and epicardial cells were

only rarely recovered from 7-dpci hearts. These cell types were recovered in increasing numbers again at 14-dpci and 28-dpci indicating ongoing regeneration. As expected, immune cells, such as B cells, macrophages and neutrophils constitute most of the cells recovered at 7-dpci. In this representation of the data, we noticed a small cluster (16 cells) which was only recovered from 7- and 14-dpci time-points. In our previous analysis, these cells clustered together with cardiomyocytes. Upon closer look, we did not find expression of muscle genes in these cells but observed expression of epithelial-to-mesenchymal transition (EMT) markers. With minor adjustment of the clustering parameters (decrease of the k-parameter from 10 to 8 in the nearest-neighbor graph construction) we obtained a more accurate separation of the cells and annotated them as “transitioning cells”. We also incorporated additional EMT markers (Fgfr2, Vim, Snai1, Ncam1, Msx1), into the heatmap to accommodate for this cell cluster (Fig. 3d).

Notably, 3 of these cells express *Cldn6* at a low level, potentially suggesting that these cells might be epicardial cells going through EMT. In addition, these cells show proximity to the epicardial cell cluster in alternative low-dimensionality embeddings hinting towards a potential similarity between these two clusters (data not shown). However, due to the low number of cells detected and the preliminary nature of this finding, we decided against putting emphasis on this at this point in the manuscript.

The new results are shown in revised Fig. 3, Extended Data 6 and 7 and the accompanying changes have been made in the results section. (Page 7-9)

- *The authors describe three clusters which express embryonic epicardial genes wt1, tcf21 and tbx15 as potential epicardial/epicardium-derived cell clusters. However, the corresponding reference to these genes are missing. Moreover, after defining the three clusters, the authors do not further*

discuss the significance of their identification. Performing scRNA-seq separately as commented above might provide further insights. In fact, in Figure 3A-B, the authors never actually mention which three clusters are potentially epicardial, even though they say this in the text. This should be mentioned in the text or the legend or in some visual manner on the figure itself.

We apologize for not including the references to these genes. In the revised manuscript we cite the original literature describing their identification and expression (Robb et al., 1998, Dev Dyn; Moore et al., 1999, Development; Kraus et al., 2001, Mech Dev.). Regarding the two other cell clusters expressing these genes (Cluster 9 and 12), as mentioned earlier these markers are not specific to the epicardium. In fact, their expression in several cell clusters justifies our alternative, non-genetic lineage tracing approach in the adult salamander heart, as well as highlighting the importance of the identification of CLDN6 as a more specific marker. We utilized expression of *Wt1*, *Tcf21* and *Tbx15* in the initial screening and annotation of the cell clusters. We now revised the results section to convey this message and keep the emphasis on bona-fide epicardial cells.

• **Fig. 3b shows 15 clusters whereas Extended Fig. 4e display 16 clusters, which is inconsistent.**

We apologize for the confusing presentation of this piece of data. **Extended Data Fig.4e** showed the original unsupervised clustering with 16 clusters where endothelial/endocardial cells were separated into two small subclusters with no distinct biological meaning in the context of this manuscript. Therefore, in **Fig. 3b** these clusters were manually combined and this was stated in the figure legend as “Note that endothelial cell clusters have been merged for simplicity, see **Extended Data Fig.4e** for original clustering.” However, we understand that this could be confusing for the readers. Therefore, in the revised version of the manuscript we omitted the manual combination of clusters and removed the

UMAP in question.

- *The authors describe that there is a decrease in *cldn6* expression at 7dpi with epicardial thickening shown in Fig. 3f, g and Extended Data Fig. 5e-g. However, Fig. 3f, g does not clearly demonstrate the thickened epicardium. Segmenting between the epicardium and myocardium using dotted line might be helpful.*

Following the reviewer's suggestion, we now included a dotted line in **Fig. 3f** and **g** for the panels where this was possible. However, we would like to acknowledge that it is not trivial to segment the two layers in the *in-situ* hybridization images in the absence of a myocardial probe and we refrained from doing so where it was not clear.

- *The authors do not explain why they used the marker chosen in Figure 3H.*

We would like to thank the reviewer for asking us to clarify our choice of focusing on *Cldn6* as a novel epicardial marker. We now add the following sentence to the revised manuscript "***Cldn6* was chosen as anovel epicardial marker in the subsequent studies as it showed the highest expression in epicardial cellsamong the tight junction genes (Fig. 3e).**" (Line 201-203)

- *The very top panel of Fig 4d shows inferred localization of CLDN6 and MHC from immunohistochemistry. Wider view of the section is necessary to provide distinction between staining patterns observed at injured compared to uninjured sites. This is important to advance the authors' claims about mechanism and about effects of experimental manipulations. If CLDN6 is a true pan-epicardium reporter, then it should be present in ALL epicardium cells not just at the injury site. If this isthe case, the toxin SHOULD cause damage across the whole epicardium, not just the injury site, considering how it is administered.*

We thank the reviewer for this insightful comment. We now revised **Fig.4** to include detailed expression analysis of CLDN6 before and after the injury, with a focus on the uninjured sites. In the original manuscript, we utilized two different antibodies against CLDN6, both of which worked well on methanol-fixed fresh frozen tissue and allowed analysis of cell-cell junctions. During the revision period, we identified a monoclonal antibody against CLDN6 that works on paraformaldehyde-fixed salamander tissue and complements our previous immunofluorescence experiments. In the revised **Fig. 4**, we now show membranous CLDN6 expression uniformly marking the uninjured epicardium (**overview and close up images in Fig. 4a**). In line with our *in-situ* hybridization results, at 7-dpci, expression of CLDN6 protein is reduced in the uninjured sites (**Fig. 4c**). Interestingly, we detected cytoplasmic CLDN6 in cells located subepicardially at this time-point, suggesting that localization of CLDN6 shifts during epithelial-to-mesenchymal transition, as epicardium-derived cells are moving towards the injury area (**Fig. 4e**). Importantly, we did not detect CLDN6 expression in the myocardium (**Extended Data Fig. 8a**). Taken together, we describe CLDN6 expression in greater detail in the results section of the revised manuscript (**Page 10**). The data show that CLDN6 is a pan-epicardium marker before the injury, it is downregulated following the injury and is present in epicardium-derived cells in the subepicardial region and the injury area. As mentioned earlier, we also present new data showing that CPE targets the epicardium in a dose dependent manner (**Extended Data Fig. 9c, please see further below**), further confirming the role of CLDN6 as a true pan-epicardium reporter.

- ***Arrowhead is missing in Extended Data Fig. 5d. Moreover, the histology seems very different than that of Extended Data Fig. 5c. Better quality of images of cd248 and cldn6 in situ hybridizations are necessary for clear visualization.***

We would like to thank the reviewer for pointing out the different histological appearance of the sections in Extended Data Fig. 5c and 5d. This was due to different tissue preparation methods (fixed frozen vs. fresh frozen tissue). We repeated the *Cd248/Cldn6 in situ* hybridization on fixed frozen tissue sections and replaced the panel which is now presented as **Extended Data Fig. 7e**.

• The data trends suggest that both epicardium and sup-epicardium are affected beyond the injury site, but this result is deemed insignificant. The authors contend that this is the expected result, when it is unclear to the reader why we should expect the toxin to only have an impact at the injury site. Without this clarification, this result seems to undermine the author's claim that CLDN6 is a true epicardium marker – unless the increased epicardial and sub-epicardial cell death outside of the injury zone is in fact significant. These issues raise questions without explanations. One possibility is that everything is boosted at the injury site anyway, and the error bars are bigger; the second possibility is that there is a jump in the epicardium and the subepicardium, which is deemed insignificant even though these distributions are much tighter. Given these results, why do the authors try to make the hard claim that the effect is restricted to the epicardium at the injury site? A more accurate interpretation of their results should be presented.

We thank the reviewer for pointing out the need to clarify the cellular target of CPE. In the revised manuscript, together with more detailed CLDN6 protein expression (please see above), we explore the dose dependent effects of the toxin. We now show that the epicardium is also a target of the toxin, but at a much higher dose. We believe this is due to the stoichiometric relationship between the toxin and membranous CLDN6 receptor present in the different CLDN6⁺ cell states (epicardial, subepicardial versus honeycomb cells in the injury area). Taking the reviewer's suggestion into account, we revised

our interpretation of the results as follows “Treatment with low dose CPE ($1.8 \mu\text{g g}^{-1}$) resulted in a significant increase in TUNEL⁺ cells that was distinctly concentrated to the injury area, with little effect observed across the epicardium and subepicardium. This suggested that CLDN6⁺ epicardium-derived cells in the injury area, rather than the epicardium itself were preferentially targeted by wt-CPE at this dose. In comparison, treatment with high dose CPE ($10 \mu\text{g g}^{-1}$) also caused cell death in the epicardium, confirming the role of CLDN6 as a pan-epicardium marker (Extended Data Fig.9c). CLDN6 staining confirmed the identity of the dying cells within the injury area (Extended Data Fig. 9d). No effect was observed across the myocardium of the heart regardless of the dose injected, confirming the specificity of the toxin (Fig. 5c, Extended Data Fig. 9a-c).” (Page 11, 12)

- *The authors demonstrate that treatment with wt-CPE showed increased TUNEL⁺ cells in injury area effecting epicardium-derived CLDN6⁺ cells. However, Extended Data Fig. 7a and b does not show CLDN6 and TUNEL co-staining which is invaluable for supporting the reduction of CHERRY⁺ epicardium-derived cells in response to wt-CPE.*

We agree with the reviewer that showing CLDN6/TUNEL co-staining is crucial. We provide this data in therevised version (Extended Data Figure 9d).

- *Extended Data Fig. 8c is confusing and unclear. According to color scheme, Cluster 6 which are T-cells show expression of muscle genes including MYL3, MYL4, and TCNN1 with pseudotime. Further clarification is required for pseudotime data analysis.*

We apologize for the unclear legend. The “Cluster” in Extended Fig. 8c (Extended Data Fig. 12d in the revised manuscript) refers to the hierarchical clustering of differentially expressed genes and is independent from the UMAP clustering as shown in Fig. 6 and Extended Data Fig.12a. We added the

following sentence to the figure legend to make this clear to the readers “**Note that “cluster” refers to the hierarchical clustering of differentially expressed genes and not the UMAP clusters**”.

- *The authors identified that during cardiac homeostasis, low levels of epicardial cell conversion into myocytes. It is mentioned in the discussion that epicardial cells contribute to different cell lineages including fibroblasts, perivascular cells, smooth muscle cells, and adipocytes. Thus, further discussion is necessary in regards to whether myocardial progenitor cells function as multipotent cells.*

We agree with the reviewer that this is an important point that needs further investigation. A detailed analysis is planned in future studies but we performed additional immunostainings for the present revision as well. We found epicardium-derived CHERRY⁺ cells that expressed the mesenchymal marker Vimentin (CHERRY⁺ MHC⁻ Vimentin⁺) and α -Smooth muscle actin (CHERRY⁺ MHC⁻ α -SMA⁺) indicating that CLDN6⁺ epicardial progenitors have multilineage potential. We did not detect any CHERRY⁺ Isolectin-B4⁺ endothelial cells. We incorporated this data into **Extended Data Fig. 4f-i** and updated the results section accordingly: “**Aside from the epicardium-derived cardiomyocytes, we occasionally observed CHERRY⁺ cells co-expressing the mesenchymal marker Vimentin or the smooth muscle cell/myofibroblast marker α -Smooth muscle actin, but not the endothelial cell marker Isolectin-B4, indicating that epicardial cells also give rise to non-myocyte lineages (Extended Data Fig.4f-i)**”. (Line 145-149)

- *The differences between conditions in the supplemental videos are not readily apparent to the untrained eye. This could be rectified by visually highlighting and emphasizing the relevant features. Perhaps they could present a repeat of the content of each video twice, once with annotations and once without. A supplemental figure with annotated static images to complement the videos may also suffice.* We would like to thank the reviewer for the excellent suggestions. In the revised version we

provide still images that complement the videos (**Extended Data Fig.11a**). We would like to highlight that we replaced some of the movies in the original manuscript to be able to provide the clearest still images.

- *The authors reiterate that epicardial cells support regeneration in many ways unrelated to their proliferation. Actually, this might undermine one of their conclusions, because the toxin could also be disrupting these other processes, not just direct cellular contributions. They should definitely acknowledge this possibility, especially for the manipulations where they actually kill the cells. They should also acknowledge the limitations of the study.*

According to this suggestion of the reviewer we modified the following sentence. "Taken together, these data show that the impaired regeneration, **at least in part**, caused by the reduction of epicardium-derived cardiomyocytes, is not compensated by contribution of new cardiomyocytes from elsewhere". (**Line 297**)

To further acknowledge the limitations of the study we also added the following paragraph to the discussion section (please also see above):

"The specificity of the Cre-mediated recombination is a key element of the present study for which we present two major lines of evidence. First, nuclear Cre is detected only in epicardium. Small amount of Cre protein diffusing into the myocardium is sequestered in the extracellular matrix, which precludes transduction of non-epicardial cells (**Extended Data Fig. 2d**)³⁶. Second, by functional assays we show that partial ablation of lineage labelled epicardial cells reduces the number of cardiomyocytes carrying the label both under homeostasis, and during regeneration (**Fig. 5l, m and Extended Data Fig. 10a-e**). It has been reported in other species that transient homotypic or heterotypic cell fusion could trigger cell cycle re-

entry of cardiomyocytes^{68,69}. Theoretically, contribution from such a mechanism cannot fully be discounted in salamanders either. However, the fact that regeneration is inhibited by the non-cytotoxic c- CPE in the absence of an effect on cardiomyocyte proliferation further supports the model that the cellular contribution by the epicardium rather than cell fusion between epicardial cells and myocytes is essential for cardiac regeneration in salamanders (Fig. 7h, i)."

Reviewer #2:

Remarks to the Author:

The study by Eroglu et al examines epicardial activation and cardiac regenerative mechanisms in the salamander *Pleurodeles waltl*. The authors employ novel and innovative approaches to trace cell lineages and perform gene expression profiling at single cell resolution after cardiac injury. Derivation of a small population of cardiac myocytes from epicardially labeled cells was observed at homeostasis and after injury. In addition, Claudin-6 and tight junctions of epicardial cells were found to be important in the regenerative response.

While there is strong novelty and potential interest in these mechanistic studies of salamander heart regeneration, there are some limitations in the approach and interpretations that are narrowly focused on epicardial cell conversion into cardiomyocytes.

Comments.

1) A recombinant Tat-Cre fusion protein was used for cell lineage tracing of epicardial cells. While strong nuclear Cre was observed predominantly in epicardial cells 30 hours after injection, there seems to still be Cre protein present later. There is not a clear demonstration of no nuclear Cre expression in myocytes after 30 hours and it seems possible that some of these cells could be

recombined without going through the epicardial intermediate at baseline.

We agree with the reviewer that the data on epicardium specific Cre-mediated recombination is a critical point. We have therefore performed new experiments and provided additional explanations to data presented in the original version of the manuscript. In the revised version, we clarify the distinction between nuclear localization and extracellular localization of Cre. Importantly, nuclear Cre is only found in the epicardial cells while Cre is exclusively found in the extracellular space in the myocardium. This is in line with previous observations of TAT-eGFP fusion protein binding to the extracellular matrix upon injection into muscle (Caron et al., 2001, Molecular Therapy). In fact, the transduction domain of the TAT protein is a heparin binding protein that interacts with heparan sulfate, proteoglycans of the cell surface and extracellular matrix (Rusnati et al., 1998, J. Biol. Chem). This affinity between the transduction domain of TAT and extracellular matrix surrounding the muscles interferes with the intracellular transduction process (Caron et al., 2001, Molecular Therapy). The following three experiments in the revised version further shows that nuclear Cre is exclusive to the epicardium. 1) Quantification of the Cre signal in the epicardial and myocardial cell nuclei at multiple time-points, (2) Co-immunostaining of Cre with wheat germ agglutinin (WGA) that marks the myocardial cell membranes and extracellular matrix, (3) Utilization of CPE to ablate TAT-Cre labelled epicardial cells under homeostasis to assess the impact on number of labelled cardiomyocytes.

- (1) Quantification of Cre signal in the epicardial and myocardial cell nuclei at multiple time-points.

Automated image quantification using ImageJ showed that at 30-hours post TAT-Cre injection (30-hpi) 25% of epicardial cell nuclei were Cre positive. Percentage of Cre⁺ epicardial cell nuclei decreased to 6.32% and 0.64% at 40- and 96-hpi respectively. In stark contrast to the epicardial nuclei, we did not detect any Cre

expressing nuclei in the myocardium among the 7134 cell nuclei we analyzed (1569 nuclei at 30-hpi, 1719 nuclei at 40-hpi, 3846 nuclei at 96-hpi) (**Extended Data Fig. 1b**). These findings are in line with qualitative observations we made in the original manuscript and complement those quantitatively to show that TAT-Cre does not transduce myocardial cells.

(2) Co-immunostaining for Cre and wheat germ agglutinin (WGA) that marks the myocardial cell membranes and extracellular matrix. Co-immunostaining of Cre with wheat germ agglutinin at 96-hpi confirmed the localization of Cre to the extracellular matrix surrounding the cardiac muscle (**Extended Data Fig. 2d**).

(3) CPE-mediated ablation of Tat-Cre labelled epicardial cells under homeostasis. To further test whether epicardial cells are indeed a source of labelled cardiomyocytes under homeostasis, we performed a CPE-mediated loss-of-function experiment. As a first step, we extended our understanding of the cellular target of the CPE, a point also raised by the reviewer. In the original manuscript, we reported the use of CPE to ablate epicardium-derived intermediate cells in the injury area (**Fig. 5c and Extended Data Fig. 9a, b**). We now provide additional data showing that epicardium is also a target of CPE toxin, but only at a much higher dose (**Extended Data Fig. 9c**). While we cannot fully explain the dose dependent sensitivity of epicardium versus epicardium-derived intermediate cells, we provide data describing reduced CLDN6 expression in post-injury epicardium (**Fig. 4a, c**), that might be responsible for reduced sensitivity to the toxin. Critically, the effect of CPE is confined to the epicardium and its immediate derivatives, and we did not observe any myocyte death at any dose we injected (**Fig. 5c and Extended Data Fig. 9a-c**). While high dose CPE was effective at ablating the epicardium, animals did not tolerate intraperitoneal injection. Therefore, we implemented

microinjection of the toxin into the pericardial cavity for the experiment described below that further proves that Cre-mediated recombination is restricted to the epicardium. First, we labelled epicardial cells of $P_w^{GFP-loxP-Cherry}$ reporter animals with TAT-Cre recombinase. Second, 40 hours post TAT-Cre injection, we microinjected 200 ng of wt-CPE to the pericardial cavity (**Extended Data Fig. 10a**). This resulted in a ~5.4-fold reduction of CHERRY⁺ epicardial cells as assessed 4 days and 8 hours later (6 days post TAT-Cre injection). In conjunction with this reduction, we observed a ~4-fold decrease in the number of labelled cardiomyocytes, confirming that epicardium gives rise to cardiomyocytes under homeostasis (**Extended Data Fig. 10b-e**).

Taken together: TAT-Cre transduces epicardial cells with decreasing efficiency over the course of 96 hours. We find no indication of myocardial presence of nuclear Cre signal. Lastly, depleting the labelled epicardial cells results in significantly less labelled myocytes. We believe these additional data provide sufficient additional support for the conclusion that epicardial cells give rise to cardiomyocytes.

These data have been incorporated in **Extended Data Fig. 1, 2, 9 and 10** and are presented in the results part of the manuscript. We also added the following paragraph to the Discussion:

“The specificity of the Cre-mediated recombination is a key element of the present study for which we present two major lines of evidence. First, nuclear Cre is detected only in epicardium. Small amount of Cre protein diffusing into the myocardium is sequestered in the extracellular matrix, which precludes transduction of non-epicardial cells (**Extended Data Fig. 2d**)³⁶. Second, by functional assays we show that partial ablation of lineage labelled epicardial cells reduces the number of cardiomyocytes carrying the label both under homeostasis, and during regeneration (**Fig. 5l, m and Extended Data Fig. 10a-e**). It has been reported in other species that transient homotypic or heterotypic cell fusion could trigger cell cycle re-

entry of cardiomyocytes^{68,69}. Theoretically, contribution from such a mechanism cannot fully be discounted in salamanders either. However, the fact that regeneration is inhibited by the non-cytotoxic c- CPE in the absence of an effect on cardiomyocyte proliferation further supports the model that the cellular contribution by the epicardium rather than cell fusion between epicardial cells and myocytes is essential for cardiac regeneration in salamanders (**Fig. 7h, i**).” (**Page 16, 17**)

2) There are no studies demonstrating the localization of Cre protein expression after injury. Since the Tat-Cre fusion protein is not intrinsically epicardial specific, additional demonstration of its specific localization in the injury setting is needed to support claims of epicardial origin of recombined cardiomyocytes.

We agree with the reviewer that assessing Cre protein expression after injury is an important point. In the revised manuscript, we perform Cre immunostaining on TAT-Cre injected $Pw^{Cherry-loxP-H2B::GFP}$ animals 5 hours post cryoinjury (hpci) to assess Cre protein localization (**Extended Data Fig. 3c**). We performed the injuries 40 hours post TAT-Cre injection. The Cre staining pattern we observe after the injury is indeed very similar to 40 hpi time-point (homeostasis), with patches of non-nuclear Cre staining in the epicardium and some non-nuclear signal in the myocardium as described for homeostasis. Importantly, we did not observe any Cre staining in the injury border muscle or the injury area (**Extended Data Fig. 3d**). We also complemented this with automated image quantification to show absence of nuclear Cre signal across the injury area and the myocardium (**Extended Data Fig. 3e**). Taken together, we did not observe any injury-induced leakage and transduction of the bordering myocardium, or the presumed dedifferentiated cardiomyocytes present in the injury area and added the following sentence to the result section: **“We confirmed lack of nuclear Cre signal in the myocardium following injury (Extended Data Fig. 3c-e).”** (**Line 129, 130**)

3) Were other cell types derived from epicardial recombined lineages observed? Under homeostatic conditions, were cardiac fibroblasts found to arise from the epicardial lineage? Additional information is needed on the epicardial-derived lineages after injury and in the NucBow experiments.

We would like to thank the reviewer for raising this important question. We now performed additional immunostainings (Isolectin-B4 to probe for endothelial cells, Vimentin for mesenchymal cells, α -SMA for smooth muscle cells/myofibroblasts) to investigate whether other cell types were derived from the recombined epicardial cells at 21 dpci. While we did not observe any CHERRY⁺Isolectin-B4⁺ cells, we found CHERRY⁺ Vimentin⁺ cells, as well as CHERRY⁺ α -SMA⁺ cells indicating multipotency of the epicardial progenitors. These data are presented in **Extended Data Fig. 4f-i** and reported in the manuscript as: **“Aside from the epicardium-derived cardiomyocytes, we occasionally observed CHERRY⁺ cells co-expressing the mesenchymal marker Vimentin or the smooth muscle cell/myofibroblast marker α -Smooth muscle actin, but not the endothelial cell marker Isolectin-B4, indicating that epicardial cells also give rise to non-myocyte lineages (Extended Data Fig.4f-i)”**. (Line 145-149) Future studies will look deeper into the multilineage potential of the salamander epicardial progenitors under various conditions.

4) What is the identity of the “honeycomb” like cells in the subepicardial injury area? Are they endothelial? Do they express specific ECM proteins? Are they part of the scar?

We would like to thank the reviewer for pointing out that we did not make this question sufficiently clear in the original manuscript. In fact, the single cell transcriptome of the intermediate cells (**Fig. 6**) are representing honeycomb forming cells. In the revised manuscript we add the following sentence “In order to molecularly profile the epicardial to cardiac muscle cell conversion and **determine the identity**

of honeycomb-forming cells, we performed scRNA sequencing on live cells isolated by FACS from the injury area of vehicle- versus wt-CPE-treated hearts at 7-dpci (Fig. 6a)." (Line 301, 302)

To assess whether these cells are endothelial, we performed Isolectin-B4/CLDN6 co-staining. We did not observe any Isolectin-B4⁺, CLDN6⁺ cells which indicate that they are not endothelial cells (Extended Data Fig. 12c).

Taking advantage of the single cell transcriptome data we identified specific expression of *Tenascin-X*, *Fibulin-5* and *Col6a1/2* in this cluster, suggesting that these cells deposit extracellular matrix molecules (Extended Data Fig. 12b).

These new findings are reported on Page 14 of the revised manuscript.

5) A cluster of cells expressing Gata4/6 and Foxc1/2 is identified as a potential source of cardiomyocytes.

However, these markers are also expressed in endothelial cell lineages, and more definitive markers of cardiomyocytes (Nkx2.5, Mef2, Tbx5) were not observed in this cluster. Potential endothelial fates of this population should be considered.

We agree with the reviewer that *Gata4/6* and *Foxc1/2* are not specific markers of the cardiomyocyte lineage. Therefore, as suggested by the reviewer, we performed Isolectin-B4 immunostainings to consider the endothelial fates of the intermediate cell population. However, we did not find any epicardium-derived Isolectin-B4⁺ cells at 21-dpci (Extended Data Fig. 4h), indicating that these cells do not adopt endothelial fate.

6) It is almost certain that the impaired regenerative response in the CPE injected hearts is due to more than the loss of the small population of epicardial-derived cardiomyocytes. Vascularization and fibroblast contributions to the scar ECM and resolution also should be monitored.

We completely agree with the reviewer that the epicardium and epicardium-derived cells are likely to play several roles during cardiac regeneration also in salamanders. We refer to findings supporting this possibility in the Discussion: *“Epicardial cells secrete paracrine factors to other cell types, including cardiomyocytes that in the regenerative zebrafish reenter the cell cycle and proliferate at large scale to replace lost cardiomyocytes¹⁹. Epicardial cells also produce extracellular matrix components which are necessary for re-vascularization and muscle regeneration⁶⁶.”*

In addition, new findings made during the revision describing a small population of epicardium-derived cells expressing markers indicative of fibroblasts/myofibroblasts, further supports the possibility that CPE treatment impinges on scar formation (**Extended Data Fig.4f-i**). We also modified a summary statement in the Results section: *“Taken together, these data show that the impaired regeneration, **at least in part**, caused by the reduction of epicardium-derived cardiomyocytes is not compensated by contribution of new cardiomyocytes from elsewhere.” (Line 297)*

While there is reason to believe that, like in other species, the salamander epicardium, does have multiple functions, we are aware that we have not addressed these functions experimentally in due details. We also agree that it is important to discern both in quantitative and qualitative terms the different roles the epicardium and its derivatives have in salamanders both during homeostasis and after injury, which we now explicitly spelled out in the Discussion *“**In addition to cardiomyocytes, we observed epicardium-derived cells co-expressing the mesenchymal marker Vimentin or the smooth muscle cell/myofibroblast marker α -Smooth muscle actin indicating that epicardial cells also give rise to non-myocyte lineages. It will be important in the future to discern both in quantitative and qualitative terms the different roles the epicardium have in salamanders both during homeostasis and after injury.**” (Page*

16) We believe that the appropriate studies would require substantial in-depth analyses which are out of the scope of the present work.

7) *The observed epicardial-derived cardiomyocyte population is a small minority of new myocytes.*

Are these cells fundamentally different from the majority of new cardiomyocytes after

regeneration? Also, is there evidence that the majority of new cardiomyocytes come from existing cardiomyocytes?

We agree with the reviewer that these are important questions. Currently, we find that epicardium-derived cardiomyocytes are morphologically indistinguishable from the rest of the myocytes and that they integrate into the myocardium in the regenerated area. At this point, we do not know whether there are molecular differences between epicardium-derived cardiomyocytes versus the rest and we are very motivated to investigate this question in future studies.

Regarding the origin of the remaining regenerated cardiomyocytes, we do not currently have a muscle-specific Cre line to answer this question and we cannot rule out alternative sources.

Minor comments:

1) While some investigators have reported conversion of epicardial cells to cardiomyocytes in mammals, this area is still controversial and I do not think that the field has reached a consensus as stated in line 54 of the introduction.

We would like to thank the reviewer for this insightful comment. In the revised version of the manuscript, we toned down the claim by saying “Nevertheless, **it has been reported** that upon stimulation with factors such as thymosin- β 4 and VEGF, epicardial cells differentiate into cardiomyocytes at a low frequency indicating the regenerative potential of the epicardium.” **(Line 54-56)**

Reviewer #3:**Remarks to the Author:**

*This manuscript described the differentiation of epicardial cells towards cardiomyocytes in the regenerating salamander species *Pleurodeles waltl*.*

*While humans and most mammals cannot efficiently regenerate cardiac muscle after injury, other vertebrate species have been described with a high regenerative capacity and the ability to replace lost cardiomyocytes with new myocardium. The cellular mechanism underlying this regenerative capacity seems to be predominantly in the ability of cardiomyocytes from the adult heart to reenter cell cycle, divide and give rise to daughter cells that then replenish the lost myocardium. Here the authors describe an alternative mechanism of myocardium regeneration in the newt *Pleurodeles waltl*, in which cells within the epicardial layer divide and give rise to cardiomyocytes. This process has also been described previously to occur upon Thymosin beta 4 administration in the mouse (doi: 10.1038/nature05383), but the article has been widely disputed in the field. Finding a similar mechanism occurring naturally in an animal species of high regenerative capacity is therefore very surprising and will be of great interest to the community.*

To identify conversion of epicardial cells into cardiomyocytes the authors use a nuclear-tagged Cre protein into the pericardial cavity. They show several control experiments to rule out leakiness of the flexed reporter line and unwanted Cre uptake of non-epicardial cells. They find that some epicardial cells give rise to underlying cardiomyocytes. This occurs during homeostasis but is enhanced upon ventricular cryoinjury. They also perform a rigorous quantification of this event and add multicolor fate mapping to study clonality.

They next try to understand more on epicardial gene signature and genes involved in this process. For this they make use of scRNAseq and identify several cell types in the new heart, describing also new markers for epicardial cells. They identify Claudin6 as a new epicardial cell marker. Using a specific toxin (CPE) blocking claudin6 they could show that this protein is not only an epicardial marker but is also playing an important role in allowing the conversion of epicardial cells to cardiomyocytes and supporting heart regeneration. They confirm that the role of this protein is related to the establishment of tight junctions through experiments using toxin variants. Methodology is sound and of high-standard: I particularly like the elegant methods to assess amount of cryoinjured area through longitudinal ultrasound measurements.

The data are well-presented and conclusions based on robust

data. Suggested improvements:

- I would recommend also citing the following paper when talking about previous results studying the fate of epicardial cells during heart regeneration in the zebrafish Dev Biol. 2012 Oct 15;370(2):173-86. doi: 10.1016/j.ydbio.2012.07.007, since here authors did not solely rely on genetic lineage tracing but included transplantations assays.*

We thank the reviewer for the recommendation and fully agree that this is a landmark study in terms of determining epicardial cell fate. We now added this reference and revised the introduction to incorporate the following sentence “Genetic lineage tracing and **transplantation** studies in zebrafish did not provide evidence for epicardial cell differentiation into cardiomyocytes.” **(Line 50)**

- I found it surprising that the authors observe a decrease in epicardial cell numbers labelled in response to injury. What has been reported in other species is that epicardial cells become very*

proliferative in response to cardiac injury. Is the explanation that they leave the epicardium and contribute to cardiomyocytes in the newt? Please provide some discussion.

We apologize for not explaining our experimental reasoning better. Salamander epicardium also becomes proliferative in response to injury. However, we aimed to quantify the number of labelled epicardial cells before the initiation of proliferation in response to the injury, to get a closer approximation of our starting population. In the revised manuscript we provide data showing an EdU pulse-chase experiment where hearts were either injected with EdU at 24- and 48-hours post cryoinjury or at 24-, 48- and 72-hours post cryoinjury. Hearts injected at 24 and 48 hours did not show pan-CK⁺, EdU⁺ cells, while hearts that received a third injection at 72 hours had EdU⁺ epicardial cells, letting us to conclude that epicardial proliferation starts around 3 dpci (**Extended Data Fig. 4a**). Taken together, it is no surprise that the number of labelled epicardial cells decrease in response to cryoinjury as assessed before the initiation of epicardial proliferation.

• The authors provide several control experiments to prove that Cre recombination is occurring only in the outer epicardial layer, and that therefore if recombined cells are seen elsewhere they have to be derived from those epicardial cells. However, I would like to see in the discussion section a paragraph of (although unlikely) alternative possibilities: There are reports on cell fusion occurring after heart injury, maybe small amounts of Cre (not to be detected by immunostaining) could pass to cardiomyocytes? e.g. doi: 10.1038/s41467-017-01555-8; doi: 10.1096/fj.201902105R.)

We would like to thank the reviewer for encouraging us to acknowledge alternative possibilities and bringing this interesting literature to our attention. We now revised the discussion section to add the following paragraph that also cites the relevant literature:

“The specificity of the Cre-mediated recombination is a key element of the present study for which we present two major lines of evidence. First, nuclear Cre is detected only in epicardium. Small amount of Cre protein diffusing into the myocardium is sequestered in the extracellular matrix, which precludes transduction of non-epicardial cells (**Extended Data Fig. 2d**)³⁶. Second, by functional assays we show that partial ablation of lineage labelled epicardial cells reduces the number of cardiomyocytes carrying the label both under homeostasis, and during regeneration (**Fig. 5l, m and Extended Data Fig. 10a-e**). It has been reported in other species that transient homotypic or heterotypic cell fusion could trigger cell cycle re-entry of cardiomyocytes^{68,69}. Theoretically, contribution from such a mechanism cannot fully be discounted in salamanders either. However, the fact that regeneration is inhibited by the non-cytotoxic c-CPE in the absence of an effect on cardiomyocyte proliferation further supports the model that the cellular contribution by the epicardium rather than cell fusion between epicardial cells and myocytes is essential for cardiac regeneration in salamanders (**Fig. 7h, i**).” (**Page 16, 17**)

Furthermore, we have performed new experiments and provided additional explanations to data presented in the original version of the manuscript. In the revised version, we clarify the distinction between nuclear localization and extracellular localization of Cre. Importantly, nuclear Cre is only found in the epicardial cells while Cre is exclusively found in the extracellular space in the myocardium. This is in line with previous observations of TAT-eGFP fusion protein binding to the extracellular matrix upon injection into muscle (Caron et al., 2001, Molecular Therapy). In fact, the transduction domain of the TAT protein is a heparin binding protein that interacts with heparan sulfate, proteoglycans of the cell surface and extracellular matrix (Rusnati et al., 1998, J. Biol. Chem). This affinity between the transduction domain of TAT and extracellular matrix surrounding the muscles interferes with the intracellular transduction

process (Caron et al., 2001, Molecular Therapy). The following three experiments in the revised version further shows that nuclear Cre is exclusive to the epicardium. 1) Quantification of the Cre signal in the epicardial and myocardial cell nuclei at multiple time-points, (2) Co-immunostaining of Cre with wheat germ agglutinin (WGA) that marks the myocardial cell membranes and extracellular matrix, (3) Utilization of CPE to ablate TAT-Cre labelled epicardial cells under homeostasis to assess the impact on number of labelled cardiomyocytes.

- (1) Quantification of Cre signal in the epicardial and myocardial cell nuclei at multiple time-points.

Automated image quantification using ImageJ showed that at 30-hours post TAT-Cre injection (30-hpi) 25% of epicardial cell nuclei were Cre positive. Percentage of Cre⁺ epicardial cell nuclei decreased to 6.32% and 0.64% at 40- and 96-hpi respectively. In stark contrast to the epicardial nuclei, we did not detect any Cre expressing nuclei in the myocardium among the 7134 cell nuclei we analyzed (1569 nuclei at 30-hpi, 1719 nuclei at 40-hpi, 3846 nuclei at 96-hpi) (**Extended Data Fig. 1b**). These findings are in line with qualitative observations we made in the original manuscript and complement those quantitatively to show that TAT-Cre does not transduce myocardial cells.

- (2) Co-immunostaining for Cre and wheat germ agglutinin (WGA) that marks the myocardial cell membranes and extracellular matrix. Co-immunostaining of Cre with wheat germ agglutinin at 96-hpi confirmed the localization of Cre to the extracellular matrix surrounding the cardiac muscle (**Extended Data Fig. 2d**).

- (3) CPE-mediated ablation of Tat-Cre labelled epicardial cells under homeostasis. To further test whether epicardial cells are indeed a source of labelled cardiomyocytes under homeostasis, we performed a CPE- mediated loss-of-function experiment. As a first step, we extended our

understanding of the cellular target of the CPE, a point also raised by the reviewer. In the original manuscript, we reported the use of CPE to ablate epicardium-derived intermediate cells in the injury area (**Fig. 5c and Extended Data Fig. 9a, b**). We now provide additional data showing that epicardium is also a target of CPE toxin, but only at a much higher dose (**Extended Data Fig. 9c**). While we cannot fully explain the dose dependent sensitivity of epicardium versus epicardium-derived intermediate cells, we provide data describing reduced CLDN6 expression in post-injury epicardium (**Fig. 4a, c**), that might be responsible for reduced sensitivity to the toxin. Critically, the effect of CPE is confined to the epicardium and its immediate derivatives, and we did not observe any myocyte death at any dose we injected (**Fig. 5c and Extended Data Fig. 9a-c**). While high dose CPE was effective at ablating the epicardium, animals did not tolerate intraperitoneal injection. Therefore, we implemented microinjection of the toxin into the pericardial cavity for the experiment described below that further proves that Cre-mediated recombination is restricted to the epicardium. First, we labelled epicardial cells of $Pw^{GFP-loxP-Cherry}$ reporter animals with TAT-Cre recombinase. Second, 40 hours post TAT-Cre injection, we microinjected 200 ng of wt-CPE to the pericardial cavity (**Extended Data Fig. 10a**). This resulted in a ~5.4-fold reduction of CHERRY⁺ epicardial cells as assessed 4 days and 8 hours later (6 days post TAT-Cre injection). In conjunction with this reduction, we observed a ~4-fold decrease in the number of labelled cardiomyocytes, confirming that epicardium gives rise to cardiomyocytes under homeostasis (**Extended Data Fig. 10b-e**).

Taken together: TAT-Cre transduces epicardial cells with decreasing efficiency over the course of 96 hours. We find no indication of myocardial presence of nuclear Cre signal. Lastly, depleting the labelled epicardial cells results in significantly less labelled myocytes. We believe these additional data provide

sufficient additional support for the conclusion that epicardial cells give rise to cardiomyocytes.

These data have been incorporated in **Extended Data Fig. 1, 2, 9 and 10** and are presented in the results part of the manuscript.

- ***Authors show how many epicardial-derived cardiomyocytes are PCNA-positive but: how many non-epicardial-derived cardiomyocytes are PCNA positive? Understanding the ration would be interesting. Also, is BrdU labelling possible in the newt? I would prefer to see this kind of labelling as PCNA also labels senescent cells “stuck in the cell cycle”.***

We agree with the reviewer that understanding the ratio would be interesting. Unfortunately, currently we do not have a lineage tracing method that allows specific labelling of the cardiomyocytes. Considering that our labelling of the epicardium-derived cardiomyocytes is not 100% efficient, we would not be able to distinguish between epicardial vs. non-epicardial-derived cardiomyocytes. However, in the revised version of the manuscript we show expression of the mitotic marker Phospho-histone H3 in epicardium-derived cardiomyocytes (**Extended Data Fig. 5j**) demonstrating progression of the cell cycle. We state this finding in the results section as follows: “Furthermore, we found that 22% of epicardium-derived cardiomyocytes were still PCNA⁺ at 21-dpci, **with occasional expression of the M-phase marker Phospho-Histone H3 (Extended Data Fig. 5g-j)**, suggesting that clonal expansion could result from proliferation of epicardium derived cells expressing cardiomyocyte markers.” (**Line 156**)

- ***Statement on line 265. “show that the impaired regeneration caused by the reduction of epicardium- derived cardiomyocytes”. This needs to be reformulated. The authors show that regeneration is incomplete upon wt-CPE treatment and that there is less number of Cherry1 cardiomyocytes. This shows that both effects of wt-CPE correlate but not necessarily that the sole***

reason that the heart does not regenerate is due to the lack of cherry+ cells. Indeed, the authors discuss this later on in lines 349-350. We agree with the reviewer that the phenotype caused by the ablation of CLDN6⁺ epicardium-derived cells by wt-CPE treatment is complex and to acknowledge that, following the reviewer's suggestion we revised that sentence as follows: "Taken together, these data show that the impaired regeneration, **at least in part**, caused by the reduction of epicardium-derived cardiomyocytes is not compensated by contribution of new cardiomyocytes from elsewhere." **(Line 297)**

We also modified the Discussion as stated above.

- ***Figure 6: For images shown in panel h or single channels need to be shown. I could not see well the myl3; gata6 stained Cherry cells.***

Following the reviewer's suggestion, we provide single channel images for *Myf3* and *Gata6* (**Fig. 6h**) in the revised manuscript.

- ***Figure 7: CPE bind to Claudin6. Can the authors show a co-labelling of CPE with Claudin 6 for example in Figure 7a?***

In the revised manuscript, we now include a panel to show co-labelling of CLDN6 and c-CPE, confirming the binding of the toxin to CLDN6 positive cells (**Fig. 7a**).

- ***Extended Data Fig 2: In panel b show also still images without the yellow highlight. In the legends authors refer to data not shown in epicardial cells proliferation at 48 hpci. Is it allowed to refer to non- shown data?? I strongly recommend to show all data.***

We would like to thank the reviewer for suggesting this modification to improve the visibility of the heart's apical region. In the revised manuscript we show still images without the yellow highlight (**Extended Data Fig.3b**). Regarding the data not shown, as mentioned earlier, we show in the revised manuscript that

epicardial proliferation starts around 3 days post cryoinjury (**Extended Data Fig. 4a**). Nevertheless, we removed this sentence from the legend as we think it is not crucial for the message we would like to convey.

Decision Letter, first revision:

Subject: Your manuscript, NCB-C46093A
Message: Our ref: NCB-C46093A

26th January 2022

Dear Dr. Chien,

Thank you for submitting your revised manuscript "Epicardium-derived cells organize through tight junctions to replenish cardiac muscle in salamanders" (NCB-C46093A). It has now been seen by the original referees and their comments are below. The reviewers find that the paper has improved in revision, and therefore we'll be happy in principle to publish it in Nature Cell Biology, pending minor revisions to satisfy the referees' final requests and to comply with our editorial and formatting guidelines.

The current version of your manuscript is in a PDF format. Please email us a copy of the file in an editable format (Microsoft Word or LaTeX)-- we can not proceed with PDFs at this stage.

Thank you again for your interest in Nature Cell Biology. Please do not hesitate to contact me if you have any questions.

Sincerely,

Jie Wang, PhD
Senior Editor
Nature Cell Biology

Tel: +44 (0) 207 843 4924
email: jie.wang@nature.com

Reviewer #1 (Remarks to the Author):

This revised version of the paper presents additional arguments that clarify the authors' points on components of the research that were initially ambiguous as well as substantial new experimentation to address concerns raised by reviewers. All the smaller issues were addressed in the revised text. The most critical concerns have also now been rectified:

The authors have provided additional evidence that the TAT-Cre approach does not lead to labeling of cell types other than epicardial cells, addressing the possibility that leaky reporter expression in cardiomyocytes might account for the label ultimately seen there post-injury.

They also addressed the question of whether epicardium is a source of cardiomyocytes during homeostasis.

The computational work now provides stronger support for the conclusions made in the paper than in the first version. The authors have adjusted parameters, revealing attributes of the cells they describe as "transitioning," such as expression of EMT markers. They've explained the computational analyses more clearly now.

The authors have further defined the CLDN6-expressing cells using alternative reagents, now showing that indeed the entire epicardium expresses this marker rather than just the epicardium at the injury site. They also noted, using the new reagent, that CLDN6 appears in cells right beneath the epicardium, but with lower expression, in the early stages of regeneration. These findings are interesting and will help future experiments aimed at determining how the derivatives of these cells regulate formation of the transient, honeycomb ECM and, ultimately, regeneration of cardiomyocytes.

Cellular targets of CPE have been further explored. There is indeed some targeting of the epicardial cells themselves, but only at higher concentrations. These issues have been clarified in the text.

Limitations of the study are now better acknowledged in the discussion.

Overall, this paper is now one that is ready for publication. It now clearly shows the interesting (and novel) finding that epicardium-derived cells contribute to heart regeneration in a salamander, and it serves a great example of how clever experimentation can allow for addressing of cell-level questions in vivo, in an organism for which not many genetic tools exist.

Reviewer #2 (Remarks to the Author):

The authors have responded to reviewer comments and the revised manuscript is improved. There are still some limitations on identification of cell types, but my major concerns have been addressed with new data or discussed in the revised text.

Reviewer #3 (Remarks to the Author):

The revised version has addressed all my questions.

In the revised version the authors have deepened into the transcriptomic characterization of claudin-6 positive cells and rest of non-cardiomyocytes. They have included more time points and also expanded on cluster analysis. Now they present more information on the population they call transition cells in Ext. Data Fig 6. The population does not express any longer myocardial markers, which is a bit difficult to understand. Also, there are now other markers for this cluster apart from Claudin-6 that seem to better define this population. I find that this is not well discussed in the manuscript.

I am now also confused: Is the intermediate cell population the same as the transition cells? And, how does the scRNA Seq in Extended Data 6 fit with the pseudotime trajectory of CLD6+ cells show in Fig. 6? Thanks for providing clarifications.

Decision letter, final requests:

Subject: NCB: Your manuscript, NCB-C46093A
Message:

Our ref: NCB-C46093A

7th February 2022

Dear Dr. Chien,

Thank you for your patience as we've prepared the guidelines for final submission of your Nature Cell Biology manuscript, "Epicardium-derived cells organize through tight junctions to replenish cardiac muscle in salamanders" (NCB-C46093A). Please carefully follow the step-by-step instructions provided in the attached file, and add a response in each row of the table to indicate the changes that you have made. Ensuring that each point is addressed will help to ensure that your revised manuscript can be swiftly handed over to our production team.

We would like to start working on your revised paper, with all of the requested files and forms, as soon as possible (preferably within one week). Please get in contact with us if you anticipate delays.

In recognition of the time and expertise our reviewers provide to Nature Cell Biology's editorial process, we would like to formally acknowledge their contribution to the external peer review of your manuscript entitled "Epicardium-derived cells organize through tight junctions to replenish cardiac muscle in salamanders". For those reviewers who give their assent, we will be publishing their names alongside the published article.

Nature Cell Biology offers a Transparent Peer Review option for new original research manuscripts submitted after December 1st, 2019. As part of this initiative, we encourage our authors to support increased transparency into the peer review process by agreeing to have the reviewer comments, author rebuttal letters, and editorial decision letters published as a Supplementary item. When you submit your final files please clearly state in your cover letter whether or not you would like to participate in this initiative. Please note that failure to state your preference will result in delays in accepting your manuscript for publication.

Cover suggestions

As you prepare your final files we encourage you to consider whether you have any images or illustrations that may be appropriate for use on the cover of Nature Cell Biology.

Nature Cell Biology has now transitioned to a unified Rights Collection system which will allow our Author Services team to quickly and easily collect the rights and permissions required to publish your work. Approximately 10 days after your paper is formally accepted, you will receive an email in providing you with a link to complete the grant of rights. If your paper is eligible for Open Access, our Author Services team will also be in touch regarding any additional information that may be required to arrange payment for your article.

Please note that Nature Cell Biology is a Transformative Journal (TJ). Authors may publish their research with us through the traditional subscription access route or make their paper immediately open access through payment of an article-processing charge (APC). Authors will not be required to make a final decision about access to their article until it has been accepted. Find out more about Transformative Journals

Authors may need to take specific actions to achieve compliance with funder and institutional open access mandates. For submissions from January 2021, if your research is supported by a funder that requires immediate open access (e.g. according to Plan S principles) then you should select the gold OA route, and we will direct you to the compliant route where possible. For authors selecting the subscription publication route our standard licensing terms will need to be accepted, including our self-archiving policies. Those standard licensing terms will supersede any other terms that the author or any third party may assert apply to any version of the manuscript.

For information regarding our different publishing models please see our Transformative Journals page. If you have any questions about costs, Open Access requirements, or our legal forms, please contact ASJournals@springernature.com.

[REDACTED]

If you have any further questions, please feel free to contact us. Many thanks!

Best regards,

Ziqian Li
Editorial Assistant
Nature Cell Biology

On behalf of

Jie Wang, PhD
Senior Editor
Nature Cell Biology

Tel: +44 (0) 207 843 4924
email: jie.wang@nature.com

Reviewer #1:

Remarks to the Author:

This revised version of the paper presents additional arguments that clarify the authors' points on components of the research that were initially ambiguous as well as substantial new experimentation to address concerns raised by reviewers. All the smaller issues were addressed in the revised text. The most critical concerns have also now been rectified:

The authors have provided additional evidence that the TAT-Cre approach does not lead to labeling of cell types other than epicardial cells, addressing the possibility that leaky reporter expression in cardiomyocytes might account for the label ultimately seen there post-injury.

They also addressed the question of whether epicardium is a source of cardiomyocytes during homeostasis.

The computational work now provides stronger support for the conclusions made in the paper than in the first version. The authors have adjusted parameters, revealing attributes of the cells they describe as "transitioning," such as expression of EMT markers. They've explained the computational analyses more clearly now.

The authors have further defined the CLDN6-expressing cells using alternative reagents, now showing that indeed the entire epicardium expresses this marker rather than just the epicardium at the injury site. They also noted, using the new reagent, that CLDN6 appears in cells right beneath the epicardium, but with lower expression, in the early stages of regeneration. These findings are interesting and will help future experiments aimed at determining how the derivatives of these cells regulate formation of the transient, honeycomb ECM and, ultimately, regeneration of cardiomyocytes.

Cellular targets of CPE have been further explored. There is indeed some targeting of the epicardial cells themselves, but only at higher concentrations. These issues have been clarified in the text.

Limitations of the study are now better acknowledged in the discussion.

Overall, this paper is now one that is ready for publication. It now clearly shows the interesting (and novel) finding that epicardium-derived cells contribute to heart regeneration in a salamander, and it serves a great example of how clever experimentation can allow for addressing of cell-level questions in vivo, in an organism for which not many genetic tools exist.

Reviewer #2:

Remarks to the Author:

The authors have responded to reviewer comments and the revised manuscript is improved. There are still some limitations on identification of cell types, but my major concerns have been addressed with new data or discussed in the revised text.

Reviewer #3:

Remarks to the Author:

The revised version has addressed all my questions.

In the revised version the authors have deepened into the transcriptomic characterization of claudin-6 positive cells and rest of non-cardiomyocytes. They have included more time points and also expanded on cluster analysis. Now they present more information on the population they call transition cells in Ext. Data Fig 6. The population does not express any longer myocardial markers, which is a bit difficult to understand. Also, there are now other markers for this cluster apart from Claudin-6 that seem to better define this population. I find that this is not well discussed in the manuscript.

I am now also confused: Is the intermediate cell population the same as the transition cells? And, how does the scRNA Seq in Extended Data 6 fit with the pseudotime trajectory of CLD6+ cells show in Fig. 6? Thanks for providing clarifications.

Author Rebuttal, first revision:

Response to Reviewer

Reviewer #3 (Remarks to the Author):

The revised version has addressed all my questions. In the revised version the authors have deepened into the transcriptomic characterization of claudin-6 positive cells and rest of non-cardiomyocytes. They have included more time points and also expanded on cluster analysis. Now they present more information on the population they call transition cells in Ext. Data Fig 6. The population does not express any longer myocardial markers, which is a bit difficult to understand. Also, there are now other markers for this cluster apart from Claudin-6 that seem to better define this population. I find that this is not well discussed in the manuscript.

I am now also confused: Is the intermediate cell population the same as the transition cells? And, how does the scRNA Seq in Extended Data 6 fit with the pseudotime trajectory of CLD6+ cells show in Fig. 6? Thanks for providing clarifications.

We thank the reviewer for raising this issue and we now provide additional clarification of how transitioning cells relate to the intermediate cells also in terms of marker gene expression. Our data suggest that transitioning cell state precedes the intermediate cell state. Some of the transitioning cells still express the epicardial marker *Cldn6* and as the reviewer pointed out correctly, have not initiated the expression of cardiac muscle genes. In turn, intermediate cells do not express *Cldn6* mRNA and have initiated the expression of cardiac muscle genes. While both populations express the EMT genes *Snail1* and *Twist1* to a certain extent, *Snail1* expression is higher in the transitioning cells and *Twist1* expression is higher in the intermediate cells. It has been reported that *Snail1* is required for the initiation of EMT, while *Twist1* is required to maintain late EMT (PMID: 22006115). Including the transitioning cells into the pseudotime trajectory, as suggested by the reviewer, further supports the notion that the transitioning cells are preceding the intermediate cell states as they are placed at the start of the pseudotime trajectory (please see below). We provide this extended pseudotime analysis as a reviewer figure only, since we decided against putting emphasis on the transitioning cell state in the current manuscript, due to the low number of cells detected.

Transitioning cells precede intermediate cells along pseudotime. **a**, Pseudotime trajectory of transitioning cells and intermediate cells combined. Cells with dark color and bright color represent the start and end of pseudotime, respectively. Arrow indicates the directionality. **b**, Transitioning cells highlighted along the trajectory. **c**, Expression of select genes among the cells making the trajectory. *Snai1* and *Twist1* are epithelial-to-mesenchymal transition markers, *Gata4* and *Gata6* are cardiogenic transcription factors, *tnnt2e* is a cardiomyocyte marker.

Final Decision Letter:

Subject: Decision on Nature Cell Biology submission NCB-C46093B
Message:

Dear Dr Chien,

I am pleased to inform you that your manuscript, "Epicardium-derived cells organize through tight junctions to replenish cardiac muscle in salamanders", has now been accepted for publication in Nature Cell Biology.

Please note that Nature Cell Biology is a Transformative Journal (TJ). Authors may publish their research with us through the traditional subscription access route or make their paper immediately open access through payment of an article-processing charge (APC). Authors will not be required to make a final decision about access to their article until it has been accepted. Find out more about Transformative Journals

Authors may need to take specific actions to achieve compliance with funder and institutional open access mandates. If your research is supported by a funder that requires immediate open access (e.g. according to Plan S principles) then you should select the gold OA route, and we will direct you to the compliant route where possible. For authors selecting the subscription publication route, the journal's standard licensing terms will need to be accepted, including self-archiving policies. Those licensing terms will supersede any other terms that the author or any third party may assert apply to any version of the manuscript.

If your paper includes color figures, please be aware that in order to help cover some of the additional cost of four-color reproduction, Nature Research charges our authors a fee for the printing of their color figures. Please contact our offices for exact pricing and details.

If you have not already done so, we strongly recommend that you upload the step-by-step protocols used in this manuscript to the Protocol Exchange (www.nature.com/protocolexchange), an open online resource established by Nature Protocols that allows researchers to share their detailed experimental know-how. All uploaded protocols are made freely available, assigned DOIs for ease of citation and are fully searchable through nature.com. Protocols and the Nature and Nature research journal papers in which they are used can be linked to one another, and this link is clearly and prominently visible in the online versions of both papers. Authors who performed the specific experiments can act as primary authors for the Protocol as they will be best placed to share the methodology details, but the Corresponding Author of the present research paper should be included as one of the authors. By uploading your Protocols to Protocol Exchange, you are enabling researchers to more readily reproduce

or adapt the methodology you use, as well as increasing the visibility of your protocols and papers. You can also establish a dedicated page to collect your lab Protocols. Further information can be found at www.nature.com/protocolexchange/about

You can use a single sign-on for all your accounts, view the status of all your manuscript submissions and reviews, access usage statistics for your published articles and download a record of your refereeing activity for the Nature journals.

With kind regards,

Jie Wang, PhD
Senior Editor
Nature Cell Biology

Tel: +44 (0) 207 843 4924
email: jie.wang@nature.com